# Deciphering cell–cell communication at single-cell resolution for spatial transcriptomics with subgraph-based graph attention network

Wenyi Yang[1,7], Pingping Wang[2,7], Shouping Xu[3,7], Tao Wang[4], Meng Luo[1], Yideng Cai[1], Chang Xu[1], Guangfu Xue[1], Jinhao Que[1], Qian Ding[1], Xiyun Jin[2], Yuexin Yang[1], Fenglan Pang[1], Boran Pang[5], Yi Lin[2], Huan Nie[1], Zhaochun Xu[2] ✉, Yong Ji[6] ✉ & Qinghua Jiang[1,2] ✉

The inference of cell–cell communication (CCC) is crucial for a better understanding of complex cellular dynamics and regulatory mechanisms in biological systems. However, accurately inferring spatial CCCs at single-cell resolution remains a significant challenge. To address this issue, we present a versatile method, called DeepTalk, to infer spatial CCC at single-cell resolution by integrating single-cell RNA sequencing (scRNA-seq) data and spatial transcriptomics (ST) data. DeepTalk utilizes graph attention network (GAT) to integrate scRNA-seq and ST data, which enables accurate cell-type identification for single-cell ST data and deconvolution for spot-based ST data. Then, DeepTalk can capture the connections among cells at multiple levels using subgraph-based GAT, and further achieve spatially resolved CCC inference at single-cell resolution. DeepTalk achieves excellent performance in discovering meaningful spatial CCCs on multiple cross-platform datasets, which demonstrates its superior ability to dissect cellular behavior within intricate biological processes.

Cell–cell communication (CCC) is a fundamental biological process that plays a crucial role in immune cooperation, organ development, stem cell niches, and other biological phenomena[1–3]. Advances in single-cell transcriptomics, particularly single-cell RNA sequencing (scRNA-seq), have revolutionized the study of individual cells, providing unprecedented insights into their composition, function, and dynamics[4]. Computational tools such as CellPhoneDB[5], CellChat[6], NicheNet[7], CytoTalk[8], scTensor[9], iTALK[10], ICELLNET[11], SingleCellSignalR[12] and Scriabin[13] have been developed to infer and decipher CCC networks[14]. Tools like CellPhoneDB and CellChat, for instance, offer deep insights into ligand-receptor interactions (LRIs), which are pivotal in understanding signal transduction across diverse cell types. These tools employ innovative methods and statistical tests to quantify the probability of each interaction, facilitating in the assessment of the null hypothesis and inference of the specific cell types involved in intercellular communication. Additionally, NicheNet and CytoTalk reveal the

[1]Center for Bioinformatics, School of Life Science and Technology, Harbin Institute of Technology, Harbin, China. [2]School of Interdisciplinary Medicine and Engineering, Harbin Medical University, Harbin, China. [3]Department of Breast Cancer, Harbin Medical University Cancer Hospital, Harbin, China. [4]School of Computer Science, Northwestern Polytechnical University, Xi'an, China. [5]Center for Difficult and Complicated Abdominal Surgery, Shanghai Tenth People's Hospital, Tongji University School of Medicine, Shanghai, China. [6]State Key Laboratory of Frigid Zone Cardiovascular Diseases (SKLFZCD), Harbin Medical University, Harbin, China. [7]These authors contributed equally: Wenyi Yang, Pingping Wang, Shouping Xu. ✉e-mail: zhaochunxu@hrbmu.edu.cn; yongji@hrbmu.edu.cn; qhjiang@hit.edu.cn

complex gene-gene interactions within cells in response to external stimuli, while scTensor employs hypergraphs to expose the intricacies of higher-order intercellular communications and visualize the complex network of interactions. Other tools, such as iTALK, ICELLNET, and SingleCellSignalR, provide further functionalities and resources for analyzing ligand-receptor (L-R) interactions. Notably, Scriabin enables the examination of intercellular communication at the single-cell level, unveiling the delicate web of cellular interactions. Nonetheless, a significant drawback of current single-cell analysis methods is their inability to fully capture spatial information[14]. The lack of cellular spatial information increases the probability of false positives, causing the incorrect categorization of physically separated cells as those engaged in intercellular communication[15]. Therefore, spatial information should be incorporated into the CCC analysis to advance the field and enhance understanding of this complex process[16].

Spatial transcriptomics (ST) provides valuable spatial information regarding cells or spots that comprise multiple or partial cells[17-21]. These techniques facilitate the measurement of spatial gene expression in two-dimensional (2D) or three-dimensional (3D) tissue samples with varying degrees of cellular resolution[22]. The incorporation of ST significantly enhances the accuracy and reliability of spatial CCC inference in biology and biomedicine. Recently, several methodologies have emerged to decipher the underlying mechanisms of CCC within a spatial context. For instance, CellPhoneDBv3(CPDB3) restricts the interactions among cell types located within the same microenvironment, as determined by spatial information[23]. stLearn associates the co-expression of ligand and receptor genes with the spatial diversity of cell types[24]. SVCA[25] and MISTy[26] employ probabilistic and machine learning models, respectively, to identify spatially correlated gene interactions between cells. NCEM[27] uses a function to fit the relationship between cell types, spatial environments, and gene expression. Contrastingly, Giotto constructed spatially proximate graphs to identify the interactions between L-R pairs binding through the membrane-bound receptors[28]. Additionally, SpaOTsc performs structured optimal transport mapping between scRNA-seq and ST data, assigning spatial positions to cells and inferring the ligand-receptor signaling network mediating spatial constraints using cell–cell distances as transport costs[29]. While this method quantifies the likelihood of intercellular interactions, its specificity for image-based spatial expression data and reliance on predefined pathways somewhat limits its versatility and applicability. COMMOT[30], an extension of SpaOTsc's optimal transport framework, deduces the direction of communication by applying optimal transport analysis tools to ST data, considering the complex ligand-receptor interactions and the constraints imposed by effective intercellular communication distance. Moreover, NICHES[31] distinguishes itself as a notable addition to this toolbox, leveraging spatial transcriptomic information to understand CCCs at a single-cell resolution. This tool proves invaluable in gaining a deeper understanding of the intricate interactions and organizational patterns inherent within tissues. These methodologies hold immense promise for the direct analysis of CCC within a spatial context. Nonetheless, they encounter limitations stemming from the gene throughput and spatial resolution of ST data. Furthermore, the majority of these methods primarily focus on identifying CCC between paired cell types, neglecting the analysis of CCC between paired individual cells. Methods capable of simultaneously overcoming the limitations of ST data and inferring CCC at a single-cell resolution remain scarce, thereby restricting our comprehension of the coordinated activities exhibited by various cells in biological processes.

Here, we develop DeepTalk, an innovative method that combines cell-specific gene expression data and spatial affinities of cells to predict CCC at single-cell resolution. DeepTalk employs a graph attention network (GAT)[32] alongside a subgraph-based GAT to unveil the intricate mechanisms underlying CCCs within the spatial context of healthy and diseased tissues. This effectively overcomes the limitations posed by the restricted gene throughput and inadequate spatial resolution of ST data by integrating it with scRNA-seq data sourced from the identical region. Extensive evaluations using diverse publicly available datasets validate the exceptional performance and robustness of DeepTalk in identifying spatial CCC. Our results demonstrate that DeepTalk has great potential to discover meaningful CCC patterns across diverse conditions and provide valuable insights into the spatial intercellular dynamics within tissues.

## Results

### Overview of the DeepTalk workflow

The two pivotal tasks in analyzing the scRNA-seq and ST data are the determination of cell types and CCC, as these tasks provide insights into fine-grained tissue organization and cellular-level communications. The proposed deep learning approach, DeepTalk, consists of two primary components: (1) DeepTalk-Integration (DT-Integration), which integrates sc/snRNA-seq and ST data to identify cell types in single-cell ST data and perform deconvolution for non-single-cell ST data, and (2) DeepTalk-CCC (DT-CCC), which predicts spatially-resolved intercellular communication at a single-cell resolution for the processed ST data (Fig. 1a). Initially, GAT was employed to 'decode' the single-cell or spot-based ST data matrix. This decoding process involves utilizing self-attention mechanisms to focus on the relationships within the scRNA-seq or ST data and cross-attention mechanisms to capture the connections between scRNA-seq data and ST data. Through this decoding step, a weight matrix is generated, representing the optimal proportions of cell types for each cell or spot (Fig. 1b). The cells are labeled with the cell type exhibiting the maximum weight for the single-cell ST data, whereas different cell types with varying weights are used for the spot-based ST data as references to project the cells from the scRNA-seq data onto spatial spots. Through an iterative process, the optimal combination of cells is further refined to reconstruct the single-cell ST data based on spots (Fig. 1c).

Subsequently, DT-CCC infers CCC at a single-cell resolution by creating a cell graph that incorporates distance information and cell expression data to establish connections between cells. This construction accounts for both the positions and similarities of cells, reflecting their spatial arrangement. Individual subgraphs are generated for each cell, encoding their local characteristics through a subgraph encoder. To capture the local features of cells within their spatial context, DT-CCC utilizes a subgraph-based GAT[33], which enables it to assess the traits of each cell based on the data from adjacent cells (Fig. 1d). The model grasps cell positions and interactions by comprehending local intercellular relationships. It integrates the distinct features of all cells and deciphers the underlying connections between them using an attention-based graph neural network. This approach enables the inference of CCC and comprehension of cell interplay at a spatial level. Furthermore, to enhance the model's generalization ability, a pre-training and fine-tuning strategy is adopted[34]. Through pre-training on a large-scale dataset, the model learns the general patterns of intercellular communication and spatial relationships. Subsequent fine-tuning of specific datasets enhances the model accuracy in predicting CCC. This training approach contributes to the robustness and accuracy of the model for inferring intercellular communication. Furthermore, DT-CCC provides visualization capabilities to intuitively showcase the composition of cell types and spatial intercellular communication (Fig. 1e). When compared to other state-of-the-art methods, DT-CCC has demonstrated strong performance when applied to spot-based ST and single-cell ST data.

### Performance comparison of DeepTalk-Integration with state-of-the-art integration methods

The integration of ST and scRNA-seq data is crucial in accurately elucidating the complex CCC within a spatial context. To enhance our

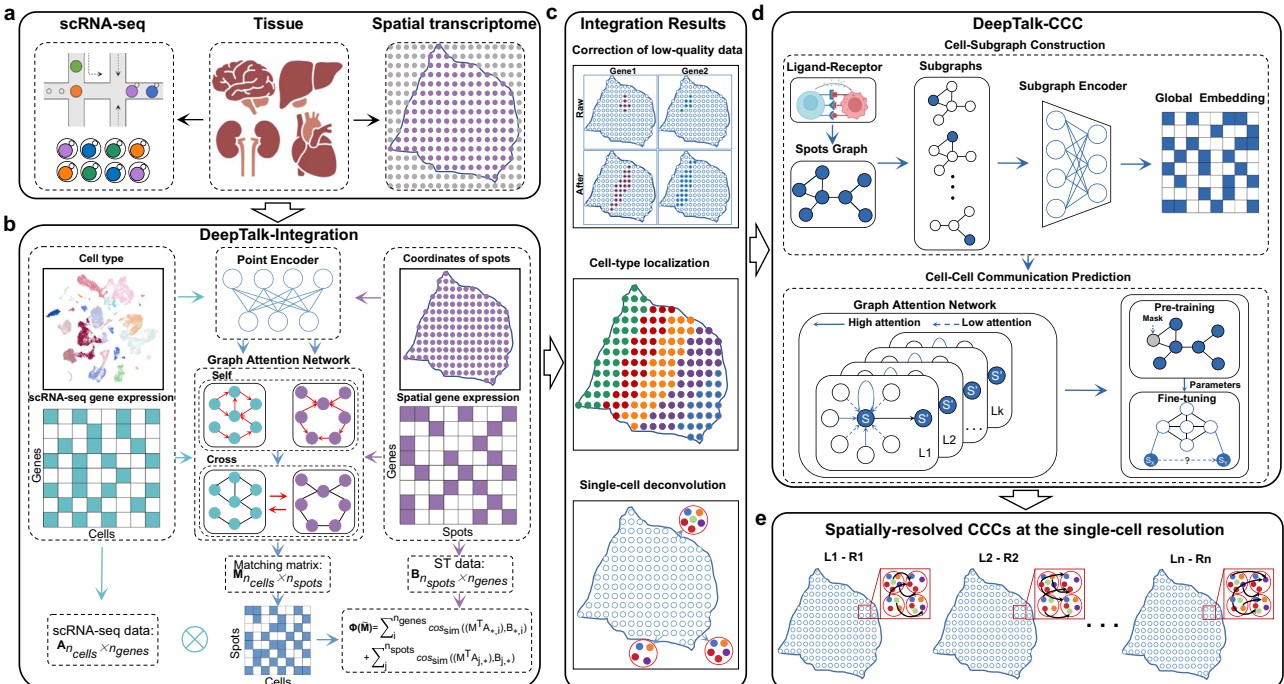

**Fig. 1 | Workflow of the DeepTalk. a** DeepTalk takes the single-cell RNA sequencing (scRNA-seq) data and spatial transcriptomics (ST) data as input. **b** DeepTalk-Integration integrates scRNA-seq and ST data using attentional graph neural networks. By employing self-attention mechanisms to capture cell relationships within scRNA-seq or ST data, and utilizing cross-attention mechanisms to explore connections between scRNA-seq and ST data, DeepTalk-Integration generate a weight matrix that represents the optimal cell type proportions for each cell or spot. **c** The integration results of DeepTalk-Integration. It's mainly contains correction of low-quality data for spatially measured genes, cell-type localization and single-cell deconvolution. **d** DeepTalk-CCC predicts the cell–cell communications using subgraph-based attentional graph neural networks. **e** DeepTalk-CCC offers visualization outputs for spatially-resolved intercellular communication at the single-cell level. The Tissue component in (**a**) was created with BioRender.com, released under a Creative Commons Attribution-NonCommercial-NoDerivs 4.0 International license.

understanding of CCC, we harness the predictive capabilities of spatial transcript distribution models for undetected transcripts and perform sophisticated cell type deconvolution analysis on tissue samples. To assess the integration efficiency of our approach, we present DT-Integration and pit it against eight cutting-edge techniques, utilizing a diverse dataset curated by Li et al.[35], which encompasses 45 paired ST and scRNA-seq datasets. The ST datasets use various techniques, including osmFISH[36], seqFISH[19], MERFISH[37], STARmap[38], ISS[39], BaristaSeq[40], ST[41], 10X Visium[42], Slide-seq[18], and HDST[43]. These datasets can be categorized into image-based and sequence-based (seq-based) methods[44]. Image-based ST methods, such as in situ hybridization and fluorescence microscopy, detect transcript spatial distributions with high resolution and accuracy; however, they may be limited in the number of detected transcripts. Seq-based ST methods capture the entire transcriptome-scale expression of RNA with spatially defined points and exhibit high technical coverage but low spatial resolution. Additionally, techniques such as Drop-seq[45], Smart-seq[46], and 10X Chromium[47] are used to generate the scRNA-seq datasets.

Initially, we undertake a rigorous evaluation of DT-Integration's performance alongside eight other integration methods: Tangram[48], gimVI[49], SpaGE[50], Seurat[51], SpaOTsc[29], novoSpaRc[52], LIGER[53], and stPlus[54]. Our assessment focuses on predicting the spatial distribution of undetected RNA transcripts within the ST datasets. To guarantee stringent and reproducible testing, we employ a ten-fold cross-validation[55] technique on the 45 paired datasets. For a systematic evaluation of the eight integration methods in predicting the spatial distribution of undetected transcripts, we utilize multiple metrics: Pearson correlation coefficient (PCC)[56], structural similarity index (SSIM)[57], root mean square error (RMSE)[58], Jensen-Shannon divergence (JSD)[59], and accuracy score (AS)[35]. These metrics offer a robust assessment of the accuracy, structural similarity, error magnitude,

divergence, and overall performance of the predictions across the 45 paired datasets (Fig. 2a, Supplementary Figs. 1 and 2). Consequently, DT-Integration proves to be the best method, surpassing other approaches in both image-based and seq-based datasets.

Second, we evaluate the performances of DT-Integration and seven other integration methods (Tangram, Cell2location[60], SpatialDWLS[61], RCTD[62], Stereoscope[63], DestVI[64], and SPOTlight[65]) for cell-type deconvolution. For the purpose of comparing their performances, we utilize the STARmap and seqFISH+ datasets as our baselines. Additionally, we simulate a "grid" that represents low spatial resolution datasets, following a precedent established in previous research[35]. In these simulated low-resolution datasets, each "spot" within the grid contains a varying number of cells, ranging from 1 to 18, much like the ST datasets that are generated using techniques such as 10X Visium or ST methods. The STARmap dataset captures 1549 cells corresponding to 15 cell types[38] (Fig. 2b). After grid transformation, the simulated dataset comprises 189 spots, each containing 1–18 cells. In identifying the positions of L4 excitatory neurons, DT-Integration demonstrated high accuracy and consistency, achieving a Pearson correlation coefficient (PCC) of 0.88, the highest among all methods. RCTD and Stereoscope closely followed with a PCC of 0.87 (Fig. 2c). For a broader analysis, we also plotted the positions of Layer 2/3(L2/3) and L5 excitatory neurons (Supplementary Fig. 3).

Additionally, we used metrics like PCC, SSIM, RMSE, and JSD to assess the accuracy of these eight integration methods in predicting cell-type composition within the simulated grid dataset (Fig. 2d). Our evaluations consistently showed that DT-Integration surpassed the other seven methods. We extended this analysis to the seqFISH+ dataset, which included 72 simulated spots[19] (Fig. 2e). Using the true positional values of L4 excitatory neurons, we found that DT-Integration achieved the highest PCC value of 0.69 (Fig. 2f). We also

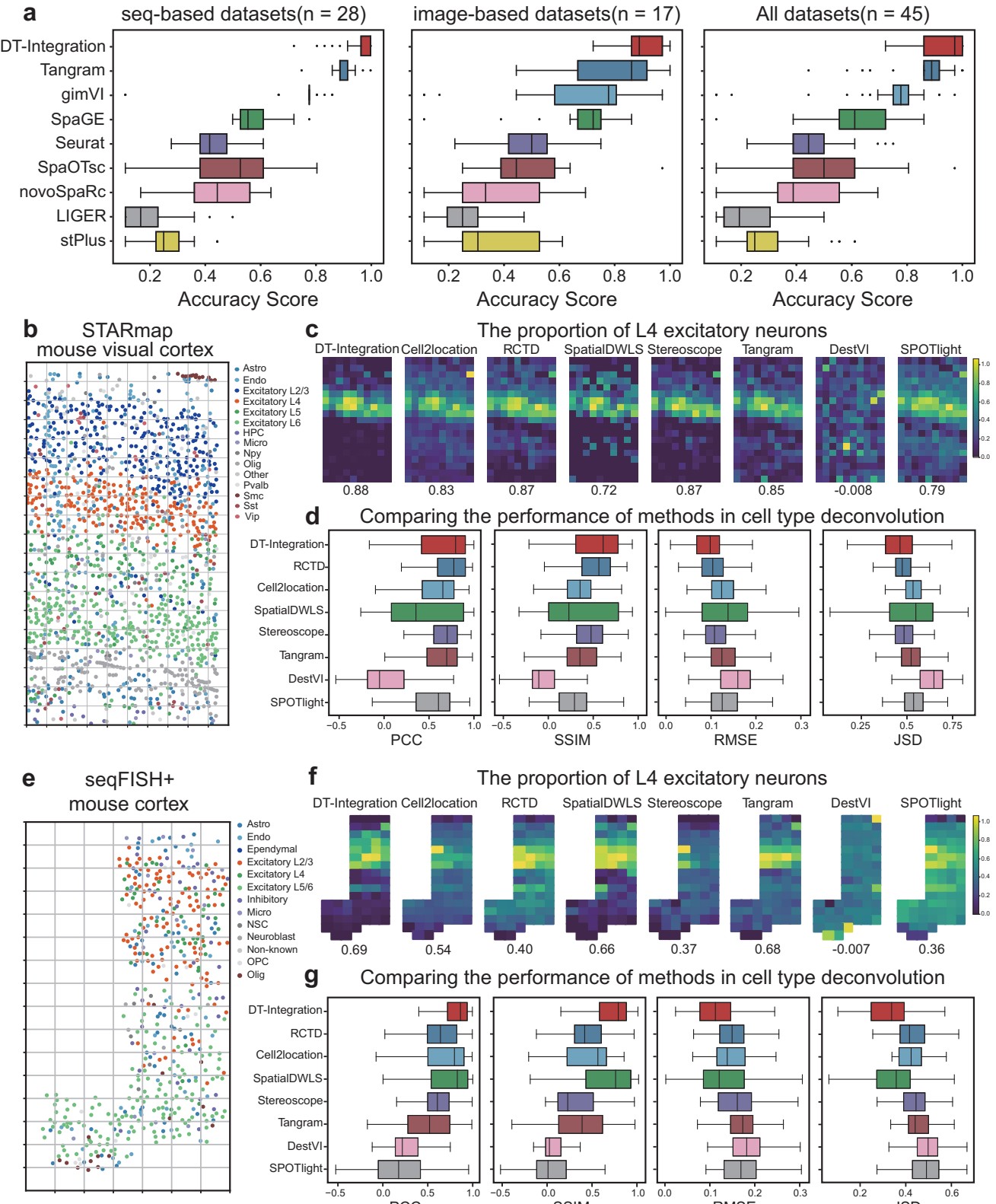

mapped the positions of L5/6 excitatory neurons for clarity (Supplementary Fig. 4). Across all cell types, DT-Integration excelled in terms of PCC, SSIM, RMSE, and JSD (Fig. 2g). To enhance the robustness of our evaluation, we incorporated 32 simulated datasets compiled by Lin et al.[35] Once again, DT-Integration proved its superiority over the other methods, consistently outperforming them across multiple metrics (Supplementary Fig. 5).

## Identification of spatially resolved CCCs in mouse visual cortex using the MERFISH dataset

DeepTalk was employed to analyze the MERFISH dataset, encompassing measurement data for 268 genes across 2399 cells originating from the VISp (visual cortex) region of the mouse[20]. To train the DeepTalk-Integration model, a total of 254 genes common to both MERFISH and snRNA-seq datasets were aligned with 11,759

**Fig. 2 | Performance comparison of DeepTalk-Integration with state-of-the-art integration methods. a** Boxplots showcasing the Accuracy Score of nine different integration methods across all 45 paired datasets, encompassing 28 sequence-based and 17 image-based datasets. The boxplots display data distribution where the box spans from the first to the third quartile, marking the median with a distinct line. The whiskers reach out to the maximum range within 1.5 times the inter-quartile range, and individual outliers are denoted by separate dots. **b** A STARmap slide from the STARmap dataset (mouse visual cortex), annotated with various cell types. Each grid portrays a simulated spot encompassing multiple cells. **c** Illustration of the proportion of Layer 4 (L4) excitatory neurons within simulated spots from the STARmap dataset, alongside predictions from eight integration methods. **d** Boxplots presenting the PCC, SSIM, RMSE, and JSD for eight integration methods, specifically analyzed for the STARmap dataset. The boxplots display data

distribution where the box spans from the first to the third quartile, marking the median with a distinct line. The whiskers reach out to the maximum range within 1.5 times the interquartile range; $n = 189$ predicted spots. **e** A seqFISH+ slide from the seqFISH+ dataset (mouse cortex), annotated with cell types. Analogous to the STARmap slide, each grid symbolizes a simulated spot with multiple cells. **f** Depiction of the proportion of L4 excitatory neurons in simulated spots from the seqFISH+ dataset, accompanied by predictions from eight integration techniques. **g** Boxplots exhibiting the metrics of PCC, SSIM, RMSE, and JSD for the eight integration methods tested on the seqFISH+ dataset. The boxplots display data distribution where the box spans from the first to the third quartile, marking the median with a distinct line. The whiskers reach out to the maximum range within 1.5 times the interquartile range; $n = 72$ predicted spots. Source data are provided as a Source Data file.

SMART-Seq2 snRNA-seq records from the VISp region[66]. This alignment served to reveal the spatial distribution of various cell types. The derived probability mappings were then fused with cell-type annotations sourced from the snRNA-seq data, resulting in spatial probability distributions for each distinct cell type (Fig. 3a). It's worth noting that glutamatergic cells demonstrated unique patterns across different cortical layers, while the majority of non-neuronal cells and GABAergic neurons showcased a granular distribution, aligning with established research findings[48]. Furthermore, a deterministic mapping was conducted by assigning the most probable cell types to their respective spatial locations, providing a visualization of the cell type distributions (Fig. 3b).

Subsequently, DT-CCC was used to predict CCC the mediated by L-R pairs for this MERFISH dataset. To assess DT-CCC's predictive prowess, ROC curves were used to represent the prediction performances of the top five L-R pairs with the most CCC under L-R pair mediation (Fig. 3c). Based on the DT-CCC prediction results, the area under curve (AUC) value of the CCC mediated by the L-R pairs was within 0.84 and 0.95, indicating that the proposed method has high accuracy and reliability in evaluating the influence of L-R pairs. Furthermore, DT-CCC was compared to other prediction methods, including statistical methods (e.g., CellCall[67], CellChat, CellPhoneDB, ICELLNET, iTALK, and SingleCellSignalR), network-based methods (e.g., Connectome and NicheNet), and ST-based methods (e.g., Giotto, CellChatv2[68], CPDB3, COMMOT and stLearn). The distance enrichment score (DES) served as a benchmark to evaluate the concordance between anticipated spatial patterns and the observed CCC[69], assessing the uniformity of spatial trends among cells. DT-CCC proved superior in CCC prediction, achieving a notable DES of 0.45, significantly outperforming other methods (Fig. 3d). By incorporating spatial data, we refined DT-CCC's predictive accuracy, based on the assumption that CCCs are more evident among adjacent cell types than distant ones. To exhibit the range of CCCs among cells at different proximities, we handpicked five cell types—L2/3 intratelencephalic neurons (IT), L4, L5 IT, L5 pyramidal tract neurons (PT), and L6 PT—known for their clustering tendencies among glutamatergic cells (Fig. 3a). We then calculated the spatial distances between these cells (Supplementary Fig. 6). Our results showed that L2/3 IT cells are near to L4, while L6 PT is the farthest from L4. Using L4 as a baseline, we computed the CCC probabilities between L4 and the other cell groups. In comparisons with 13 other methods (Fig. 3e and Supplementary Fig. 7), DT-CCC clearly excelled in differentiating between closely and distantly interacting cell types.

To further test the predictive prowess of DT-CCC at the single-cell resolution, we benchmarked it against NICHES and Scrabin, two leading techniques in inferring intercellular communications. We selected five specific cell types—L2/3 IT, L4, L5 IT, L5 PT, and L6 PT—for our analysis. By using low-dimensional embedding, we visualized their CCC patterns, with each dot on the visualization signifying a CCC event, particularly focusing on the receiver cell (Fig. 3f). This visualization technique underscores DT-CCC's proficiency in

identifying nuanced communication patterns among various cell types. Additionally, we compared the CCC strength at the single-cell resolution. Both DT-CCC and NICHES agreed that spatially proximate cells exhibit more pronounced CCCs compared to distant cells, aligning with previous hypotheses (Fig. 3g). However, Scrabin, which doesn't incorporate spatial data, tends to overlook these spatial relationships, resulting in less accurate CCC pattern evaluations (Fig. 3h). Overall, DT-CCC excelled in differentiating CCC patterns, emphasizing the significance of spatial location in shaping the cellular microenvironment.

We also selectively analyzed the glutamatergic cells at varying spatial distances and showed that the CCCs are mediated by L-R pairs (Fig. 3i, Supplementary Figs. 8 and 9). Apolipoprotein (Apoe) E[70], a critical protein involed in lipid metabolism and transport, may regulate neural function in the mouse visual cortex through its association with the metabotropic glutamate receptor 5(Grm5)[71]. This interaction potentially modulates the synaptic plasticity and stability of neural connections, which is crucial in visual signal transmission and information processing. The presence of Apoe may modify the activation pattern of Grm5, thereby affecting the transmission of glutamatergic signals and neuronal activity. By binding to Grm5, Apoe can regulate the release and concentration of glutamatergic neurotransmitters, influencing the synaptic transmission between neurons in the mouse visual cortex and modulating visual information processing and perception. Additionally, the binding of Apoe regulates the neuroinflammatory responses and cellular apoptosis, exerting protective effects against the damage and development of diseases in the mouse visual cortex. The interaction between Apoe and Grm5 significantly influences the mouse visual cortex by regulating the glutamatergic signal transmission, synaptic plasticity, and neural connection stability, thus impacting the normal functioning of visual information and neural function. The visualization of heat maps demonstrates a higher frequency of CCC occurring between neighboring cell types, revealing the importance of cell distance in facilitating CCC. In addition to demonstrating the distance-related CCC patterns, we investigated the top 50 communications between L2/3 and L4 cells mediated by Apoe-Grm5, aiming to gain deeper insights into CCCs at single-cell resolution.

## Single-cell level delineation of spatially resolved CCC in adult mouse brain with 10X Visium data

The DeepTalk model was used to analyze intercellular communication in the low-resolution ST (10X Visium) data obtained from the brain of an adult mouse[72]. The cortical clusters were investigated and analyzed, and this dataset provided, insights into the transmission of signals and intercellular interactions. Initially, the standard Scanpy functions were used to preprocess the Visium data[48,73] (as shown in Methods), identifying 324 spots and 16,562 genes (Fig. 4a). To overcome the limitations imposed by low resolution, Squidpy functions were used to assign cells to each voxel, ensuring an equal number of segmented slices per cell. This deterministic approach enabled the precise

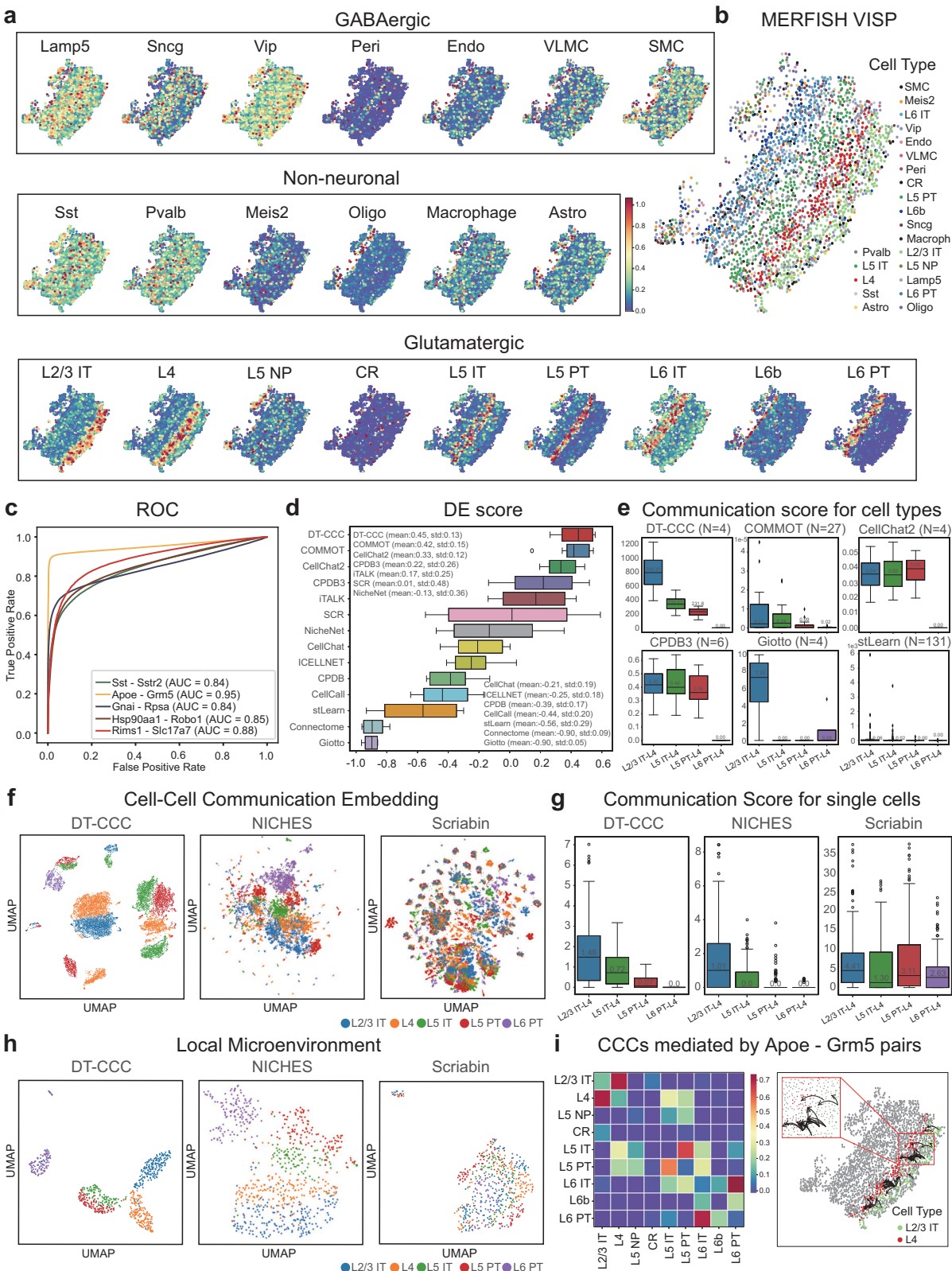

mapping of each cell within each voxel, thereby predicting cellular localization at a single-cell resolution (Fig. 4b). These predictions were instrumental in correlating cell types with their spatial positions, thereby revealing important patterns and mechanisms of intercellular communication.

To further enhance prediction accuracy, we trained a model by combining 10X Visium and scRNA-seq data, incorporating 1291 genes[74]. Subsequently, this model was applied to the scRNA-seq data obtained from the adult mouse cortical region, which comprised 21,697 cells[66]. The probability distributions for different cell types within the spatial context were obtained by merging the probability mapping results with cell-type annotations acquired from the snRNA-seq data (Fig. 4a). Based on the deterministic mapping results, the most probable cell type was assigned to each spatial location to

**Fig. 3 | DeepTalk-CCC detects spatially CCCs for mouse visual cortex region from MERFISH data. a** Probabilistic mapping of snRNA-seq data on MERFISH data. Assigns a likelihood score to each cell subset within the three main categories, indicating their potential cell type. **b** Deterministic mapping of cell types. Displays segmented cells on the MERFISH slide, color-coded based on the most probable snRNA-seq profile mapped using DeepTalk-Integration. **c** ROC Curve Evaluation of DeepTalk-CCC (DT-CCC) using the MERFISH Dataset. This curve illustrates the performance of DT-CCC in terms of true positive rate vs. false positive rate. **d** DES Rankings of CCC Tools Evaluation. A comparative analysis of various cell–cell communication tools, ranked according to their performance in the evaluation process. The boxplots display data distribution where the box spans from the first to the third quartile, marking the median with a distinct line. The whiskers reach out to the maximum range within 1.5 times the interquartile range, and individual outliers are denoted by separate dots; $n = 5$ repeated independent tests. **e** Comparison of communication scores between adjacent and distant cell types. Highlights spatial proximity-based differences in communication scores between Layer 4 (L4) cells and other cell type, including L2/3 intratelencephalic neurons (IT), L5 IT, L5 pyramidal tract neurons (PT), and L6 PT. The boxplots display data distribution where the box spans from the first to the third quartile, marking the median with a distinct line. The whiskers reach out to the maximum range within 1.5 times the interquartile range, and individual outliers are denoted by separate dots; N: number of predicted L-R(ligand–receptor) pairs. **f** Visualization of CCC via low-dimensional embedding. Each point in the graph represents a CCC event, with the cell serving as the receptor cell. **g** Comparison of the CCC score between spatially adjacent and distant cells at single-cell level. This plot compares the communication scores between spatially adjacent cells and distant cells at the single-cell level, revealing differences in communication strength based on proximity. The boxplots display data distribution where the box spans from the first to the third quartile, marking the median with a distinct line. The whiskers reach out to the maximum range within 1.5 times the interquartile range, and individual outliers are denoted by separate dots; $n = 169$ L4 cells. **h** Local microenvironment viewed via low-dimensional embedding. Each point in this diagram represents a collection of CCC events where the cell of interest serves as the receiver. **i** Predicted CCCs mediated by Apoe - Grm5 pairs. It includes the scores of CCCs mediated by Apoe - Grm5 pairs between glutamatergic cells (left) and the predicted CCCs from L2/3 IT cells to L4 cells (right). Source data are provided as a Source Data file.

visually represent the cell type distribution (Fig. 4b). This integrated approach yielded valuable insights into cell localization, cell type classification, and the overall structure of intercellular communication networks.

To investigate the details of intercellular communication mechanisms, the DT-CCC model was employed to predict the CCC mediated by various L-R pairs. This study focused on glutamatergic cells as the target cells owing to their varying spatial distances. Based on the strength of CCC with other glutamatergic cells, the interaction between the L5 IT cells and top 50 L-R pairs was primarily explored. The heatmap visualization of these interaction pairs revealed that the cells in closer proximity to L5 IT cells exhibited stronger communication (Fig. 4c). This indicates an intimate connection and communication between L5 IT cells and glutamatergic cells at specific spatial locations. To assess the predictive capabilities of the DT-CCC model, ROC curves were generated for the top 10 L-R pairs involved in CCC (Fig. 4d). Based on the DT-CCC predictions, the highest AUC value achieved for the CCC mediated by L-R pairing was 0.96. Utilizing DES as a performance indicator, we compared DT-CCC's efficacy with other prediction methods (Fig. 4e).

To understand the intricate communication patterns among cells situated at varying distances, we specifically selected five glutamatergic cell clusters—L2/3 IT, L4, L5 IT, L6 IT, and L6 corticothalamic neurons (CT)—noted for their clustering behavior (Fig. 4a). We determined the spatial relationships between these cells (Supplementary Fig. 11), revealing that L2/3 IT cells are closely adjacent to L4, whereas L6 CT cells are farthest from L4. With L4 as a reference, we calculated the communication probabilities between L4 and the other cell types. In a comparative analysis with 13 alternative methods (Fig. 4f and Supplementary Fig. 12), DT-CCC demonstrated its superiority in distinguishing between closely and distantly communicating cell types.

In evaluating DT-CCC's precision in predicting CCC at the single-cell level, we compared it to NICHES and Scrabin. We narrowed our focus to five cell types: L2/3 IT, L4, L5 IT, L6 IT, and L6 CT. Utilizing low-dimensional embedding, we represented each CCC event as a point, emphasizing the receiving cell (Fig. 4g). This approach underscores DT-CCC's ability to differentiate distinct intercellular communication patterns. Additionally, we assessed CCC strength at the single-cell level, finding that both DT-CCC and NICHES accurately predicted stronger CCC among spatially proximate cells (Fig. 4h). Scrabin, however, fell short in this evaluation. To gain deeper insights into the local cellular environment, we visualized neighboring cells using low-dimensional embedding, with each point representing a cluster of CCC events centered around a specific receiving cell (Fig. 4i). Once again, DT-CCC excelled in discerning diverse CCC patterns.

The results of CCC between glutamatergic cells mediated by the gene pairs Psen1-Pparg are also presented (Fig. 4j, Supplementary Figs. 13 and 14). Psen1 (Presenilin 1) is intricately linked to Alzheimer's disease and fulfills crucial functions in intracellular signal transduction as well as membrane protein transport[75]. Conversely, Pparg (peroxisome proliferator-activated receptor gamma) belongs to the nuclear receptor superfamily and functions as a transcription factor, overseeing diverse biological processes encompassing adipocyte differentiation, glucose metabolism, and inflammatory response[76]. Glutamate-facilitated communication among cells essentially depends on the meticulous release and reception of the neurotransmitter glutamate, a process that is meticulously modulated by a range of factors, including intracellular signaling cascades and the transcriptional activities of regulatory factors. The heatmap visualization effectively demonstrates this intricate relationship. Additionally, our analysis identified the top 50 pairs of L2/3 and L4 cells where Psen1-Pparg mediation was observed. These results shed light on the patterns of intercellular connections and the strength of communication under certain conditions. A deep exploration of these CCCs offers valuable insights into their significance in cellular and tissue functions, enhancing our understanding of the complex interactions within neural networks. By elucidating these connections, we can better comprehend the intricacies of neural communication and potentially develop more targeted therapeutic approaches for neurological disorders.

In summary, we have tentatively applied the DT-CCC model to analyze intercellular communication in low-resolution ST data, and have obtained some promising preliminary results. Through the model's predictions of cell-type localization, probability mapping, and CCCs, we have uncovered the intricate structure and organization of intercellular communication networks.

**Spatial representation of CCCs at the single-cell resolution in human pancreatic ductal adenocarcinoma using the ST dataset**
The DeepTalk algorithm was used to analyze the ST data acquired from the human pancreatic ductal adenocarcinoma (PDAC) dataset[41], which was generated using the ST technology and encompassed various tissue regions, including cancer, pancreatic, ductal, and stromal regions (Fig. 5a)[41]. To ensure data accuracy and reliability, expert histology professionals annotated these regions using the H&E staining. First, we reconstructed the single-cell ST data, which were integrated and analyzed to investigate the distribution and gene expression characteristics of various PDAC cell types (Fig. 5a). To address any uncertainty, deconvolution was performed using the matched scRNA-seq data (PDAC-A) from the same individuals (Fig. 5b). DT-CCC

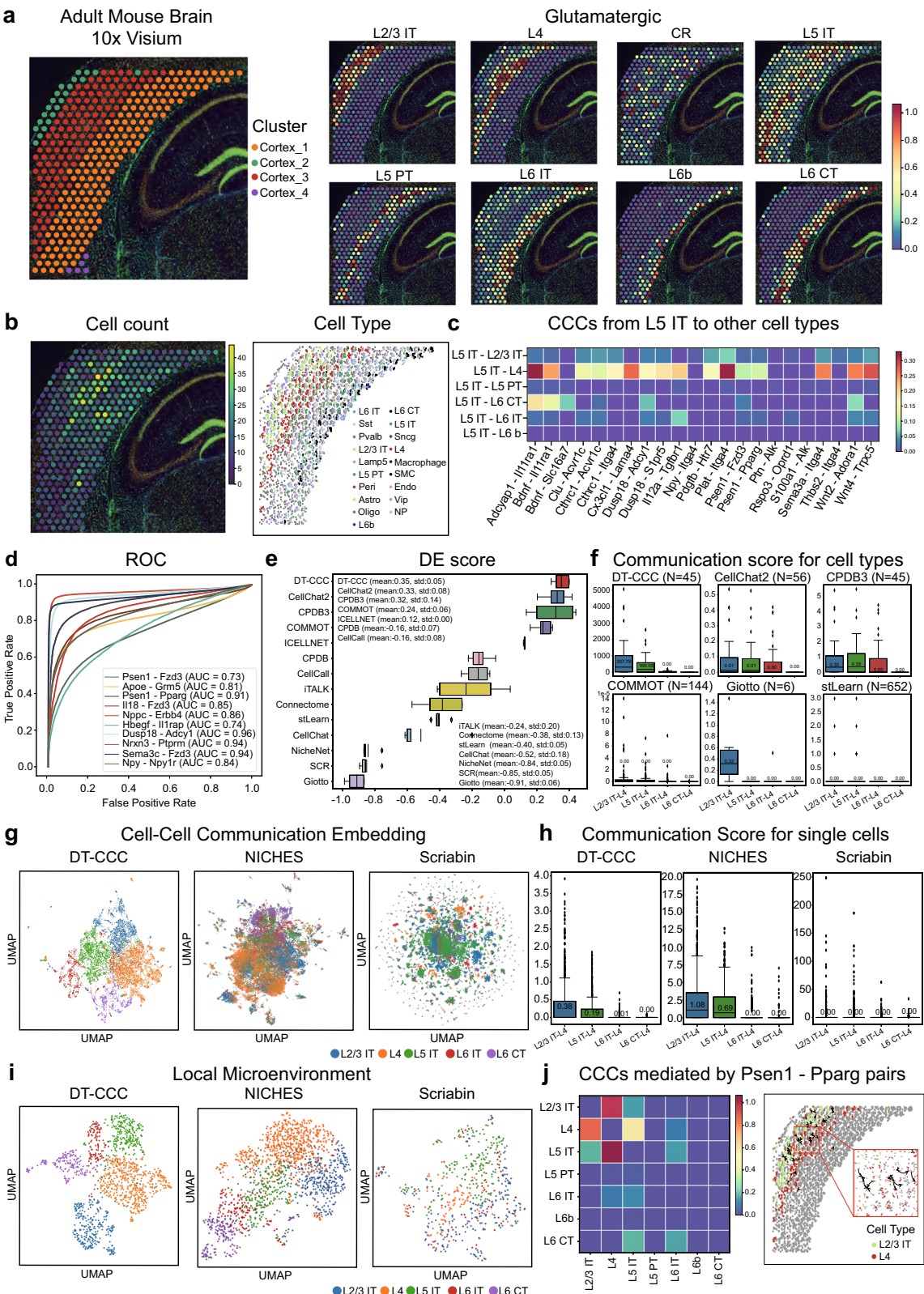

assigned different cell types to their specific tissue regions, facilitating the exploration of intricate cellular composition and spatial relationships within PDAC.

DT-CCC was employed to predict communication among cells in the PDAC dataset. Heatmaps were generated to visually represent the strength of communication between cancer clone A, cancer clone B, and neighboring cells, mediated by specific L-R pairs (Fig. 5c). Notably,

the results demonstrated a positive correlation between the predicted strength of the interaction and the spatial distance between cell types, this suggests that cells in closer proximity are more likely to engage in communication, providing a deeper understanding of the transmission of cell-to-cell information and interaction mechanisms in PDAC tissues.

To evaluate the predictive accuracy of DT-CCC, we generated ROC curves for the top 10 L-R pairs, focusing on CCC accuracy

**Fig. 4 | DeepTalk-CCC detects spatially CCCs for adult mouse brain from 10X Visium data. a** Probabilistic mapping of scRNA-seq data onto 10X Visium data. The color bar represents the probability of mapping glutamergic cells. **b** Number of cells per spot(left) and 10X Visium slide with segmented cells colored by the cell-type annotation of the most likely snRNA-seq profile mapped on that position by DeepTalk-Integration (right). **c** The score of predicted cell–cell communications (CCCs) from Layer 5 intratelencephalic neurons (L5 IT) cells to other cell types. **d** ROC Curve Evaluation of DeepTalk-CCC (DT-CCC) using the 10X Visium Dataset. This curve illustrates the performance of DT-CCC in terms of true positive rate vs. false positive rate. **e** DES Rankings of CCC Tools Evaluation. A comparative analysis of various CCC tools, ranked according to their performance in the evaluation process. The boxplots display data distribution where the box spans from the first to the third quartile, marking the median with a distinct line. The whiskers reach out to the maximum range within 1.5 times the interquartile range, and individual outliers are denoted by separate dots; $n = 5$ repeated independent tests. **f** Comparison of communication scores between adjacent and distant cell types. Highlights spatial proximity-based differences in communication scores between L4 cells and other cell type, including L2/3 IT, L5 IT, L6 IT, and L6 corticothalamic neurons (CT). The boxplots display data distribution where the box spans from the

first to the third quartile, marking the median with a distinct line. The whiskers reach out to the maximum range within 1.5 times the interquartile range, and individual outliers are denoted by separate dots; N: number of predicted L-R(ligand–receptor) pairs. **g** Visualization of CCC via low-dimensional embedding. Each point in the graph represents a CCC event, with the cell serving as the receptor cell. **h** Comparison of the CCC score between spatially adjacent and distant cells at single-cell level. This plot compares the communication scores between spatially adjacent cells and distant cells at the single-cell level, revealing differences in communication strength based on proximity. The boxplots display data distribution where the box spans from the first to the third quartile, marking the median with a distinct line. The whiskers reach out to the maximum range within 1.5 times the interquartile range, and individual outliers are denoted by separate dots; $n = 600$ L4 cells. **i** Local microenvironment viewed via low-dimensional embedding. Each point in this diagram represents a collection of CCC events where the cell of interest serves as the receiver. **j** Predicted CCCs mediated by Psen1 - Pparg pairs. The scores of CCCs mediated by Psen1 - Pparg pairs among glutamergic cells (left) and the predicted CCCs from L2/3 IT cells to L4 cells (right). Source data are provided as a Source Data file.

(Fig. 5d). Our DT-CCC predictions indicate that the AUC value for CCC mediated by L-R pairing falls between 0.82 and 0.93. Additionally, according to the DES metric, DT-CCC demonstrates superior CCC prediction accuracy compared to other methods (Fig. 5e).

To gain insights into the intricate interactions among various cell types depending on their distances, we carefully selected five cell populations for detailed analysis: Cancer clone A, Cancer clone B, Ductal terminal, Ductal antigen-presenting, and Ductal centroacinar cells (Fig. 4a). By consulting the cell locations mapped in Supplementary Fig. 16, we precisely calculated the spatial distances among these cells. Our investigation revealed a noteworthy pattern: Cancer clone B cells are in close proximity to Cancer clone A cells, whereas Ductal centroacinar cells are situated farthest apart. Adopting Cancer clone A cells as a benchmark, we explored the likelihood of interactions with other cell groups. When contrasting DT-CCC's predicted communication intensities against 13 other methods (Fig. 5f and Supplementary Fig. 17), it becomes evident that DT-CCC shines in distinguishing between cells communicating at close and far distances.

To rigorously assess the accuracy of DT-CCC's CCC predictions at the single-cell level, we compared it to NICHES and Scrabin. Focusing on our five key cell types, we employed low-dimensional embedding to visualize their CCC patterns. In this visualization, each CCC event is represented as a point, with a focus on the receiving cell (Fig. 5g). This underscores DT-CCC's remarkable ability to differentiate between various intercellular communication patterns, despite the challenges posed by similar communication patterns among cell subtypes. Additionally, we analyzed CCC strength at the single-cell level, finding that both DT-CCC and NICHES indicate more intense CCC among spatially adjacent cells (Fig. 5h). While NICHES reflects spatial information, Scrabin falls short in this aspect. To delve deeper, we used low-dimensional embedding to visualize cells surrounding a specific receiving cell, with each point representing a cluster of CCC events centered around that cell (Fig. 5i). Despite the challenges in distinguishing local microenvironments, DT-CCC outperforms other methods in reflecting the local communication environment of cells.

The interaction between the ligand EFNA5[77] and the receptor EPHA2[78] plays a crucial role in various physiological and pathological processes in the human body, particularly in regulating intercellular communication. EPHA2, a receptor tyrosine kinase, engages in bidirectional signaling with various Ephrin-A ligands, including EFNA5, thereby influencing cellular processes such as migration, adhesion, proliferation, and differentiation. In the context of PDAC, these cellular behaviors are particularly significant as they are intimately linked to tumor growth, invasion, and metastasis. As a ligand for EPHA2, EFNA5 interacts with the receptor to induce bidirectional signaling,

modulating the adhesion, organization, and development of neurons, vascular systems, and epithelial cells. In PDAC, this interaction can affect the adhesive properties and migratory abilities of tumor cells, altering the tumor's growth pattern and metastatic potential. The heatmaps presented here illustrate the CCC potential mediated by EFNA5-EPHA2 in PDAC tissues. These heatmaps offer invaluable insights for deeper investigations into related signaling pathways and cellular functions (Fig. 5j). Additionally, we showcase the top 50 CCCs from cancer clone A cells to ductal end cells at a single-cell resolution, mediated by EFNA5-EPHA2. These findings unveil the various interaction patterns and strengths of communication between specific cells under different conditions (Supplementary Figs. 18 and 19). This detailed information provides a more holistic understanding of dynamic CCCs and their crucial roles in functional regulation at both the cellular and tissue levels.

## Discussion

DeepTalk is a powerful approach devised to elucidate the mechanisms and functions of CCC. It utilizes an attention mechanism-based GNN to accurately predict the L−R pairs that mediate intercellular communication and visualize CCC at multiple scales. The validation experiments have demonstrated the remarkable capability of DeepTalk to identify and visualize the spatial communication mediated by the significantly enriched intercellular L−R pairs. To substantiate the efficacy and versatility of DeepTalk, diverse representative experimental datasets were utilized, such as the single-cell ST dataset obtained from MERFISH and the spot-based ST dataset acquired using ST and 10x Visium. These datasets include the ST information pertaining to various technical platforms and experimental conditions, encompassing intricate and diverse CCC. They confirm the applicability of DeepTalk to different data types, further validating its generalizability and reliability.

DeepTalk emerges as an ingenious approach applicable to both single-cell and spot-based ST datasets. This is achieved by amalgamating scRNA-seq and ST datasets, elevating CCC analysis. For single-cell ST datasets, it employs a similarity-driven classification approach. Rather than relying on expression patterns, it meticulously categorizes and analyzes these datasets by pinpointing the most similar and top-ranked cell clusters. Its attention mechanism-driven GNN uncovers pertinent relationships and intercellular correlation patterns, bolstering classification accuracy and reliability. When dealing with spot-based ST datasets, DeepTalk introduces a cutting-edge data reconstruction technique. By carefully selecting and mapping optimal cell combinations, it recreates ST maps at a single-cell level, encompassing details about known cell types. This tactic reveals intercellular

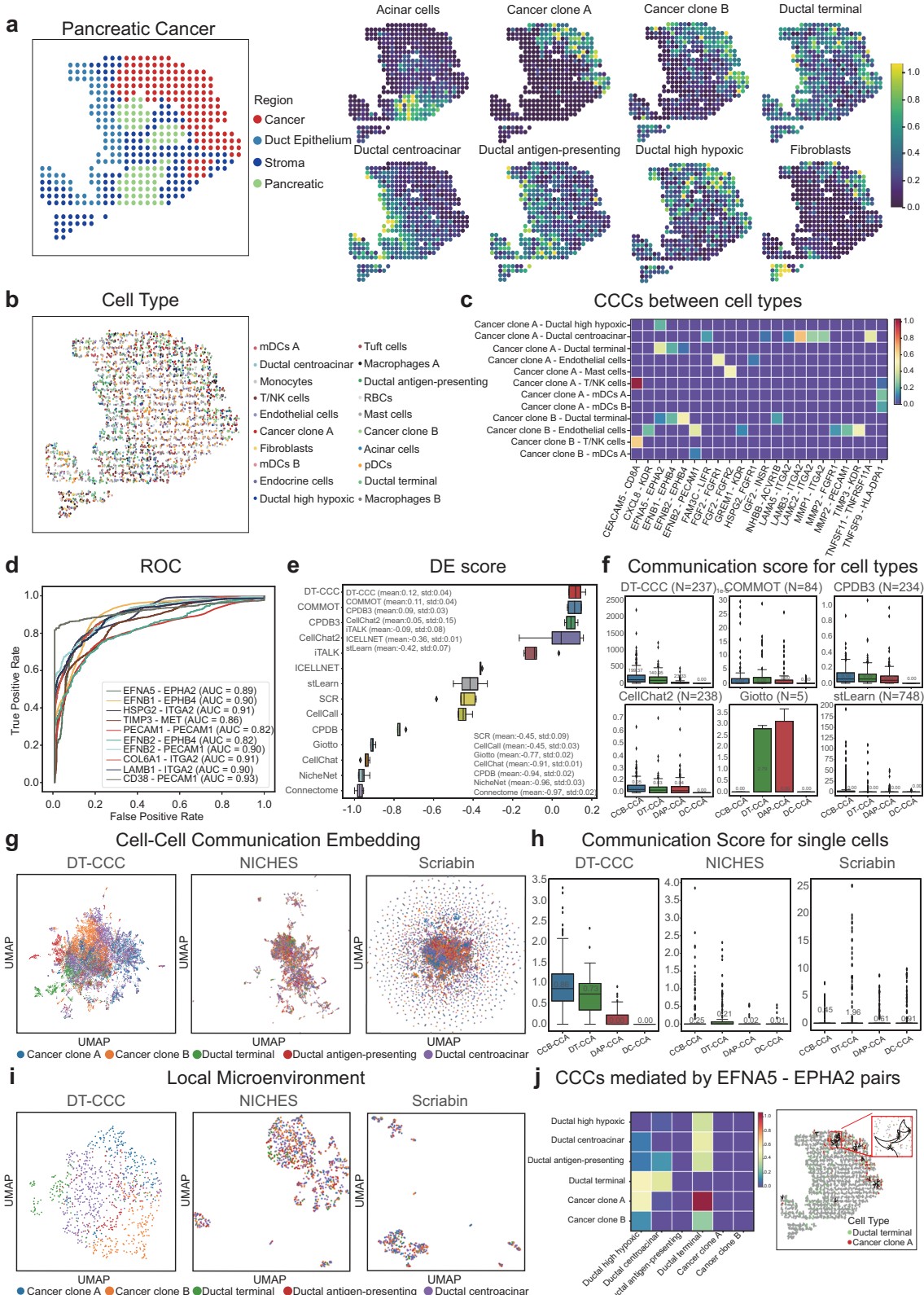

connections and spatial distribution patterns, effectively transforming spot-based ST data into single-cell resolution ST data. This not only enhances data interpretability but also boosts information accuracy. The integration of scRNA-seq and ST datasets broadens our comprehension of CCC across various dimensions and levels. scRNA-seq data offers intricate insights into intracellular gene expression, while ST data exposes intercellular interactions and correlations. By amalgamating these data types, DeepTalk presents a holistic and multifaceted perspective, fostering a deeper understanding of CCC mechanisms.

DeepTalk explores the communication preferences between different cell types and identifies the trends in these preferences in various ST datasets. DeepTalk analysis can be used to accurately characterize the spatial distribution of each cell type at a reconstructed

**Fig. 5 | DeepTalk-CCC detects spatially CCCs for Human pancreatic ductal adenocarcinoma from ST data. a** Annotated spatial regions of PDAC data (left) juxtaposed with the probabilistic mapping of scRNA-seq data onto ST data (right). **b** ST slide showcasing segmented cells, colored according to the most probable scRNA-seq profile mapped by DeepTalk-Integration. **c** Scores indicating predicted cell–cell communications (CCCs) between cancer clone A, cancer clone B, and other cell types. **d** ROC curve evaluation of DeepTalk-CCC (DT-CCC) using the ST dataset. This curve illustrates the performance of DT-CCC in terms of true positive rate vs. false positive rate. **e** DES rankings of CCC tools evaluation. A comparative assessment of various CCC tools, ranked based on their evaluation performance. The boxplots display data distribution where the box spans from the first to the third quartile, marking the median with a distinct line. The whiskers reach out to the maximum range within 1.5 times the interquartile range, and individual outliers are denoted by separate dots; $n = 5$ repeated independent tests. **f** Comparison of communication scores between adjacent and distant cell types. Highlights spatial proximity-based differences in communication scores between Cancer clone A (CCA) cells and other cell types, including Cancer clone B (CCB), Ductal terminal (DT), Ductal antigen-presenting (DAP), and Ductal centroacinar (DC). The boxplots

display data distribution where the box spans from the first to the third quartile, marking the median with a distinct line. The whiskers reach out to the maximum range within 1.5 times the interquartile range, and individual outliers are denoted by separate dots; N: number of predicted L-R(ligand–receptor) pairs. **g** Visualization of cell–cell communication. Represents CCC events as points in a low-dimensional embedding, with the receiver cell indicated. **h** Single-cell level comparison of communication scores. Compares communication scores between adjacent and distant cells, revealing proximity-based differences in strength. **i** Local microenvironment visualization. Represents collections of CCC events where the cell of interest serves as the receiver in a low-dimensional embedding. The boxplots display data distribution where the box spans from the first to the third quartile, marking the median with a distinct line. The whiskers reach out to the maximum range within 1.5 times the interquartile range, and individual outliers are denoted by separate dots; $n = 127$ cancer clone A cells. **j** Predicted CCCs mediated by EFNA5-EPHA2 pairs. Displays scores of CCCs mediated by EFNA5-EPHA2 pairs between ductal cells and cancer cells and predicted CCCs from Cancer clone A cells to Ductal terminal cells. Source data are provided as a Source Data file.

single-cell resolution, furnishing information on the proximity relationships between different cell types. Utilizing a subgraph-based GNN, DeepTalk examines cellular interactions at multiple levels, fostering a multifaceted analytical approach that aids in constructing intercellular communication networks and unveils intricate interconnection patterns. By investigating these networks, the communication patterns and relationship features between different cell types can be determined with improved accuracy. Furthermore, DeepTalk facilitates the statistical analysis and visualization of LRIs in proximity, expanding the understanding of the CCC mechanisms and patterns by providing a visual representation of the dynamic communication networks between cells. Furthermore, LRI analysis can identify the specific L–R pairs that mediate intercellular communication, further enriching our understanding of CCC.

Indeed, analyzing and visualizing spatial CCC at single-cell resolution poses challenges for scRNA-seq data. Generally, CCC is interpreted as L–R pairs between different cell types. However, DeepTalk overcomes these challenges by incorporating spatial information, allowing the selection of adjacent co-expressing cell pairs in space. This approach provides information-rich methods for analyzing and visualizing LRIs and their mediated CCC from different perspectives, including in the context of disease pathophysiology. The excellent performance of DeepTalk on the benchmark ST dataset demonstrates its ability to decipher the CCC mechanisms in both healthy and diseased tissues. It offers a nuanced and precise interpretation of CCC by disclosing intercellular LRIs and their functions in CCC. Furthermore, it can predict and visualize the intercellular communication at single-cell resolution, and analyzes the associated L–R pairs, which is crucial for understanding the physiological and pathological processes and developing therapeutic approaches. By amalgamating scRNA-seq and ST datasets, DeepTalk elevates our anatomical understanding of CCC, offering more granular insights; it also presents a multifaceted and integrated approach to comprehending CCC through the exploration and visualization of intercellular LRIs.

In essence, DeepTalk offers a significant method that utilizes GNNs and attention mechanisms to predict and illustrate CCC mediated by L–R pairs. By combining scRNA-seq and ST datasets, it enhances our analytical capabilities in exploring CCC, ultimately providing detailed and accurate understanding of cellular interactions. In spite of challenges like long-distance communication and multimodal data amalgamation, DeepTalk stands as a crucial tool for unraveling the intricacies of CCC by probing the cell-to-cell dialog and visually presenting these interactions. Its applications extend to studying physiological and pathological processes, as well as aiding in the development of innovative therapeutic approaches. Additionally, integrating other omics data, particularly those generated by cutting-

edge technologies like 10x Multiome and digital spatial profiling, can further refine our examination of spatial control in CCC. By amalgamating multimodal data, we can achieve a deeper comprehension of the mechanisms that drive CCCs. To advance CCC research, DeepTalk can evolve to encompass spatial constraints and multimodal data integration, thereby enabling more precise and detailed inference analyses.

## Methods
### Preprocessing of datasets
Several preprocessing steps were performed for each dataset. Quality control of the scRNA-seq data was performed using Scanpy. Cells were filtered based on several criteria to ensure the quality of the downstream analysis. Specifically, cells with a mitochondrial gene expression percentage exceeding 20% were excluded, as a high mitochondrial content can indicate stressed or apoptotic cells. However, it's important to note that the mitochondrial gene expression percentage threshold can be adjusted based on specific circumstances. Cells with an abnormally high total count or an excessively large number of expressed genes were also removed. The specific thresholds for total counts and the number of genes were carefully chosen based on the characteristics of each dataset to retain only the highest quality cells for further analysis. Additionally, Scanpy's filter_cells and filter_genes functions were utilized to further purify the dataset by removing low-quality cells and genes. For both scRNA-seq and ST datasets, normalization of the expression matrix is essential. We employed the widely-used normalize_total function from the Scanpy package to perform this normalization. The normalization can be mathematically represented as:

$$X'_{ij} = \frac{\mathbf{X}_{ij}}{\sum_j \mathbf{X}_{ij}} * T, \tag{1}$$

where $\mathbf{X}_{ij}$ is the raw count of gene $i$ in cell $j$, $\mathbf{X}_{ij}$` is the normalized expression value, and $T$ is the target sum of counts after normalization, ensuring consistency across all cells.

During the integration of scRNA-seq and ST data, the selection of training genes is paramount. To enhance the discriminative power of ST data, we carefully curate a set of marker genes that are uniformly distributed across different cell types in the scRNA-seq data as training genes. Specifically, we employ the rank_genes_groups functionality in Scanpy to identify the most highly expressed genes within specific cell types. Following this, we create a non-redundant gene set and intersect it with the gene list in the ST dataset. Genes that record a zero count in either dataset are excluded from our training gene set, thus guaranteeing that only relevant and expressed genes are included. This

procedure results in a refined gene set comprising 'n' genes, which will serve as the foundation for model training. For model validation, we adopt a leave-one-out validation strategy. Specifically, we sequentially designate each gene in the refined gene set as the test gene, while the remaining 'n-1' genes function as training genes. This training process is iterated 'n' times, omitting a different gene each time, to ensure that each gene receives an independent prediction evaluation.

## DeepTalk algorithm

The DeepTalk model has two components: integration of the scRNA-seq and ST datasets (DeepTalk-Integration) and prediction of the spatial CCCs (DeepTalk-CCC). The former focuses on determining the cell-type composition of the single-cell or spot-based ST datasets, whereas the latter is designed to predict the spatially mediated CCCs influenced by the L–R pairs in the spatial context.

## Integration of the scRNA-seq and ST datasets

An attention-based GNN was used to integrate the scRNA-seq and ST datasets (Supplementary Fig. 20a). This network was tasked with generating matching descriptors, denoted as $\mathbf{f}_i \in \mathbb{R}^D$, where '$D$' represents the dimensionality of the feature space, determining the length or number of components in the feature vector. These descriptors were created through feature communication among the initial features, which encompassed raw or minimally processed data such as transcriptomic information and cell positions. Initially, we developed a point encoder combining the transcriptomic data and cell positions for each cell, denoted as $i$. By employing a multilayer perceptron (MLP), the cell positions were embedded into a high-dimensional vector, represented as:

$$\mathbf{x}_i^{(0)} = \mathbf{d}_i + \mathrm{MLP}_{\mathrm{enc}}(\mathbf{p}_i), \tag{2}$$

where $\mathbf{d}_i$ refers to the gene expression data obtained from the transcriptome and $\mathbf{p}_i$ represents the positional information of the cells. This point encoder allows the GNN to simultaneously leverage the $\mathbf{d}_i$ and $\mathbf{p}_i$ during subsequent inference stages.

Subsequently, we created a graph integrating two omics with nodes representing the cells from both transcriptomics. Self-edges connected each cell $i$ to all other cells within the same omics, whereas cross-edges linked cell $i$ to all other cells with different omics. To propagate information effectively, message-passing equations were employed, enabling the diffusion of information along the self- and cross-edges. This approach resulted in a multi-GNN where each node started with a high-dimensional state. At each layer, the updated representations were computed by simultaneously aggregating messages from all edges of all the nodes within the graph.

In the proposed framework, the intermediate representation of element $i$ in the scRNA-seq A at layer $l$ is denoted by $\mathbf{x}_{Ai}^{(l)}$. The message $\mathbf{m}_{\varepsilon \to i}$ represents the aggregation of information from all cells $j$, such that $\{j : (i,j) \in \varepsilon\}$, where $\varepsilon \in \{\varepsilon_{self}, \varepsilon_{cross}\}$. The residual message-passing update for all $i$ in the scRNA-seq $A$ can be expressed as follows:

$$\mathbf{x}_{Ai}^{(l+1)} = \mathbf{x}_{Ai}^{(l)} + \mathrm{MLP}\left(\left[\mathbf{x}_{Ai}^{(l)} || \mathbf{m}_{\varepsilon \to i}\right]\right) \tag{3}$$

where the concatenation operator $[\cdot || \cdot]$ is used for concatenation. A comparable update can be simultaneously applied to all the points in omics B. To create a hierarchical structure, a predetermined number of layers L with distinct parameters are linked together and the information is alternately aggregated along the self- and cross-edges; starting from $l = 1$, $\varepsilon = \varepsilon_{self}$ when $l$ is an odd number and $\varepsilon = \varepsilon_{cross}$ when $l$ is an even number. This approach enabled iterative representation aggregation along different edges, thereby facilitating effective representation learning.

The aggregation and computation of the message $\mathbf{m}_{\varepsilon \to i}$ were performed by an attention mechanism. The self- and cross-edges used self- and cross-attention, respectively. As in database retrieval, the query $\mathbf{q}_i$ was used to retrieve values $\mathbf{v}_j$ for specific elements based on their attributes, represented as keys $\mathbf{k}_j$. Thereafter, the message was computed by taking the weighted average of the retrieved values as follows:

$$\mathbf{m}_{\varepsilon \to i} = \sum_{j:(i,j)\in\varepsilon} a_{ij}\mathbf{v}_j, \tag{4}$$

The attention weight $a_{ij}$ is calculated as the softmax of the key-query similarities and is represented as $a_{ij} = \mathrm{Softmax}(\mathbf{q}_i^T\mathbf{k}_j)$. The key, query, and value are obtained by applying linear projections to the deep features of the GNN. Considering that the query point $i$ belongs to the scRNA-seq dataset $Q$ and all source points reside in the ST dataset $S$, this relationship can be expressed as $(Q,S) \in \{A,B\}^2$.

$$\mathbf{q}_i = W_1\mathbf{x}_{Qi}^{(l)} + \mathbf{b}_1 \tag{5}$$

$$\begin{matrix} \mathbf{k}_j \\ \mathbf{v}_j \end{matrix} = \begin{matrix} W_2 \\ W_3 \end{matrix} \mathbf{x}_{Sj}^{(l)} + \begin{matrix} \mathbf{b}_2 \\ \mathbf{b}_3 \end{matrix}, \tag{6}$$

The projection parameters are specific to each layer $l$; these parameters are learned and shared across all points in both datasets. Multi-head attention was employed to enhance the expressiveness of the model, enabling the representation of both geometric transformations and assignments. The resulting matching descriptors were obtained via linear projections as follows:

$$\mathbf{f}_i^A = W \cdot \mathbf{x}_{Ai}^{(l)} + \mathbf{b}, \forall_l \in A, \tag{7}$$

They were similarly obtained for points in $B$.

Creating individual representations for every potential match in a matrix of size $n_{cells} \times n_{spots}$ is impractical. Therefore, an alternative approach was adopted by representing the pairwise scores as a similarity matrix $M \in \mathbb{R}^{ncells \times nspots}$ capturing the similarity of the matching descriptors.

$$M_{i,j} = \langle \mathbf{f}_i^A, \mathbf{f}_j^B \rangle, \forall (i,j) \in A \times B, \tag{8}$$

where $\langle \cdot, \cdot \rangle$ is the inner product. The matching descriptors are not normalized, and their magnitude may vary on a per-feature basis throughout the training process to reflect the confidence level of the predictions. To derive the mapping matrix, the following objective function was minimized with respect to $M$:

$$\phi\left(\widetilde{M}\right) = \sum_k^{n_{genes}} \cos_{\mathrm{sim}}((\mathbf{M}^T\mathbf{A}_{*,k}),\mathbf{B}_{*,k}) + \sum_k^{n_{spots}} \cos_{\mathrm{sim}}((\mathbf{M}^T\mathbf{A}_{j,*}),\mathbf{B}_{j,*}), \tag{9}$$

where $\cos_{\mathrm{sim}}$ is the cosine similarity function and * indicates matrix slicing.

## Spatial CCC prediction

During pretraining, we employed a random-walk strategy to generate pretraining subgraphs $G_C$ for each node in the graph. This process involved masking and predicting nodes within the random walks, effectively capturing the graph's overall connectivity patterns (Supplementary Fig. 20b). For each node v in the graph during pretraining (and for each node-pair during fine-tuning), we generated individual subgraphs denoted as $g_c \in G_c$. Each subgraph $g_c$ was encoded as a set of nodes represented by $g_c = (v_1, v_2, \ldots, v_{|V_c|})$, where $|V_c|$ denotes the number of nodes in $g_c$.

We assigned a low-dimensional vector representation to each node $\mathbf{v}_i$ in the subgraph. This representation was obtained by mapping the attributes (such as gene expression data or cellular phenotypes) and the structure-based embedding of $\mathbf{v}_i$ using the function $f_{\mathrm{attr}}(.)$. The

resulting stacked vector was denoted as $\mathbf{h}_i = W_e f_{\text{attr}}(\mathbf{v}_i)$, where $W_e$ is a learnable embedding matrix. Collectively, the node embeddings within the subgraph $g_c$ were represented as $H_c = (\mathbf{h}_1, \mathbf{h}_2, \ldots, \mathbf{h}_{|V_c|})$. This flexible representation approach allowed us to incorporate both node and relation attributes into low-dimensional embeddings. Alternatively, these embeddings could also be initialized using the output embeddings from other global feature generation methods that capture the multi-relational graph structure. Specifically, for the initialization of node features in our study, we merged pretrained node representation vectors from node2vec with gene expression data acquired from cells.

For a subgraph $g_c$, where $V_c$ represents the set of nodes and $H_c \in \mathbb{R}^{d \times |V_c|}$ denotes their corresponding global input embeddings, the main objective of contextual learning is to transform these global embeddings to reflect the most representative roles of the nodes within the structure of $g_c$. This transformation was achieved via a series of layers, with the model having the flexibility to incorporate multiple layers to account for higher-order relationships within the graph. To capture higher-order relational dependencies between nodes—including indirect and multi-step interactions—we introduced a semantic association matrix, denoted as $\bar{A}$, which acts as an asymmetric weight matrix. This asymmetry originated from the different influences that the two cells may have on each other within a subgraph. The weights of the matrix $\bar{A}^k$ were iteratively learned in each translation layer $k$ by considering the connectivity between nodes through the local context subgraph $g_c$ and larger global graph $G$.

In the $k+1$ translation layer, the semantic association matrix $\bar{A}^k \in \mathbb{R}^{|V_c| \times |V_c|}$ was updated via the transformation operation, which involves performing message passing across all nodes within the subgraph $g_c$ and updating the node embeddings $H_c^k = (\mathbf{h}_1^k, \mathbf{h}_2^k, \ldots, \mathbf{h}_{|V_c|}^k)$ to $H_c^{k+1}$. The update process ensures that the embeddings capture the evolving representations of the nodes based on the contextual information derived from the message passing and relationship updates in the subgraph. Specifically, the updated node embeddings $H_c^{k+1}$ are computed as follows:

$$H_c^{K+1} = f_{\text{NN}}(W_s H_c^k \bar{A}^k + H_c^K), \tag{10}$$

Here, $f_{\text{NN}}$ represents a non-linear function, and $W_s \in \mathbb{R}^{d \times d}$ is a learnable transformation matrix.

The non-linear function and the transformation matrix were used to compute the corresponding entry $\bar{A}_{ij}^k$ in the semantic association matrix. To retain contextual embeddings from the previous step, we incorporate a residual connection. This ensures that global relations are preserved by passing the original global embeddings through the layers. For the two nodes $\mathbf{v}_i$ and $\mathbf{v}_j$ within the subgraph $g_c$, the calculation of $\bar{A}_{ij}^k$ utilizes a multihead attention mechanism with $N_h$ heads, allowing us to capture the relational dependencies within different subspaces. For each head, $\bar{A}_{ij}^k$ was computed as follows:

$$A_{i,j}^{-k} = \frac{\exp\left(\left(W_1 h_i^k\right)\left(W_2 h_j^k\right)\right)}{\sum_{t=1}^{|V_c|} \exp\left(\left(W_1 h_i^k\right)^T \left(W_2 h_t^k\right)\right)}, \tag{11}$$

where the transformation matrices $W_1$ and $W_2$ are learnable parameters. By applying multiple translation layers, multiple embeddings were generated for each node within the subgraph. Considering the various embeddings in downstream tasks, the node embeddings learned from different layers $\{h_i^k\}_{k=1,\ldots,K}$ were embedded into the contextual embedding $\tilde{h}_i$ for each node. This aggregation was performed as follows:

$$\tilde{\mathbf{h}}_i = \mathbf{h}_i^1 \oplus \mathbf{h}_i^2 \oplus \cdots \oplus \mathbf{h}_i^K, \tag{12}$$

After obtaining the embedding vectors $\{\tilde{\mathbf{h}}_i\}_{i=1,2,\ldots,|V_c|}$ for the nodes within $g_c$, these embeddings can be used as inputs for prediction tasks. During pretraining, a linear projection function was applied to the embeddings to predict the probabilities of the masked nodes. In the fine-tuning step, we utilized a single-layer feed-forward network with a softmax activation function for binary link prediction, facilitating predictions regarding the presence or absence of links between nodes.

Pretraining in the proposed model involves training a self-supervised node-prediction task. For each node in $G$, a node $g_c$ with a diameter (the maximum shortest distance between any pair of nodes) was created using the aforementioned generation methods. Subsequently, a single node within the subgraph was randomly masked for prediction without altering the graph structure. Therefore, pretraining was accomplished by maximizing the probability of correctly predicting the masked node $\mathbf{v}_m$ based on the given context $g_c$. The probability was computed in the following form:

$$\theta = \text{argmax}_\theta \Pi_{g_c \in G_C} \Pi_{\mathbf{v}_m \in g_c} p(\mathbf{v}_m | g_c, \theta), \tag{13}$$

where $\theta$ represents the set of model parameters.

To fine-tune the model further, we focused on a contextualized link prediction task. Multiple fine-grained contexts were generated for each node pair considered for link prediction. During this stage, the model was trained to maximize the probability of observing a positive edge ($e_p$) given its corresponding context ($g_{cp}$). Simultaneously, the model learned to assign low probabilities to the negatively sampled edges ($e_n$) and their associated contexts ($g_{cn}$). The overall objective was constructed by summing over two subsets of training data: positive edges ($D_p$) and negative edges ($D_n$). By optimizing this objective, the model improved its ability to accurately predict the positive and negative edges.

$$L = \boldsymbol{\Sigma}_{(e_p, g_{cp}) \in D_p} \log\left(\text{P}\left(e_p | g_{cp}, \theta\right)\right) + \boldsymbol{\Sigma}_{(e_n, g_{cn}) \in D_n} \log\left(1 - \text{P}(e_n | g_{cn}, \theta)\right), \tag{14}$$

The probability of an edge between two nodes, denoted by $e = (\mathbf{v}_i, \mathbf{v}_j)$, was calculated using the similarity score $\text{S}(\mathbf{v}_i, \mathbf{v}_j)$, which can be mathematically expressed as $\text{S}(\mathbf{v}_i, \mathbf{v}_j) = \sigma(\tilde{\mathbf{h}}_i^T \cdot \tilde{\mathbf{h}}_j)$, where $\tilde{\mathbf{h}}_i$ and $\tilde{\mathbf{h}}_j$ are embeddings of $\mathbf{v}_i$ and $\mathbf{v}_j$, respectively. $\sigma(\cdot)$ represents sigmoid function. The probability of an edge between two nodes was thereby calculated based on the similarity of their embeddings.

## Definition of cell type for the ST dataset

To analyze the single-cell ST dataset, the cell type with the highest coefficient was assigned to each individual cell type. For the 10X Visium fluorescence dataset used in this study, squidpy.im.segment () was used to segment the tissue image. For the ST dataset, the maximum cell number ($N_{cell}$) was defined for each spot, and $N_{cell}$ was set to 20 based on a recent review[79]. To determine the optimal combination ($\boldsymbol{\omega}$) of cells for each spot, the following function was used:

$$\boldsymbol{\omega}_i(i \in 1,2,\ldots,k) = \begin{cases} [N_{cell}\beta_i] + 1 & (\{N_{cell}\beta_i\} \geq 0.5 \\ [N_{cell}\beta_i] & \{N_{cell}\beta_i\} < 0.5 \end{cases}, \tag{15}$$

where $\left[N_{\text{cell}}\beta_i\right]$ and $\{N_{\text{cell}}\beta_i\}$ represent the integer and fractional parts of $N_{cell}\beta_i$, respectively. Thereafter, a subset of cells ($n = \sum_{i=1}^k \boldsymbol{\omega}_i$) was randomly selected from the total cell population ($S$) and the merged expression profile ($\boldsymbol{\epsilon}$) of the cell was compared with the ground truth using the following function:

$$\text{arg min}\{n < N_{\text{cell}}\}\sum_{i=1}^n \left(Y_i - \sum_{j=1}^m \boldsymbol{\epsilon}_i^j\right)^2, \tag{16}$$

To assign the coordinates ($x', y'$) to each sampled cell, we introduced a probabilistic distribution based on the ratio ($R$) of the same

cell type in neighboring spots, allowing us to determine the probability of locating a cell in a specific region within a spot. The distribution was calculated using the following equation:

$$x' = x_0 + \frac{a d_{\min} \cos\left(\frac{\theta\pi}{180}\right)}{2}, \quad (17)$$

$$y' = y_0 + a d_{\min} \sin\left(\frac{\theta\pi}{180}\right)/2, \quad (18)$$

where $d_{\min}$ represents the minimum spatial distance to the closest neighbor spot and $\alpha \in (0,1]$ and $\theta \in (0,360]$ represent the weight for dmin and angle toward the spot center $(x_0, y_0)$, respectively. Notably, $\theta$ is determined by the following probability equation:

$$\widetilde{P}(\theta) = \frac{R_q + 1}{\sum_i^Q (R_i + 1)}, \theta \in (90q - 90, 90q], \quad (19)$$

where $q$ is the qth neighbor spot among $Q$ spots. Practically, $Q$ was set to 4, dividing the space around the spot into four quadrants and filtering the nearest neighbor in each quadrant. After determining $\theta$, the corresponding neighbor spot $(x', y')$ was selected to calculate the probabilistic distribution of $\alpha$ using the following equation:

$$\widehat{P}(\alpha) = \begin{cases} \frac{R_{x_0,y_0} + 1}{R_{x_0,y_0} + R_{x',y'} + 2}, & \alpha \in (0, 0.5] \\ \frac{R_{x_0,y_0} + 1}{R_{x_0,y_0} + R_{x',y'} + 2}, & \alpha \in (0.5, 1] \end{cases}, \quad (20)$$

where $R_{x_0,y_0}$ and $R_{x',y'}$ represent the ratios of the given cell type in each spot to its neighboring spot, respectively. These optimal cellular combinations were integrated for all spots to reconstruct the single-cell resolution ST dataset for the spot-based ST dataset.

### Definition of the CCC score

To generate the cell–cell distance matrix $D$, the spatial coordinates of individual cells were used to calculate their Euclidean distances. However, to focus on nearby secretion and paracrine signaling within a specific range, we only considered the cells that were 200 μm apart[79]. Subsequently, the K-nearest neighbors (KNN) algorithm was applied to select the $K$ closest cells from the distance matrix $D$, aiding the construction of a cell graph network by establishing connections between the selected cells. The receptor was used as the query node to ensure the biological relevance of the identified CCCs. A random walk algorithm was employed to filter and score the downstream-activated transcription factors (TFs). Thus, the TFs activated in response to a queried receptor were identified; consequently, we considered only cells with activated TFs as receptor cells. This approach provides a more accurate representation of the intercellular information transfer and communication as it reflects the actual cellular response to signaling events.

To ascertain the co-expression of a specific ligand-receptor pair between the sender cells (of type $A$) expressing a given ligand and the receiver cells (of type B) expressing the corresponding receptor in the cell graph network, we computed the number of cell–cell pairs $(C^0_{A_i,B_j})$ exhibiting this ligand-receptor interaction. This involved identifying the direct neighboring nodes (1-hop away) of the sender cells expressing ligand $i$ and the receiver cells expressing receptor $j$. For each ligand-receptor interaction between cell types $A$ and $B$, there may be distinct cell–cell pair counts.

A permutation test was employed to gauge the significance of these observed counts. This entailed randomly reassigning cell labels and recomputing the ligand-receptor interaction counts. This procedure was iterated ($Z$) times, generating a background distribution $C = C^1_{A_i,B_j}, C^2_{A_i,B_j}, \ldots, C^Z_{A_i,B_j}$. The P-value was then determined by

juxtaposing the observed cell–cell pair counts for the specified ligand-receptor interactions against this background distribution.

Mathematically, the P-value was computed as follows:

$$P_{A_i B_j} = \frac{\mathrm{card}\left\{x \in C | x \geq C^0_{A_i,B_j}\right\}}{Z}, \quad (21)$$

P-values less than 0.05 were considered statistically significant and were used to calculate the CCC score of the ligand-receptor interaction from senders to receivers. This score was computed as $S_{A_i,B_j} = \sqrt{L_{A_i} \times R_{B_j}}$, where $L_{A_i}$ is the gene expression of the ligand $L$ in cell $i$ of cell type $A$ and $R_{B_j}$ is the gene expression of the receptor $R$ in cell $j$ of cell type $B$.

### Benchmark metrics

**Benchmark metrics for integration methods.** Five metrics were used to evaluate the integration methods, one of the metrics being the Pearson correlation coefficient (PCC), which is calculated using the following equation:

$$\mathrm{PCC} = \frac{E[(\widetilde{x}_i - \widetilde{u}_i)(x_i - u_i)]}{\widetilde{\sigma}_i \sigma_i}, \quad (22)$$

where $\mathbf{x}_i$ and $\widetilde{\mathbf{x}}_i$ represent the spatial expression vectors of the $i$-th gene in the ground truth and the predicted results, respectively. Similarly, $u_i$ and $\widetilde{u}_i$ correspond to the average expression value of the $i$-th gene in the ground truth and the predicted result, respectively, and $\sigma_i$ and $\widetilde{\sigma}_i$ represent the standard deviation of the spatial expression of the $i$-th gene in the ground truth and the predicted result, respectively. While evaluating a specific gene, a higher PCC value indicates a higher prediction accuracy for that gene. The PCC value measures the degree of linear association between the ground truth and predicted results for a particular gene.

The evaluation of the integration methods also used the structural similarity index (SSIM). To prepare the data for SSIM calculation, the expression matrix was scaled by adjusting the expression values of each gene to lie in the range of 0–1 as follows:

$$x'_{ij} = \frac{x_{ij}}{\max(\{x_{i1}, \ldots, x_{iM}\})}, \quad (23)$$

where $x_{ij}$ denotes the expression of the $i$-th gene in the $j$-th spot, and $M$ is the total number of spots. Normalizing the expression values facilitated a consistent and comparable evaluation of integration methods using the SSIM metric. After scaling the gene expression values between 0 and 1, the SSIM value for each gene was calculated using the following equation:

$$\mathrm{SSIM} = \frac{\left(2\widetilde{u}_i u_i + C_1^2\right)\left(2\mathrm{cov}(\mathbf{x}'_i, \widetilde{\mathbf{x}}_i) + C_2^2\right)}{\left(\widetilde{u}_i^2 + u_i^2 + C_1^2\right)\left(\widetilde{\sigma}_i^2 + \sigma_i^2 + C_2^2\right)}, \quad (24)$$

To calculate the SSIM value for each gene, we utilized the same definitions of $u_i$, $\widetilde{u}_i$, $\sigma_i$ and $\widetilde{\sigma}_i$ as in the calculation of the PCC value, but for the scaled gene expression. Additionally, $C_1$ and $C_2$ were introduced as constant values and set to 0.01 and 0.03, respectively. The term $\mathrm{cov}(\mathbf{x}_i, \widetilde{\mathbf{x}}_i)$ represents the covariance between the expression vectors of the $i$-th gene in the ground truth ($\mathbf{x}_i$`) and the predicted result ($\widetilde{\mathbf{x}}_i$`). Similar to the PCC value, a higher SSIM value indicates a higher level of prediction accuracy for a given gene.

The z-score for the spatial expression of each gene was calculated for all spots. The root mean square error (RMSE) was computed as

follows:

$$\text{RMSE} = \sqrt{\frac{1}{M}\sum\nolimits_{j=1}^{M}(\widetilde{\mathbf{z}}_{ij} - \mathbf{z}_{ij})^2}, \tag{25}$$

where $\mathbf{z}_{ij}$ and $\widetilde{\mathbf{z}}_{ij}$ are the z-scores of the spatial expression of the $i$-th gene in the $j$-th spot in the ground truth and predicted results, respectively. For a given gene, a lower RMSE value indicates a higher level of prediction accuracy.

The Jensen−Shannon divergence (JSD) uses the relative information entropy, particularly the Kullback−Leibler divergence, to quantify the difference between the two distributions. To calculate the spatial distribution probability of each gene, the following steps were performed:

$$\mathbf{P}_{ij} = \frac{\mathbf{x}_{ij}}{\sum_{j=1}^{M}\mathbf{x}_{ij}}, \tag{26}$$

To calculate the spatial distribution probability of each gene, we assign $\mathbf{x}_{ij}$ as the expression value of the $i$-th gene in the $j$-th spot, where $M$ is the total number of spots and $\mathbf{P}_{ij}$ is the distribution probability of the $i$-th gene in the $j$-th spot. After calculating the spatial distribution probability, the JSD value for each gene was evaluated using the following equations:

$$\text{JSD} = \frac{1}{2}\text{KL}\left(\widetilde{\mathbf{P}}_i\bigg|\bigg|\frac{\widetilde{\mathbf{P}}_i + \mathbf{P}_i}{2}\right) + \frac{1}{2}\text{KL}\left(\mathbf{P}_i\bigg|\bigg|\frac{\widetilde{\mathbf{P}}_i + \mathbf{P}_i}{2}\right), \tag{27}$$

$$\text{KL}\left(\mathbf{a}_i||\mathbf{b}_i = \sum\nolimits_{j=0}^{M}\left(\boldsymbol{a}_{ij}\times\log\frac{\boldsymbol{a}_{ij}}{\boldsymbol{b}_{ij}}\right)\right), \tag{28}$$

where $\mathbf{P_i}$ and $\widetilde{\mathbf{P}}_i$ represent the spatial distribution probability vectors of the $i$-th gene in the ground truth and predicted result, respectively; $\text{KL}(\mathbf{a}_i||\mathbf{b}_i)$ denotes the Kullback−Leibler divergence between the two probability distributions $\mathbf{a}_i$ and $\mathbf{b}_i$; $\mathbf{a}_{ij}$ and $\mathbf{b}_{ij}$ represent the predicted and real probabilities of the $i$-th gene in the $j$-th spot, respectively. For a given gene, a lower JSD value indicates a higher level of prediction accuracy.

To evaluate the relative accuracy of the integration methods for each dataset, an accuracy score (AS) was defined by combining the PCC, SSIM, RMSE, and JSD metrics. For a given dataset, the average PCC, SSIM, RMSE, and JSD values were calculated for all the genes predicted by each integration method. Subsequently, the PCC and SSIM values of the integration methods were sorted in ascending order to obtain $\text{RANK}_{\text{PCC}}$ and $\text{RANK}_{\text{SSIM}}$, respectively. The integration method with the highest PCC/SSIM value had $\text{RANK}_{\text{PCC/SSIM}}$ equal to N, whereas the method with the lowest PCC/SSIM value had the $\text{RANK}_{\text{PCC/SSIM}}$ value of 1. Similarly, the RMSE and JSD values of the integration methods were sorted in the descending order to obtain $\text{RANK}_{\text{RMSE}}$ and $\text{RANK}_{\text{JSD}}$, respectively. The integration method with the highest RMSE/JSD value had $\text{RANK}_{\text{RMSE/JSD}} = 1$, whereas the method with the lowest RMSE/JSD value had $\text{RANK}_{\text{RMSE/JSD}} = \text{N}$. Finally, the average values of $\text{RANK}_{\text{PCC}}$, $\text{RANK}_{\text{SSIM}}$, $\text{RANK}_{\text{RMSE}}$, and $\text{RANK}_{\text{JSD}}$ were determined to obtain the AS value for each integration method as follows:

$$\text{AS} = \frac{1}{4}\left(\text{RANK}_{\text{PCC}} + \text{RANK}_{\text{SSIM}} + \text{RANK}_{\text{RMSE}} + \text{RANK}_{\text{JSD}}\right), \tag{29}$$

The method with the highest AS value exhibited the best performance among the integration methods.

## Benchmark metrics for the CCC prediction method

The Wasserstein distance concept was introduced as a metric to assess the spatial communication tendency in a specific ligand-receptor (L-R) pair. Here, $L$ and $R$ represent the gene expression distributions of the ligand and receptor, respectively. For brevity, we refer to the actual Wasserstein distances between these distributions as $d_{\text{real}}$. To establish a comparative baseline, we constructed random gene expression distributions, $L_r$ and $R_r$, by permuting the coordinates of each data point in L and R. By repeatedly permuting (1000 times in our case) and calculating the Wasserstein distance between $L_r$ and $R_r$, denoted as $d_{\text{simulation}}$, we obtained a set of $d_{\text{simulation}}$ values. Subsequently, the spatial communication tendency was quantified by computing the ratio of $d_{\text{real}}$ to the mean of the $d_{\text{simulation}}$ set, referred to as $d_{\text{ratio}}$. This ratio serves as a measure of the spatial communication tendency specific to the $L$-$R$ pair under consideration.

$$d_{\text{ratio}} = \frac{d_{\text{real}}}{\sum_{i=1}^{n} d_{\text{simulation}_i}/n'} \tag{30}$$

By increasing the number of permutations ($n$), we constructed a null distribution of $d_{\text{real}}$ using the $d_{\text{simulation}}$ set. This null distribution was then utilized in a one-sided permutation test to derive a P-value, indicating the significance of the observed spatial communication tendency. Additionally, left- and right-sided P-values were calculated to distinguish between short- and long-range communications. To quantify the consistency between expected and observed spatial distance tendencies, we employed the DES metric, where a higher value signifies better consistency[69]. Based on their $d_{\text{ratio}}$ and P-values, short- and long-range communications were ranked to form expected communication lists, $L_s$ and $L_l$, respectively. Subsequently, we extracted communications from the CCC tool's results and created observed communication lists, $S$, for each cell type pair. These lists were denoted as $S_n$ and $S_f$, for nearby and distant cell type pairs, respectively. To compute the DES for a particular cell type pair, we considered weighted P-value proportions ($P_{\text{match}}$ and $P_{\text{unmatch}}$) while iterating through the expected communication list. The presence or absence of communication in the observed list determined the addition or subtraction of the corresponding weights, respectively. This approach allowed us to assess the consistency between expected and observed communications for a given cell type pair. A similar methodology was applied to compute the DES for distant cell type pairs. The $P_{\text{match}}$ and $P_{\text{unmatch}}$ values for the $j$-th interaction in $L_s$ are defined as follows:

$$P_{\text{match}}\left(S_n,j\right) = \sum\nolimits_{\substack{lr_j \in s_n \\ j \le i}} \frac{1 - \text{Pvalue}_j}{\Sigma_{lr_j \in s_n}(1 - \text{Pvalue}_j)}, \tag{31}$$

where n represents the total number of matched interactions between $S_n$ and $L_s$. The DES represents the maximum deviation of ($P_{\text{match}} - P_{\text{unmatch}}$) from 0, providing a quantitative measure of consistency between expected and observed spatial interaction tendencies.

## Comparison with other methods

To compare the predictive performance of DeepTalk with that of other methods for predicting the spatial distribution of undetected transcripts, we used a dataset comprising 45 paired ST and scRNA-seq datasets curated by Li et al.[35] These datasets were generated using various techniques, including FISH, osmFISH, seqFISH, MERFISH, STARmap, ISS, EXseq, BaristaSeq, ST, 10X Visium, Slide-seq, Seq-scope, and HDST. STARmap and seqFISH+ST datasets were employed to assess the accuracy of DeepTalk and other methods for cell type decomposition. For the single-cell ST dataset, the cells were separated into distinct groups based on fixed spatial distances and combined to create simulated spots, resulting in a reference dataset. The performances of Tangram, Cell2location, SpatialDWLS, RCTD,

Stereoscope, DestVI, and SPOTlight in predicting cell-type compositions within each spot were evaluated by comparing them with true cell-type compositions, using metrics such as PCC, SSIM, RMSE, JSD, and AS. By utilizing a benchmark dataset comprising the MERFISH, 10X Visium, and ST datasets, we compared the performance of DeepTalk with that of other methods for inferring CCC, including CellCall, CellChat, CellChatV2, CellPhoneDB, CellPhoneDBV3, ICELLNET, iTALK, SingleCellSignalR, Giotto, stLearn, Connectome, NicheNet, COMMOT. All methods were evaluated using their default parameters. For the comparison of NICHES and Scrabin, two methods for inferring CCC at the single-cell resolution, we utilize the same ground-truth ligand-receptor pairs obtained from OmniPath[80] for this analysis.

## Visualize the CCC patterns using UMAP

To visualize the CCC patterns using UMAP, we predicted CCC events mediated by various L-R pairs at single-cell resolution. Each predicted event was assigned a quantitative score reflecting the communication strength, resulting in a matrix where rows represent distinct CCC events, and columns correspond to unique L-R pairs. For dimensionality reduction and visualization, we employed Scanpy, a robust tool for single-cell analysis. Initially, we scaled the data using sc.pp.scale() to normalize the feature values. This was followed by principal component analysis (PCA) using sc.tl.pca(), which helped reduce the dimensionality of the dataset while preserving its main structure. Next, we constructed a neighborhood graph using sc.pp.neighbors(), which identifies cells that are close to each other in the high-dimensional space. This step is crucial for subsequent manifold learning techniques. Finally, we used sc.tl.umap() to craft the UMAP visualizations, thereby enabling the depiction of intricate CCC patterns within a two-dimensional framework.

## Reporting summary

Further information on research design is available in the Nature Portfolio Reporting Summary linked to this article.

## Data availability

This study made use of publicly available datasets. The detailed information of 45 paired spatial transcriptomics and scRNA-seq datasets, along with 32 simulated datasets for assessing the effectiveness of the integration method, were retrieved from https://drive.google.com/drive/folders/1pHmE9cg_tMcouV1LFJFtbyBJNp7oQo9J?usp=sharing[35]. MERFISH VISp data and Smart-Seq2 VISp snRNA-seq data were available at http://github.com/spacetx-spacejam/data. 10x Genomics Visium Fluorescent dataset is available from https://support.10xgenomics.com/spatial-gene-expression/datasets and adult mouse cortical region scRNA-seq data were obtained through GEO under accession number GSE115746. The scRNA-seq and ST data of the human PDAC data were obtained through GEO under accession number GSE111672. Source data for the main figures are provided with this paper. Source data are provided with this paper.

## Code availability

DeepTalk is implemented in the open-source Python using PyTorch, and source code are publicly available at (https://github.com/JiangBioLab/DeepTalk)[81].

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

## Acknowledgements

This work was supported by the National Natural Science Foundation of China (no. T2325009, Q.J.; no. 62032007, Q.J.; no. 32270789, Z.X.), National Science and Technology Major Project of China (no. 2022ZD0117702, Q.J.), and Science, Technology & Innovation Project of Xiongan New Area in China (no. 2022XAGG0117, Q.J.).

## Author contributions

Q.J., Y.J., and Z.X. conceived and designed the study; Q.J., W.Y., Z.X., S.X., M.L., and C.X. performed the research; Z.X., P.W., Y.C., J.Q., B.P., and H.N. collected and constructed the benchmark datasets; W.Y., Z.X., G.X., and Y.C. designed and implemented the computational framework with guidance from Q.J.; Y.J., Z.X., P.W., and X.J. completed downstream analysis work. Z.X., Y.Y., Y. L., Q.D., and F.P. released the source code on GitHub; Q.J., Y.J., W.Y., Z.X., T.W., and M.L. wrote the paper with input from all other authors. All authors read and approved the manuscript.

## Competing interests

The authors declare no competing interests.
