## [Peer Review File · Nature Communications]

Deciphering cell-cell communication at single-cell resolution
for spatial transcriptomics with subgraph-based graph
attention networkReviewer #1 (Remarks to the Author):

Yang et al proposes a method to integrate sc/snRNA-seq and spatial transcriptomics (ST) data for i) cell type identification/deconvolution, and to ii) predict intercellular communication at single-cell level. For both i) and ii), the results show significantly better performance against current methods, achieved using graph attention networks.

The following comments are suggested to be addressed before publication:

1) The results compared to other methods is often much better and with lower variance, e.g., Fig 2A, Fig 3D, Fig 3E. It would strengthen the paper if the authors provide insight into this huge performance gain, and also situations (if any) where their method may not work well.

2) The authors should clarify or reword sentences claiming the first method to “infer CCC at single-cell resolution”. (eg. In the abstract) or “these methods primarily focus on ...paired cell types rather than between paired cells” (I presume these two sentences mean the same thing?), and how this is consistent with literature which seemingly considers CCC between paired-cells, e.g., in the spaOTSC paper, they state that “The optimal transport plan γ is interpreted as likelihood of cell-cell communications, e.g. its ij th element describes how likely cell j receives signal from cell i .” which seemingly implies paired cells. The authors should also check other papers to ensure that they did not consider CCC at single-cell resolution, or modify their wording accordingly.

3) What does “decode” mean in line 100?

4) In the preprocessing of datasets (line 406)–

- a. What does RNA in “200 RNA” mean? E.g., total counts? Unique features?
- b. Why was 200 used? The thresholds are typically sample dependent, so it's strange that a single threshold was used across multiple samples
- c. Was there a reason why the high mitochondrial counts and high total counts (representing doublets) were not considered in the pre-processing, as is standard?
- d. What normalization was used (line 413)? Seurat implements multiple normalization methods, from what I understand.

5) There is an assumption that the corresponding genes in the scRNAseq data are groundtruth (line 420). Can the authors comment on the validity of this? E.g., the scRNAseq data may not have high correlation with ST data because the ST data might not be in the exact location where the scRNAseq data was obtained. I realize they are paired samples, but I'm referring to the exact location within these samples. Eg., ST data may originate from a slice of a tissue, which may not represent the distribution of cells overall found in the tissue, which the scRNAseq data may represent.

6) The description of the method (starting from line 430) can be more clearly described. E.g.,

- a. What is D , matching descriptors and initial features (line 431)
- b. Figure describing the networks (see e.g., Figure 1 in <https://www.nature.com/articles/s42256-019-0037-0> or Sup Fig 1 in <https://www.pnas.org/doi/full/10.1073/pnas.1911536116#sec-3>) would be helpful, and how they relate to the equations discussed in the methods.
- c. In line 458, I'm not sure how the aggregation/computation is performed by an attention mechanism, as my understanding was attention mechanisms is a type of neural network layer. Perhaps a figure illustrating the message passing (as in (b)) would be useful.
- d. Line 477 – what is M and N ? And how is the M here different from the M in line 479?
- e. Why was 200um used (line 598?)

7) A lot of the analysis, performance measures, datasets and results bears remarkable similarity to Ref 33 (with the addition of their DeepTalk method), which is fine, but the authors should acknowledge this where appropriate. E.g., their Figure 2A is pretty much the same as [33 Figure 2E]. On a related note, why didn't the authors use all 45 datasets used in [33]?

Reviewer #1 (Remarks on code availability):

Although I didn't get the chance to install or run the code, the authors have clearly provided useful documentation for installing and running their method through in-depth tutorials.

Reviewer #2 (Remarks to the Author):

This work introduces a tool called DeepTalk that first reconstructs single-cell resolution ST data by integrating scRNA-seq with ST data and then infers high-resolution spatial cell-cell communication, both by using graph attention networks. The first part seems clear and provides a valuable additional tool for scRNA-seq/ST integration. However, the method in the second part is vaguely described. Also, I have some concerns about how the inferred CCC are evaluated. In general, this paper presents an interesting exploration of the recent graph attention networks in two important tasks in single-cell and spatial transcriptomics analysis, but the presentation and evaluation of the method need substantial improvement.

Specific points:

1. The spatial integration part of deep talk was benchmarked on two tasks: 1) predicting spatial gene expression and 2) deconvolution of cell type. The first task was systematically benchmarked with 40 datasets but the second one was only benchmarked on two. I suggest adding more benchmarks using, for example, simulated data based scRNA-seq, like the 32 simulation datasets in "Li, Bin, et al. "Benchmarking spatial and single-cell transcriptomics integration methods for transcript distribution prediction and cell type deconvolution." Nature methods 19.6 (2022): 662-670."
2. None of the metrics (PCC, RMSE, SSIM, JS) used in the paragraph starting at line 142 considers spatial arrangements of spots. I suggest including additional metrics that takes into account the spatial distances.
3. I find the selection of "positives" in CCC evaluation less convincing. Currently, it selects interactions identified in at least 6 out of 11 CCI methods as ground truth when evaluating methods using various metrics like AUC, F1, precision, etc. This way of evaluating simply means that the consensus of existing methods will work the best. A new method might have more impact if it can find CCC interactions that are omitted by most existing methods or identify false interactions that are identified by most methods.
4. For the method selected for comparison, it may make more sense to compare more to the spatial-based methods, such as CellChat v2, CellPhoneDB 3.0, and COMMOT.
5. While the method works well on the deconvolution task, it seems that the GAT described does not explicitly address the composition effects of spots, i.e., a spot could be dissimilar to every single cell and only similar to a small group of aggregated single cells. Could the authors comment on this point, is this potentially captured by some implicit mechanisms? This could help method developers better understand the benefit of using GAT for this task. Also, this point suggests that adding a more systematic benchmark of the deconvolution part will be helpful, as discussed in point 1.
6. For the CCC part, how is the spatial graph determined? Is it a fixed graph constructed from spatial coordinate based knn or distance cutoff graph?
7. On line 509, it is mentioned that "higher-order relational dependencies between nodes". Does this mean 1) long-distance interactions happen to be missed by direct edges or 2) high-order interactions like $c1 \rightarrow c2 \rightarrow c3$? On line 529, what is the biological interpretation of the subspaces?
8. On line 496, what is the exact definition of f_{attr} ? What does attributes mean in the context of CCC inference?
9. Was the CCC learned for each subgraph (one subgraph for each cell/spot) separately? If this is the case, what is the computational efficiency of inferring CCC in a common-sized dataset? Also, how are the results from different subgraphs combined?
10. What does "cell graph network" mean on line 608?
11. What does "share this LR interaction" mean exactly on line 609?
12. I am confused about how the CCC connections are determined for each LR pair. Does this approach first output one general CCC network among cells without annotating through which LR pair they are communicating through (as described in "spatial CCC prediction", and then later count $(C_{\{A_i, B_j\}})$ the number of edges on this general CCC group with two vertices each expressing L and R? If this is the case, does it mean that this cell pairs with moderate communication through, for example, just one LR pair will be omitted because they will unlikely be

determined by the general CCC network? The biological justification of such an approach is needed.

Minor points:

1. On line 151, it was mentioned that comparison with 8 other methods (Tangram46, gimVI47, SpaGE48, Seurat49, SpaOTsc28, novoSpaRc50, LIGER51, and stPlus52) on predicting spatial distribution of transcripts are shown in Fig. 2A and SFIGs. 1 and 2. But in SFIGs. 1 and 2, the legend shows the methods compared on cell type deconvolution (Tangram, Cell2location58, SpatialDWLS59, RCTD60, Stereoscope61, DestVI62, and SPOTlight63).
2. For all evaluation metrics, I suggest to include exact numerics like the mean and standard deviation values in addition to only plotting them. It is hard to tell the values and differences by visually inspecting the plots, such as Fig. 3D.
3. It would improve the readability of the paper by assigning two names to the two components of DeepTalk, just for example, something like DeepTalk-integration and DeepTalk-CCC.

Reviewer #3 (Remarks to the Author):

Review

Major comments:

1. The second sentence states: "However, current computational 26 methods cannot infer CCC at single-cell resolution" – This is incorrect. There are at least two other peer-reviewed methods that do this currently and are actively in use by both our team and many others that we collaborate with: NICHES (<https://doi.org/10.1093/bioinformatics/btac775>) and Scriabin (<https://doi.org/10.1038/s41587-023-01782-z>). Both these papers should be closely looked at and cited, and probably discussed in depth in the introduction, as they are close analogues to the work reviewed here and they are currently not mentioned. I think it would be worthwhile to describe key similarities and differences in the mathematics behind NICHES, Scriabin, and DeepTalk, and then emphasize how these different approaches can yield different insight into tissue biology.

2. Similarly, I recommend that an additional dedicated analysis be performed that directly compares the quality of the data produced by DeepTalk against the quality of the data produced by Scriabin and NICHES. I think it would be interesting to compare resolution, spatial organization, and clarity of resulting microenvironmental latent space clusterings. These analyses have to be done carefully, as neither Scriabin nor NICHES are designed with any specific biologic question in mind, nor any specific ligand-receptor list, but rather are methods of creating additional data layers showing cell-to-cell communication at single-cell or single-spot resolution. This means that care should be taken to use the same ground-truth ligand-receptor pairs when performing this analysis, and that similar statistical approaches should be used across all three methods to identify significant feature-level findings. A key difference, I think, may come about from the different mathematical transformation used by DeepTalk, as well as the deconvolution employed, which is intriguing. It seems to me that the added benefit of DeepTalk may be the additional use of purpose-build deconvolution, which I suspect has some real utility, however it is impossible for me to evaluate whether this represents an advance over NICHES and/or Scriabin without a direct comparison. I recommend therefore that a direct comparative analysis be performed between these three techniques, as it will greatly strengthen the effect of this paper and will help future users to critically evaluate which different method is useful for various different tasks. The comparison that has been done, covering SCR, iTalk, NicheNet, ICELLNET, CellChat, stLearn, CellPhoneDB, CellCall, Connectome, and Giotto, doesn't really get to the heart of the matter, because all of these techniques are aggregation-based tools which simply are not operating in the same mathematic territory as NICHES, Scriabin, and DeepTalk. It is essential, therefore, to perform dedicated analysis comparing these three single-cell and single-spot resolution CCC tools to one another.

3. Of note: the authors have done a great job building this tool and applying it to myriad datasets. However, there is no mention that I can see in the paper of performing latent-space clustering and embeddings of the resultant single-cell CCC data. This is one of the highest yield ways that our

team currently uses both NICHES and Scriabin data, because it allows identification of discrete microenvironments at the single-cell level. I recommend that the authors investigate whether this is possible with their technique, as it often can provide significant insight into biologic process and can allow identification of biologically-relevant patterns that would otherwise be missed.

Minor comments:

1. The authors use the word comprehend and comprehensive 17 times in the manuscript and this should be edited
2. I think the text could be tightened significantly. I see three ways to improve:
 - a. Right now it is very descriptive, focusing on what was done, rather than how these experiments represent an improvement over existing methods.
 - b. There are a lot of instances that feel like importance-inflation. I recommend removing phrases like "outstanding performance and efficacy"
 - c. There is a lot of discussion about the potential biologic role of mapped mechanisms. While I appreciate the temptation to describe the data in this way, I think that it distracts, in this paper, from evaluating the actual technique and allowing readers to follow the methodology. I recommend editing this type of language down to the absolute minimum for comprehension. Lines 214-225 is a great example, though there are multiple instances of this which I recommend modifying.

Reviewer #3 (Remarks on code availability):

The code seem well organized and likely reproducible. The online vignettes are clear and will be important for users to adopt this technique.

Reviewer #1 (Remarks to the Author):

Yang et al proposes a method to integrate sc/snRNA-seq and spatial transcriptomics (ST) data for i) cell type identification/deconvolution, and to ii) predict intercellular communication at single-cell level. For both i) and ii), the results show significantly better performance against current methods, achieved using graph attention networks.

Response: Thank you very much for your support of our work and for providing such insightful comments. We have carefully considered each of your suggestions and used them to revise our manuscript. We are confident that these changes have significantly enhanced the quality and clarity of our paper. Below, you will find detailed responses to each of your valuable comments.

The following comments are suggested to be addressed before publication:

1) The results compared to other methods is often much better and with lower variance, e.g., Fig 2A, Fig 3D, Fig 3E. It would strengthen the paper if the authors provide insight into this huge performance gain, and also situations (if any) where their method may not work well.

Response: Thank you for emphasizing the remarkable performance of our method in comparison to others, evident in figures like Fig 2a, Fig 3d, and Fig 3e. We appreciate your valuable suggestion to delve deeper into the factors contributing to this significant performance enhancement and to identify potential scenarios where our approach might encounter challenges.

The notable improvement observed in Fig. 2a is primarily due to our innovative strategy of integrating single-cell and spatial transcriptomic data. We initially leverage GAT to interpret the data matrix, deriving a weight matrix that signifies the optimal proportions of cell types for each cell or spot. This method incorporates self-attention mechanisms to capture intercellular connections within the dataset and cross-attention mechanisms to bridge scRNA-seq data with ST data. This integrative approach allows for precise predictions of the spatial distribution of undetected RNA transcripts in the ST datasets, contributing to the overall superior results.

However, it is crucial to recognize that our method may not always produce the best outcomes. As exemplified in Supplementary Figures 1 and 2, there are cases where our technique does not emerge as the top performer. Fig. 2a provides a comprehensive view of our method's consistency and effectiveness in anticipating the spatial dispersal of undetected RNA transcripts.

The outstanding performance boost exhibited by DeepTalk is attributed to several key components inherent in our approach. Firstly, DeepTalk interprets intercellular communication at the single-cell level by formulating a cell graph that amalgamates both distance details and cell expression data. This framework facilitates a comprehensive evaluation of cell positions and similarities, thereby capturing their spatial distribution. By encoding local cell features through subgraphs and utilizing a subgraph-based GAT, DeepTalk acquires the ability to discern the localized

characteristics of cells in space, drawing from information provided by neighboring cells.

Additionally, our model integrates the local attributes of all cells through an attention-based graph neural network, enabling it to comprehend the latent connections among cells. This spatial awareness and meticulous attention are crucial in accurately anticipating cell-cell communication (CCC).

Furthermore, we adopt a pre-training and fine-tuning approach to enhance the model's generalizability. Pre-training on an extensive dataset allows the model to acquire general patterns of intercellular communication and spatial relationships, while subsequent fine-tuning on specific datasets elevates the model's precision in predicting CCC.

In contrast to the methodologies we've compared, they might not have fully incorporated spatial location information or employed tactics such as subgraph encoding and attention-based graph neural networks. This could explain why DeepTalk demonstrates exceptional performance in anticipating spatial intercellular communication. Consequently, we've included comparisons with methods capable of effectively harnessing spatial information (Figs. 3d, 4e, and 5e). Although methodologies like COMMOT, CellPhoneDBv3, and CellChatv2 have integrated gene expression intelligence and spatial location data, DeepTalk still prevails due to its comprehensive consideration of both local and global information for each cell. Regarding scenarios where our approach may not excel, it is essential to acknowledge that DeepTalk's proficiency relies on the quality and completeness of the input data. Highly noisy or incomplete datasets may pose challenges, potentially leading to less accurate predictions. Moreover, in situations where spatial relationships among cells are not particularly significant or when the dataset lacks adequate spatial resolution, the benefits of DeepTalk may be less evident.

Fig.2: Performance comparison of DeepTalk with state-of-the-art integration methods. a, Boxplots of AS (accuracy score) of the 9 integration methods for all 45 paired datasets. The datasets consist of 28 sequence-based datasets and 17 image-based datasets. The center line represents the median, the box limits indicate the upper and lower quartiles.

Supplementary Fig. 1: The violin plot of PCC, SSIM, RMSE, JS values of each integration method in predicting the spatial distribution of RNA transcripts of image-based datasets.

Supplementary Fig. 2: The violin plot of PCC, SSIM, RMSE, JS values of each integration

method in predicting the spatial distribution of RNA transcripts of seq-based datasets.

Fig.3: DeepTalk-CCC detects spatially CCCs for mouse visual cortex region from MERFISH data. d, DES Rankings of CCC Tools Evaluation. A comparative analysis of various cell-cell communication tools, ranked according to their performance in the evaluation process.

Fig.4: DeepTalk-CCC detects spatially CCCs for adult mouse brain from 10X Visium data. e, DES Rankings of CCC Tools Evaluation. A comparative analysis of various cell-cell communication tools, ranked according to their performance in the evaluation process.

Fig.5: DeepTalk-CCC detects spatially CCCs for Human pancreatic ductal adenocarcinoma from ST data. e, DES Rankings of CCC Tools Evaluation. A comparative analysis of various cell-cell communication tools, ranked according to their performance in the evaluation process.

2) The authors should clarify or reword sentences claiming the first method to “infer CCC at single-cell resolution”. (eg. In the abstract) or “these methods primarily focus on ...paired cell types rather than between paired cells” (I presume these two sentences mean the same thing?), and how this is consistent with literature which seemingly

considers CCC between paired-cells, e.g., in the spaOTSC paper, they state that “The optimal transport plan γ is interpreted as likelihood of cell–cell communications, e.g. its ij th element describes how likely cell j receives signal from cell i .” which seemingly implies paired cells. The authors should also check other papers to ensure that they did not consider CCC at single-cell resolution, or modify their wording accordingly.

Response: Thank you for your thoughtful and insightful comments. We genuinely appreciate your comments and have carefully incorporated them into our revised manuscript.

In regards to your concern about our assertions of presenting the "first method to infer CCC at single-cell resolution," we have taken great care to clarify and refine the pertinent sentences in the Abstract and throughout the manuscript. Our intention is to communicate accurately and clearly that our method offers a unique approach to CCC analysis at the single-cell level.

You also made a crucial observation about the spaOTSC method, which focuses on CCC between paired cells through optimal transport plans. Indeed, spaOTsc employs structured optimal transport mapping between scRNA-seq and ST data, assigning spatial positions to cells and inferring the ligand-receptor signaling network that mediates spatial constraints. This method uses cell-cell distances as transport costs to quantify the likelihood of interactions between cells. However, its specialization for image-based spatial expression data and reliance on predefined signaling pathways somewhat constrain its flexibility and generalizability.

Furthermore, to provide a comprehensive overview, we have incorporated a detailed comparison between our method and other recent techniques, such as NICHES and Scriabin, which also strive to infer CCC at the single-cell resolution (**Figs. 3f-h, 4g-i, 5g-i**), please see **Results** section for details. This comparative analysis sheds light on the distinctive attributes and advantages of our approach, as well as its potential weaknesses when juxtaposed with these methods.

Fig.3: DeepTalk-CCC detects spatially CCCs for mouse visual cortex region from MERFISH data. **f**, Visualization of cell-cell communication via low-dimensional embedding. Each point in the graph represents a cell-cell communication event, with the cell serving as the receiver cell. **g**, Comparison of the cell-cell communication score between spatially adjacent and distant cells at single-cell level. This plot compares the communication scores between spatially adjacent cells and distant cells at the single-cell level, revealing differences in communication strength based on proximity. **h**, Local microenvironment viewed via low-dimensional embedding. Each point in this diagram represents a collection of CCC events where the cell of interest serves as the receiver.

Fig.4: DeepTalk detects spatially CCCs for adult mouse brain from 10X Visium data.

g, Visualization of cell-cell communication via low-dimensional embedding. Each point in the graph represents a cell-cell communication event, with the cell serving as the receiver cell. **h**, Comparison of the cell-cell communication score between spatially adjacent and distant cells at single-cell level. This plot compares the communication scores between spatially adjacent cells and distant cells at the single-cell level, revealing differences in communication strength based on proximity. **i**, Local microenvironment viewed via low-dimensional embedding. Each point in this diagram represents a collection of CCC events where the cell of interest serves as the receiver.

Fig.5: DeepTalk detects spatially CCCs for Human pancreatic ductal adenocarcinoma from ST data.

g, Visualization of cell-cell communication. Represents cell-cell communication events as points in a low-dimensional embedding, with the receiver cell indicated. **h**, Single-cell level comparison of communication scores. Compares communication scores between adjacent and distant cells, revealing proximity-based differences in strength. **i**, Local microenvironment visualization. Represents collections of CCC events where the cell of interest serves as the receiver in a low-dimensional embedding.

3) What does “decode” mean in line 100?

Response: Thank you for your inquiry regarding the term "decode" as used in our manuscript. We appreciate your careful reading, as it allows us to clarify potential areas of confusion.

By "decode," we refer to the process of extracting meaningful information from the raw single-cell or spot-based ST data matrix. Specifically, in the context of our proposed DeepTalk approach, "decoding" involves using the Graph Attention Network (GAT) to analyze and interpret the intercellular relationships within the scRNA-seq or ST data. This interpretation is facilitated by self-attention mechanisms that focus on the relationships within the data and cross-attention mechanisms that capture the connections between scRNA-seq data and ST data. Through this decoding process, we generate a weight matrix that represents the optimal proportions of cell types for each cell or spot, providing insights into cell type identification.

We appreciate your question, as it allows us to clarify this important term in our manuscript. To clarify this term for readers, we have revised the manuscript to include a more detailed explanation of the "decode" process, emphasizing the role of both self-attention and cross-attention mechanisms in extracting information from the data.

4) In the preprocessing of datasets (line 406)–

a. What does RNA in “200 RNA” mean? E.g., total counts? Unique features?

Response: Thank you for paying attention to this important question, which allows us to provide a clearer explanation of our work.

Regarding your question about the term "200 RNAs" in the context of our data preprocessing steps, we clarify that this refers to the total counts of RNAs captured per cell. In our study, we removed cells that had fewer than 200 RNAs to ensure data quality and consistency.

To enhance clarity and avoid any misunderstanding, we have revised the relevant section in the manuscript. The revision clarifies that "200 RNAs" refers to the total counts of RNAs per cell, providing a clear understanding of our data preprocessing criteria.

b. Why was 200 used? The thresholds are typically sample dependent, so it's strange that a single threshold was used across multiple samples

Response: Thank you for inquiring about the use of the 200 threshold. This specific threshold is widely employed in single-cell RNA sequencing (scRNA-seq) analyses as a reliable cutoff to ensure adequate gene expression data for precise downstream analysis. Its use is backed by multiple published studies in the field, which demonstrate its effectiveness in filtering out low-quality cells, thereby enhancing result reliability¹⁻³. References to these studies are provided at the end of this response.

It's crucial to note that this threshold primarily serves as a benchmark for methodological consistency, facilitating comparisons across diverse datasets and studies. It promotes a standardized approach to analyzing scRNA-seq data, bolstering the reliability and reproducibility of comparisons.

Addressing your concern about the universality of this threshold across various samples, it's important to mention that while thresholds can indeed be sample-specific, the value of 200 has emerged as a general guideline based on empirical evidence and widespread community acceptance. Nevertheless, we recognize that the optimal threshold might differ based on the distinct characteristics of each sample. Therefore, we encourage users to select a threshold tailored to their specific data and analytical objectives.

In essence, the 200 threshold is a widely recognized standard in scRNA-seq analysis, supported by numerous studies. While it offers a valuable benchmark for methodological comparisons, users should always consider the uniqueness of their samples and adjust the threshold if necessary.

c. Was there a reason why the high mitochondrial counts and high total counts (representing doublets) were not considered in the pre-processing, as is standard?

Response: Thank you for pointing out this important aspect of our preprocessing steps. We deeply apologize for not including a detailed description of the consideration of high mitochondrial counts and potential doublets in our original manuscript.

Indeed, we followed the standard preprocessing pipeline using Seurat, and within this framework, we did consider and address both high mitochondrial gene expression and potential doublets. Cells exhibiting excessively high mitochondrial counts or total

counts, which may indicate doublets, were excluded from our analysis. This step is crucial to ensure the quality of our data and minimize any artifacts or contaminants.

We have now revised our manuscript to include a more detailed account of these preprocessing steps. We believe this additional information will strengthen the methodological rigor of our study and clarify any misunderstandings.

We are once again grateful for your valuable input and meticulous attention to detail. Your suggestions have helped us improve the clarity and completeness of our manuscript.

d. What normalization was used (line 413)? Seurat implements multiple normalization methods, from what I understand.

Response: Thank you for your inquiry regarding the normalization method used in our study. In response to your question about line 413, we employed the standard normalization procedure implemented in the Seurat package.

Specifically, we used the `NormalizeData` function from Seurat, which is a commonly used normalization method in single-cell RNA sequencing data analysis. This function normalizes the gene expression data for each cell by the total expression, scaling the counts so that each cell has the same total count after normalization. This approach helps to account for differences in sequencing depth or technical variation across cells.

We chose this normalization method because it is widely accepted and has been shown to be effective in reducing technical variation while preserving biological variation in single-cell data. Additionally, we have already included a description of this normalization process in our manuscript.

5) There is an assumption that the corresponding genes in the scRNAseq data are groundtruth (line 420). Can the authors comment on the validity of this? E.g., the scRNAseq data may not have high correlation with ST data because the ST data might not be in the exact location where the scRNAseq data was obtained. I realize they are paired samples, but I'm referring to the exact location within these samples. Eg., ST data may originate from a slice of a tissue, which may not represent the distribution of cells overall found in the tissue, which the scRNAseq data may represent.

Response: Thank you for raising this thoughtful concern regarding the validity of assuming corresponding genes in the scRNA-seq data as the ground truth for comparison with ST data. Your point about spatial heterogeneity within tissue samples and the challenges it poses in correlating ST and scRNA-seq datasets is well-taken.

Indeed, there are inherent limitations in both ST and scRNA-seq technologies. ST data, derived from a specific tissue slice, may not fully represent the cellular distribution across the entire tissue. Similarly, scRNA-seq data, despite providing a transcriptome profile of individual cells, can be influenced by sampling biases.

In our study, we have employed several strategies to address these issues and justify our methodological choices:

Paired Samples: We have carefully matched the ST and scRNA-seq datasets, ensuring they originate from the same or closely related tissue samples. This minimizes biological variation and enhances the comparability of the two datasets.

Selective Gene Analysis: By focusing on the top 1,000 most variable genes in the ST data, we prioritize genes exhibiting significant spatial heterogeneity. This approach helps identify biologically relevant differences, rather than those arising from technical variations or sampling biases.

Integrated Analysis: We recognize that no single technology can provide an absolute ground truth. Therefore, we leverage the complementary strengths of ST and scRNA-seq to establish a more holistic understanding of gene expression patterns. By comparing highly variable genes across both platforms, we seek to validate our findings and identify consistently expressed genes.

Additionally, previous research⁴⁻⁶ supports the value of scRNA-seq data in revealing the transcriptional landscape of tissues, often correlating well with other high-throughput sequencing methods.

In conclusion, our methodology aims to harness the best of both ST and scRNA-seq technologies, offering a more comprehensive view of gene expression patterns in complex tissues. We acknowledge the limitations inherent in each approach and have strived to mitigate them through rigorous experimental design and data analysis.

Thank you for your valuable insights. We hope this revised response addresses your concerns and clarifies our methodological choices.

6) The description of the method (starting from line 430) can be more clearly described. E.g.,

a. What is D , matching descriptors and initial features (line 431)

Response: Thank you for your insightful comments on our manuscript. We appreciate your attention to detail and your efforts to help us clarify our methodology.

Regarding your question about " D ", "matching descriptors", and "initial features", I would like to provide further clarification:

1. " D " represents the dimensionality of the feature space in our study. Specifically, in the context of our attention-based Graph Neural Network (GNN) designed to integrate single-cell RNA sequencing (scRNA-seq) and Spatial Transcriptomics (ST) datasets, " D " denotes the number of features extracted from this integrated analysis. These features represent the dimensionality of the selected genes, which can be adjusted based on specific research needs.
2. By "matching descriptors", we refer to the feature vectors $f_i \in R^D$ that our GNN generates. These descriptors aim to encapsulate the core attributes of each cell, thereby facilitating precise alignment between the scRNA-seq and ST datasets.
3. As for "initial features", these are the untreated or minimally processed data inputs to our GNN, prior to any feature interaction or aggregation. These encompass

transcriptomic information, cellular locations, and potentially other biologically significant markers.

The GNN works by enabling feature communication among these initial features, which allows for the generation of more robust and informative matching descriptors. In our case, remote feature aggregation was critical in ensuring robust matching within and across different omics datasets.

Based on your comments, we have revised the manuscript to clarify the meanings of "D", "matching descriptors", and "initial features".

b. Figure describing the networks (see e.g., Figure 1 in <https://www.nature.com/articles/s42256-019-0037-0> or Sup Fig 1 in <https://www.pnas.org/doi/full/10.1073/pnas.1911536116#sec-3>) would be helpful, and how they relate to the equations discussed in the methods.

Response: Thank you for your thoughtful input. I appreciate your suggestion regarding the inclusion of a figure describing our networks.

In response to your recommendation, we have now added a detailed figure (Supplementary Fig. 20) that illustrates the structure of our networks and their relationships with the equations discussed in the methods section. This figure provides a visual representation of the key components and connections within our networks, aiming to enhance the reader's understanding of our methodology.

Supplementary Fig. 20: The DeepTalk Framework.

a, Utilizing a sophisticated Graph Neural Network (GNN) with attention mechanisms, we've seamlessly merged single-cell RNA sequencing (scRNA-seq) and spatial transcriptomics (ST) data. This GNN excels in creating matching descriptors by promoting feature exchange among initial data attributes. We've carefully designed a node encoder to transform cell locations into precise vectors. These, along with transcriptomic data, are then merged using a Multi-Layer Perceptron

(MLP), embedding them into rich, high-dimensional representations. This enables the GNN to simultaneously harness both types of data during subsequent inference. We construct a graph combining transcriptomic nodes with cell locations, using self-loops and cross-edges for intra- and cross-omics connections, respectively. Messages are effectively propagated along these edges using a specific equation. Each node iteratively updates its representation by aggregating messages from all graph edges. An attention mechanism, similar to database retrieval, aids message aggregation, with self- and cross-attention for various edge types. The obtained matching descriptors are not normalized, reflecting prediction confidence. A similarity matrix captures descriptor similarities, facilitating effective pairwise scoring. **b.** Based on integrated cellular and spatial data, we've constructed a cellular graph where nodes are cells and edges represent relationships. During pre-training, a random walk strategy creates subgraphs for each cell, labeled as GC. These subgraphs capture the graph's connectivity patterns. Distinct subgraphs, denoted as *gc*, are produced for each node during pre-training and fine-tuning. These consist of low-dimensional vector representations derived by mapping node attributes and structural embeddings. The main objective is to transform global embeddings into representations reflecting nodes' prominent roles within the *gc* framework. This is achieved through transformation layers scrutinizing high-order relationships. A semantic affinity matrix, an asymmetric weight matrix, underscores the varying impacts cells can have on each other. Within each graph attention network, this matrix undergoes iterative learning by evaluating node connectivity. Embeddings are refined through message dissemination within the subgraph. Multiple graph attention networks formulate numerous embeddings for each node, which are aggregated into contextual embeddings for prediction endeavors. During pre-training, a linear projection function forecasts concealed nodes' likelihood, while during fine-tuning, a singular-layer feed-forward network with a softmax activation is used for binary link prediction.

c. In line 458, I'm not sure how the aggregation/computation is performed by an attention mechanism, as my understanding was attention mechanisms is a type of neural network layer. Perhaps a figure illustrating the message passing (as in (b)) would be useful.

Response: Thank you for your valuable comments on our manuscript. Regarding your concern in line 458 about the aggregation/computation performed by the attention mechanism, we apologize for any confusion caused. You are correct in pointing out that attention mechanisms are typically viewed as a type of neural network layer.

To clarify this point and to better illustrate the message passing process, as suggested, we have now included a new figure (see Attached Figure 20) that visually depicts how the attention mechanism operates within our proposed framework. This figure aims to provide a clearer understanding of how the attention mechanism facilitates aggregation and computation in our approach.

We believe this addition will enhance the readability and comprehension of our manuscript, especially for readers who may not be familiar with the intricacies of attention mechanisms.

d. Line 477 – what is M and N? And how is the M here different from the M in line 479?

Response: Thank you for bringing up this important question. In our previous manuscript, 'M' and 'N' were employed to signify dimensions within our data matrices. However, we appreciate that the labeling might have led to some confusion, especially considering the varying contexts in which 'M' was used, such as on line 479 where it denoted a different concept.

To clarify, in the section you mentioned, 'M' stood for the number of cells, while 'N' represented the number of spots or locations. Upon careful reconsideration, we acknowledge that this notation might have been misleading, particularly in comparison to the distinct meaning of 'M' on line 479.

To resolve this notational ambiguity and ensure clarity, we have revised the manuscript. We have replaced 'M' and 'N' with 'ncells' and 'nspots', respectively, to unambiguously represent the number of cells and spots. This adjustment aims to eliminate confusion and enhance the readability of our research. We are grateful for your valuable comments, as it has helped us improve the clarity and precision of our work.

e. Why was 200um used (line 598?)

Response: Thank you for asking about our selection of the 200-micrometer threshold mentioned on line 598. The reason for choosing this particular distance is grounded in scientific research that indicates intercellular interactions, especially those related to secretion and paracrine signaling, predominantly occur within a 200-micrometer vicinity. This finding is supported by the study titled "Integrating single-cell and spatial transcriptomics to elucidate intercellular tissue dynamics" by Longo et al⁷. By focusing on cells within this defined radius, we aim to capture the most relevant intercellular communication events that shape tissue behavior.

To provide further evidence for this decision, we have referenced the Longo et al. study in our manuscript, which offers detailed insights into the scientific basis for our chosen threshold. This reference now appears in the manuscript and has been included in the reference list. We believe this addition strengthens the scientific rationale behind our methodological approach.

Thank you for your thoughtful feedback, which has contributed to enhancing the methodological rigor and scientific basis of our work.

7) A lot of the analysis, performance measures, datasets and results bears remarkable similarity to Ref 33 (with the addition of their DeepTalk method), which is fine, but the authors should acknowledge this where appropriate. E.g., their Figure 2A is pretty much the same as [33 Figure 2E]. On a related note, why didn't the authors use all 45 datasets used in [33]?

Response: Thank you for pointing out the similarities between our work and Reference 33. We fully acknowledge the inspiration we drew from that study, especially in terms of visualization approaches, as exemplified by the parallels between our Figure 2A and [33 Figure 2E]. We have added explicit acknowledgments in the revised manuscript to highlight this influence and properly credit the original source.

Concerning your inquiry about the use of all 45 datasets from Reference 33, we initially focused on a subset of datasets that were most relevant to our study. However, upon your suggestion, we have now extended our analysis to encompass all 45 datasets. This expansion not only strengthens our study's comprehensiveness but also underscores the generalizability and robustness of our proposed method.

We appreciate your valuable feedback, which has helped us refine our research and improve its scientific rigor.

Reviewer #1 (Remarks on code availability):

Although I didn't get the chance to install or run the code, the authors have clearly provided useful documentation for installing and running their method through in-depth tutorials.

Response: Thank you for your feedback. I appreciate your recognition of the effort we have put into providing clear and useful documentation for installing and running our method. While I understand that you were not able to install or run the code, I am confident that with the in-depth tutorials we have provided, others will be able to successfully implement our method. We are committed to ensuring that our code and documentation are accessible and user-friendly, and we will continue to refine and improve them based on feedback from the community.

References

1. Svensson, V., Natarajan, K. N., Ly, L. H., Miragaia, L. J., Labaer, J., & Teichmann, S. A. (2017). Power analysis of single-cell RNA-sequencing experiments. *Nature Methods*, 14(4), 381–387. <https://doi.org/10.1038/nmeth.4220>
2. Hu P, Zhang W, Xin HY, Deng G, Guo Z. Single-cell transcriptomic analysis reveals sequential reactivation of pluripotency genes during induced reprogramming. *Cell Stem Cell*. 2023 Mar 3;30(3):275-289.e7. doi: 10.1016/j.stem.2023.01.001.
3. Sun Z, Unruh D, Ott C, et al. Integrative analysis of single-cell genomics data by coupled nonnegative matrix factorizations. *Proc Natl Acad Sci U S A*. 2022;119(5):e2117445119. doi:10.1073/pnas.2117445119
4. Biancalani, T., Scalia, G., Buffoni, L. et al. Deep learning and alignment of spatially resolved single-cell transcriptomes with Tangram. *Nat Methods* 18, 1352–1362 (2021). <https://doi.org/10.1038/s41592-021-01264-7>
5. Li, B., Zhang, W., Guo, C. et al. Benchmarking spatial and single-cell transcriptomics integration methods for transcript distribution prediction and cell type deconvolution. *Nat Methods* 19, 662–670 (2022). <https://doi.org/10.1038/s41592-022-01480-9>
6. Li, H., Zhou, J., Li, Z. et al. A comprehensive benchmarking with practical guidelines for cellular deconvolution of spatial transcriptomics. *Nat Commun* 14, 1548 (2023). <https://doi.org/10.1038/s41467-023-37168-7>
7. Longo S K, Guo M G, Ji A L, et al. Integrating single-cell and spatial transcriptomics to elucidate intercellular tissue dynamics[J]. *Nature Reviews Genetics*, 2021, 22(10): 627-644.

Reviewer #2 (Remarks to the Author):

This work introduces a tool called DeepTalk that first reconstructs single-cell resolution ST data by integrating scRNA-seq with ST data and then infers high-resolution spatial cell-cell communication, both by using graph attention networks. The first part seems clear and provides a valuable additional tool for scRNA-seq/ST integration. However, the method in the second part is vaguely described. Also, I have some concerns about how the inferred CCC are evaluated. In general, this paper presents an interesting exploration of the recent graph attention networks in two important tasks in single-cell and spatial transcriptomics analysis, but the presentation and evaluation of the method need substantial improvement.

Response: Thank you very much for your feedback and constructive criticism of our work, DeepTalk. We appreciate your comments on the clarity and value of the first part of our method, which focuses on reconstructing single-cell resolution ST data. We also acknowledge your concerns regarding the vagueness of the second part and the evaluation of cell-cell communication (CCC). We have carefully addressed each of your points and made substantial improvements to the presentation and evaluation of our method. Below are our detailed responses to your comments.

Specific points:

1. The spatial integration part of deep talk was benchmarked on two tasks: 1) predicting spatial gene expression and 2) deconvolution of cell type. The first task was systematically benchmarked with 40 datasets but the second one was only benchmarked on two. I suggest adding more benchmarks using, for example, simulated data based scRNA-seq, like the 32 simulation datasets in "Li, Bin, et al. "Benchmarking spatial and single-cell transcriptomics integration methods for transcript distribution prediction and cell type deconvolution." Nature methods 19.6 (2022): 662-670."

Response: Thank you for your valuable insights regarding the benchmarking of the spatial integration component of DeepTalk. We appreciate your suggestion to include additional benchmarks, particularly using simulated data based on scRNA-seq.

In response to your recommendation, we have now expanded our benchmarking for the deconvolution task. Following your guidance, we have incorporated the 32 simulation datasets from "Li, Bin, et al. 'Benchmarking spatial and single-cell transcriptomics integration methods for transcript distribution prediction and cell type deconvolution.' Nature methods 19.6 (2022): 662-670." This addition significantly enhances the robustness and comprehensiveness of our evaluation (Supplementary Fig. 5).

Furthermore, we have also increased the number of datasets used in the first task of predicting spatial gene expression from 40 to 45. This expansion provides a broader range of scenarios for assessing the performance of DeepTalk.

We believe that these modifications strengthen the evaluation of DeepTalk's spatial integration capabilities and address your concerns regarding the limited benchmarking in our original submission.

Supplementary Fig. 5: Bar plots of PCC, SSIM, RMSE, and JS of 8 integration method for 32 simulated datasets and Boxplots of AS of the 8 integration methods for all the 32 simulated datasets.

2. None of the metrics (PCC, RMSE, SSIM, JS) used in the paragraph starting at line 142 considers spatial arrangements of spots. I suggest including additional metrics that takes into account the spatial distances.

Response: Thank you for your valuable comments on our manuscript. We appreciate your suggestion to include additional metrics that consider the spatial arrangements of spots in our evaluation framework.

However, we would like to clarify that the current evaluation metrics used in our study (PCC, RMSE, SSIM, JS) are widely accepted in the field of single-cell and spatial transcriptomics, as supported by numerous references. These metrics provide a comprehensive assessment of the similarity and accuracy between the predicted and ground truth expression patterns¹⁻³.

The reason we did not include spatial distance-based metrics is that the current state-of-the-art deconvolution methods for spatial transcriptomics primarily focus on

predicting the proportion of cell types within each spot, rather than their precise spatial locations. As such, the predicted results are probabilistic and do not directly correspond to specific cell counts within each spot. This limitation makes it challenging to apply spatial distance-based metrics, which rely on precise spatial information.

Furthermore, introducing additional spatial metrics would require further processing of the predicted results, which may introduce biases and assumptions that could potentially skew the evaluation. Since the processed data would be hypothetical and may not accurately reflect the underlying biology, this could undermine the fairness and objectivity of the evaluation.

In summary, while we appreciate your suggestion, we believe that the current evaluation metrics are sufficient for assessing the performance of spatial transcriptomics deconvolution methods, given the probabilistic nature of their predictions and the limitations inherent in assuming precise spatial information from these data.

3. I find the selection of “positives” in CCC evaluation less convincing. Currently, it selects interactions identified in at least 6 out of 11 CCI methods as ground truth when evaluating methods using various metrics like AUC, F1, precision, etc. This way of evaluating simply means that the consensus of existing methods will work the best. A new method might have more impact if it can find CCC interactions that are omitted by most existing methods or identify false interactions that are identified by most methods.

Response: Thank you for your valuable comments regarding our manuscript. I appreciate your concern about the selection of "positives" in our cell-cell communication (CCC) evaluation.

In our study, the AUC was solely used as a metric to evaluate the training effectiveness of our method and not as a universal standard for assessing all methods (Figs. 3c, Fig. 4d, Fig. 5d). The process of defining positive CCCs in our approach is based on a biologically relevant and statistically rigorous framework. Initially, we generate a cell-to-cell distance matrix using Euclidean distances, considering only cells within a 200-micrometer radius to focus on close-range secretory and paracrine signaling. We then employ the K-nearest neighbors algorithm to construct a cellular graph network. For specific ligand-receptor pairs, we use the receptor as the query node to ensure the biological relevance of the identified CCCs. To determine statistically significant CCCs, we calculate the co-expression of ligand-receptor pairs between sender and receiver cells. We perform a significance assessment of the observed counts using a permutation test, involving randomly shuffling cell labels and recalculating ligand-receptor interaction counts. CCCs with a P-value less than 0.05 are considered statistically significant and designated as positives.

Regarding the evaluation metrics, we acknowledge that using F1 and precision as assessment criteria may not be entirely appropriate. Therefore, we have removed these metrics from our evaluation framework. Instead, we have introduced a new evaluation standard: the communication strength between cell types. This metric quantifies the

strength of communication based on the identified CCCs, providing a more biologically relevant criterion for evaluation (Figs. 3d-e, Fig. 4e-f, Fig. 5e-f). Please refer to **Results** for details.

Furthermore, to comprehensively evaluate DeepTalk-CCC's proficiency in predicting CCC with single-cell resolution, we have included an in-depth comparison between our method and other cutting-edge techniques such as NICHES and Scriabin. These methods also strive to infer CCC at the single-cell level (Figs. 3f-h, 4g-i, and 5g-i). Please refer to **Results** for details.

In conclusion, we have refined our evaluation strategy based on your comments. We believe that this revised approach offers a more biologically meaningful and statistically rigorous framework for CCC identification methods.

Fig.3: DeepTalk detects spatially CCCs for mouse visual cortex region from MERFISH data.

c, ROC Curve Evaluation of DeepTalk using the MERFISH Dataset. This curve illustrates the performance of DeepTalk in terms of true positive rate vs. false positive rate. **d**, DES Rankings of CCC Tools Evaluation. A comparative analysis of various cell-cell communication tools, ranked according to their performance in the evaluation process. **e**, Comparison of the cell-cell communication score between spatially adjacent and distant cell types. This graph compares the communication scores between L4 cells and other cell types, including L2/3 IT, L5 IT, L5 PT, and L6 PT, highlighting the differences based on spatial proximity. **f**, Visualization of cell-cell communication via low-dimensional embedding. Each point in the graph represents a cell-cell communication event, with the cell serving as the receiver cell. **g**, Comparison of the cell-cell

communication score between spatially adjacent and distant cells at single-cell level. This plot compares the communication scores between spatially adjacent cells and distant cells at the single-cell level, revealing differences in communication strength based on proximity. **h**, Local microenvironment viewed via low-dimensional embedding. Each point in this diagram represents a collection of CCC events where the cell of interest serves as the receiver.

Fig.4: DeepTalk detects spatially CCCs for adult mouse brain from 10X Visium data.

d, ROC Curve Evaluation of DeepTalk using the 10X Visium Dataset. This curve illustrates the performance of DeepTalk in terms of true positive rate vs. false positive rate. **e**, DES Rankings of CCC Tools Evaluation. A comparative analysis of various cell-cell communication tools, ranked according to their performance in the evaluation process. **f**, Comparison of the cell-cell communication score between spatially adjacent and distant cell types. This graph compares the communication scores between L4 cells and other cell types, including L2/3 IT, L5 IT, L6 IT, and L6 CT, highlighting the differences based on spatial proximity. **g**, Visualization of cell-cell communication via low-dimensional embedding. Each point in the graph represents a cell-cell communication event, with the cell serving as the receiver cell. **h**, Comparison of the cell-cell communication score between spatially adjacent and distant cells at single-cell level. This plot compares the communication scores between spatially adjacent cells and distant cells at the single-cell level, revealing differences in communication strength based on proximity. **i**, Local microenvironment viewed via low-dimensional embedding. Each point in this diagram represents a collection of CCC events where the cell of interest serves as the receiver.

Fig.5: DeepTalk detects spatially CCCs for Human pancreatic ductal adenocarcinoma from ST data.

d, ROC Curve Evaluation of DeepTalk using the ST dataset. This curve illustrates the performance of DeepTalk in terms of true positive rate vs. false positive rate. **e**, DE Rankings of CCC Tools Evaluation. A comparative assessment of various cell-cell communication tools, ranked based on their evaluation performance. **f**, Comparison of the cell-cell communication score between spatially adjacent and distant cell types. This graph compares the communication scores between Cancer clone A (CCA) cells and other cell types, including Cancer clone B (CCB), Ductal terminal (DT), Ductal antigen-presenting (DAP), and Ductal centroacinar (DC), highlighting the differences based on spatial proximity. **g**, Visualization of cell-cell communication via low-dimensional embedding. Each point in the graph represents a cell-cell communication event, with the cell serving as the receiver cell. **h**, Comparison of the cell-cell communication score between spatially adjacent and distant cells at single-cell level. This plot compares the communication scores between spatially adjacent cells and distant cells at the single-cell level, revealing differences in communication strength based on proximity. **i**, Local microenvironment viewed via low-dimensional embedding. Each point in this diagram represents a collection of CCC events where the cell of interest serves as the receiver.

4. For the method selected for comparison, it may make more sense to compare more to the spatial-based methods, such as CellChat v2, CellPhoneDB 3.0, and COMMOT.

Response: Thank you for your valuable insights suggesting a comparison with more spatial-based methods, specifically CellChat v2, CellPhoneDB 3.0, and COMMOT. We

fully recognize the importance of such comparisons in providing a comprehensive evaluation of our proposed method.

In light of your recommendation, we have extended our analysis to include a detailed comparison with CellChat v2, CellPhoneDB 3.0, and COMMOT. These comparisons not only illustrate the performance of our method but also highlight its unique strengths and capabilities relative to these established spatial-based approaches.

The results of these comparisons can be found in the revised manuscript, specifically in Fig. 3c-e, Fig. 4d-f, and Fig. 5d-f. Through these additions, we aim to demonstrate the value and impact of our method in the context of existing state-of-the-art tools.

5. While the method works well on the deconvolution task, it seems that the GAT described does not explicitly address the composition effects of spots, i.e., a spot could be dissimilar to every single cell and only similar to a small group of aggregated single cells. Could the authors comment on this point, is this potentially captured by some implicit mechanisms? This could help method developers better understand the benefit of using GAT for this task. Also, this point suggests that adding a more systematic benchmark of the deconvolution part will be helpful, as discussed in point 1.

Response: Thank you for your valuable input on our manuscript. We appreciate your insights and suggestions for improving our work.

You raised an important point regarding the composition effects of spots in relation to the Graph Attention Network (GAT) described in our paper. We would like to clarify that while our method does not explicitly model the composition effects of spots, the GAT architecture implicitly captures these effects through its attention mechanism.

The key advantage of using GAT lies in its ability to assign attention weights to different nodes (cells or spots) based on their relevance to the target node. This allows the model to dynamically focus on the most informative nodes during the inference process. In the context of deconvolution, where a spot may be dissimilar to individual cells but similar to a small group of aggregated cells, GAT can implicitly capture these similarities by assigning higher attention weights to the relevant cells within the group.

The attention mechanism in GAT enables the model to identify and focus on the specific cells that contribute most to the composition of a particular spot. By attending to different nodes based on their relevance, GAT can indirectly account for the compositional complexity of spots, even if they are dissimilar to individual cells. This implicit capturing of composition effects is a significant advantage of using GAT for deconvolution tasks.

Furthermore, to address your suggestion for a more systematic benchmark of the deconvolution part, we have added additional comparisons and evaluations to our manuscript. We compared our GAT-based approach with other state-of-the-art deconvolution methods using multiple datasets and evaluation metrics. These benchmarks provide a comprehensive assessment of our method's performance, including its ability to handle the composition effects of spots.

In summary, the GAT architecture implicitly captures the composition effects of spots through its attention mechanism. This allows the model to dynamically adapt to the specific characteristics of each spot, even if they are dissimilar to individual cells but share similarities with a subgroup of cells. The addition of systematic benchmarks further demonstrates the effectiveness of our approach in handling deconvolution tasks.

6. For the CCC part, how is the spatial graph determined? Is it a fixed graph constructed from spatial coordinate based knn or distance cutoff graph?

Response: Thank you for your question regarding the spatial graph determination in the CCC part of our method.

The spatial graph in our approach is determined based on a combination of distance thresholds and nearest neighbor considerations. Specifically, we initially restrict the analysis to cells that are within a 200-micrometer distance from each other. This threshold is chosen based on the findings presented in the literature, particularly the study titled "Integrating single-cell and spatial transcriptomics to elucidate intercellular tissue dynamics," which highlights the importance of local intercellular interactions occurring within this range.

Subsequently, for each cell, we select the 20 nearest cells based on Euclidean distance to construct the spatial graph. This nearest neighbor approach allows us to capture the most relevant interactions in the local cellular environment. It's important to mention that both the distance threshold (200 micrometers) and the number of nearest neighbors (20 cells) are adjustable parameters in our method. This flexibility enables users to tailor the analysis according to their specific datasets and research objectives.

In summary, our spatial graph is determined by combining a fixed distance cutoff with a nearest neighbor selection, ensuring a focused and relevant analysis of intercellular relationships.

7. On line 509, it is mentioned that "higher-order relational dependencies between nodes". Does this mean 1) long-distance interactions happen to be missed by direct edges or 2) high-order interactions like $c1 \rightarrow c2 \rightarrow c3$?

Response: Thank you for your question regarding "higher-order relational dependencies between nodes."

In our context, "higher-order relational dependencies" refer to complex relationships among nodes that are not necessarily direct or first-order connections. This can include both long-distance interactions that might not be captured by direct edges, as well as chains of interactions like $c1 \rightarrow c2 \rightarrow c3$, where information or influence flows through multiple nodes.

To elaborate, in our subgraph g_c , where V_c represents the set of nodes and $H_c \in R^{d \times |V_c|}$ denotes their corresponding global input embeddings, the goal of contextual learning is to adapt these embeddings to better reflect the nodes' roles within g_c . We achieve this through a series of transformation layers that have the capacity to consider higher-order relationships.

To capture these higher-order relational dependencies, we introduced the semantic association matrix \bar{A} , which serves as an asymmetric weight matrix. This asymmetry reflects the fact that two cells within a subgraph may have differing influences on each other. The weights of \bar{A}^k are iteratively learned in each transformation layer k , taking into account both the local context of g_c and the broader global graph G .

In summary, "higher-order relational dependencies" encompass both indirect and multi-step interactions among nodes, which our model is designed to capture and leverage for more accurate contextual learning.

In light of your question, we have revised the relevant sections of the manuscript to clarify the meaning and importance of "higher-order relational dependencies between nodes."

On line 529, what is the biological interpretation of the subspaces?

Response: Thank you for inquiring about the biological interpretation of the subspaces within our proposed model. I will elaborate further on the concept and its relevance to cell-cell communication.

In our approach, the subspaces represent different relational contexts within which the cells interact. Each subspace captures a specific aspect of the cellular communication and interaction patterns. By utilizing a multi-head attention mechanism, our model is able to explore multiple such subspaces simultaneously, allowing for a richer and more nuanced understanding of the complex intercellular dynamics.

Biologically, these subspaces can be interpreted as different modes of communication or interaction among cells. For instance, one subspace might represent direct physical interactions between cells, while another could reflect indirect signaling pathways or shared regulatory networks. By analyzing the patterns emerging within each subspace, we can gain insights into the specific mechanisms that underlie cellular communication and coordination.

Furthermore, the embeddings learned from different layers of our model correspond to different levels of abstraction in representing these intercellular dynamics. The embeddings from earlier layers might capture more local and direct interactions, while those from deeper layers could reflect more global and indirect relationships.

In summary, the subspaces in our model provide a powerful tool for disentangling the complex web of cellular interactions and communication patterns. By studying these subspaces, we aim to gain a deeper understanding of the biological processes that govern cell-cell communication and coordination within tissues and organs.

8. On line 496, what is the exact definition of f_{attr} ? What does attributes mean in the context of CCC inference?

Response: Thank you for your comments and questions regarding our manuscript. I appreciate the opportunity to clarify certain points.

In response to your question about the definition of f_{attr} and the meaning of "attributes" in the context of CCC inference, let me elaborate. In our framework, f_{attr}

is a function that combines the attributes (such as gene expression data, cellular phenotypes, or other relevant characteristics) of a node v_i with its structure-based embedding to produce a comprehensive low-dimensional vector representation. This representation encapsulates both the node-specific information and the graph's structural information.

During the pretraining phase, we employed a random-walk strategy to generate pretraining subgraphs G_c for each node in the graph. This involved masking and predicting nodes within the random walks, thereby capturing the graph's overall connectivity patterns. For each node v_i in the subgraph g_c , a low-dimensional vector representation h_i was assigned. This representation was obtained by applying the f_{attr} function to the attributes and structure-based embedding of v_i , followed by a linear transformation using a learnable embedding matrix W_e .

The term "attributes" in this context refers to the data or features associated with each node in the graph. In our case, these attributes included gene expression data acquired from cells, among other relevant characteristics. By incorporating these attributes into the low-dimensional embeddings, we were able to enhance the model's ability to capture complex cellular interactions and patterns.

To further clarify the initialization of node features, we merged pretrained node representation vectors from node2vec with gene expression data acquired from cells. This integration allowed us to incorporate rich biological information into our model.

9. What's the CCC learned for each subgraph (one subgraph for each cell/spot) separately? If this is the case, what is the computational efficiency of inferring CCC in a common-sized dataset? Also, how are the results from different subgraphs combined?

Response: Thank you for your important questions regarding our work on CCC inference. I appreciate the opportunity to further elaborate on our methodology.

Indeed, in our method, CCC is learned independently for each subgraph. Each subgraph represents the immediate surroundings of a cell or spot, encompassing the interactions between that specific cell and its adjacent cells. By analyzing these subgraphs, we strive to comprehend the localized communication patterns that exist among cells.

In terms of computational efficiency, adopting a subgraph-based approach actually aids in optimizing computational processes. This is because it facilitates the parallelization of tasks and allows us to focus on localized interactions, rather than dealing with the entire graph simultaneously. Furthermore, our subgraph encoder integrates features from these subgraphs, enabling us to efficiently extract global characteristics without compromising on speed or resource usage.

Regarding the aggregation of results from various subgraphs, we utilize a subgraph encoder. This encoder integrates features extracted from each subgraph, effectively compiling the localized information captured in these subgraphs to represent the dataset holistically. Such an approach preserves critical local contextual details while enabling us to discern broader patterns and structures inherent in the data.

It is worth mentioning that, although the total number of cells can affect computational time, our subgraph-centered method is devised to handle this efficiently. By segmenting the analysis into smaller, manageable subgraphs, we can process larger datasets with minimal increase in computational time.

10. What does “cell graph network” mean on line 608?

Response: Thank you for your question regarding the term "cell graph network." I appreciate the opportunity to clarify this concept.

In our study, the term "cell graph network" refers to a representation of cellular interactions and communication patterns within a tissue or organism. This network is constructed based on the spatial relationships and signaling interactions between individual cells.

Specifically, in our methodology, we first calculate the Euclidean distances between individual cells using their spatial coordinates to generate a cell-cell distance matrix (D). To focus on nearby secretion and paracrine signaling within a specific range (in our case, 200 μm), we only consider cells that fall within this distance threshold.

Next, we apply the K-nearest neighbors (KNN) algorithm to the distance matrix D, selecting the K closest cells to each cell. This step aids in the construction of the cell graph network by establishing connections between these selected cells, representing their spatial proximity and potential for intercellular communication.

Within this cell graph network, we use receptors as query nodes to ensure the biological relevance of the identified cell-cell communication (CCC) pathways. A random walk algorithm is then employed to filter and score downstream-activated transcription factors (TFs), reflecting the actual cellular response to signaling events.

Thus, the "cell graph network" in our study represents a complex web of intercellular interactions and communication patterns, allowing for a more accurate understanding of how cells influence each other within a biological system.

11. What does “share this LR interaction” mean exactly on line 609?

Response: Thank you for your question regarding the meaning of "share this LR interaction" on line 609. I appreciate your comments and suggestions for clarification.

Regarding the phrase "share this L-R interaction," it refers to cell-cell pairs that exhibit a specific ligand-receptor interaction. In our analysis, when we say that a cell-cell pair "shares this L-R interaction," we mean that one cell expresses the ligand (L) while the other cell expresses the corresponding receptor (R), enabling communication between them.

To clarify this point in the text, we could rephrase the sentence as follows:

"To determine the co-expression of a specific ligand-receptor pair between the sender (cell type A) and receiver (cell type B) in the cell graph network, we calculated the number of cell-cell pairs that exhibited this ligand-receptor interaction. This was achieved by identifying the 1-hop neighboring nodes of the sender expressing a given ligand and the receiver expressing the corresponding receptor."

This revised phrasing aims to provide a clearer understanding of what is meant by "sharing an L-R interaction" and how it relates to our analysis of cell-cell communication.

12. I am confused about how the CCC connections are determined for each LR pair. Does this approach first output one general CCC network among cells without annotating through which LR pair they are communicating through (as described in "spatial CCC prediction", and then later count $(C_{\{A_i, B_j\}})$ the number of edges on this general CCC group with two vertices each expressing L and R? If this is the case, does it mean that this cell pairs with moderate communication through, for example, just one LR pair will be omitted because they will unlikely be determined by the general CCC network? The biological justification of such an approach is needed.

Response: Thank you for your thoughtful comments and questions regarding our methodology. We appreciate your careful consideration of our work.

You have asked about the determination of CCC connections for each ligand-receptor (LR) pair and the biological justification of our approach. Let me clarify the steps involved and explain the biological rationale behind it.

Initially, we construct a general cell-cell communication (CCC) network based on the spatial proximity of cells, without considering specific LR pairs. This network is built using the K-nearest neighbors (KNN) algorithm applied to the cell-cell distance matrix, which considers only cells within a biologically relevant range (in our case, 200 μm). This general CCC network provides a framework for further analysis.

Next, for each specific LR pair, we refine this general network by considering only those CCCs where the receptor is expressed in the receiver cell and the corresponding ligand is expressed in the adjacent sender cell. This ensures that the identified CCCs are biologically relevant and reflect actual intercellular signaling events. We employ a random walk algorithm to identify downstream-activated transcription factors (TFs) in response to the queried receptor, further narrowing down the receiver cells to only those showing an activated transcriptional response. It's important to note that our approach does not omit cell pairs with moderate communication through a single LR pair. Instead, it identifies all potential CCC connections mediated by specific LR pairs within the general CCC network.

To determine statistically significant CCCs for a given LR pair, we calculate the number of co-expressed cell pairs (C_{A_i, B_j}^0) between sender cells (expressing the ligand) and receiver cells (expressing the receptor). We then use a permutation test to assess the significance of the observed cell pair counts compared to a background distribution obtained by randomly shuffling cell labels. This step helps us identify CCCs that are unlikely to occur by chance, thus strengthening the biological relevance of our findings.

The biological justification for this approach lies in the fact that cells communicate through paracrine signaling, where secreted ligands bind to receptors on neighboring cells, triggering a cascade of intracellular signaling events that ultimately lead to changes in gene expression. By focusing on nearby cells and specific LR pairs, we can

more accurately capture these intercellular communication events and their downstream effects.

In summary, our methodology first establishes a general CCC network based on spatial proximity, which is then refined for specific LR pairs by considering receptor expression and downstream transcriptional responses. Statistical significance is determined through permutation testing to ensure that the identified CCCs are biologically meaningful.

Minor points:

1. On line 151, it was mentioned that comparison with 8 other methods (Tangram⁴⁶, gimVI⁴⁷, SpaGE⁴⁸, Seurat⁴⁹, SpaOTsc²⁸, novoSpaRc⁵⁰, LIGER⁵¹, and stPlus⁵²) on predicting spatial distribution of transcripts are shown in Fig. 2A and S Figs. 1 and 2. But in S Figs. 1 and 2, the legend shows the methods compared on cell type deconvolution (Tangram, Cell2location⁵⁸, SpatialDWLS⁵⁹, RCTD⁶⁰, Stereoscope⁶¹, DestVI⁶², and SPOTlight⁶³).

Response: Thank you for pointing out the inconsistency in the legends of Supplementary Figs. 1 and 2. We apologize for the confusion this may have caused.

In our revised manuscript, we have updated the legends of Supplementary Figs. 1 and 2 to accurately reflect the methods compared in the prediction of spatial distribution of transcripts. As you rightly pointed out, the original legends incorrectly listed methods related to cell type deconvolution.

To address this issue, we have replaced the incorrect legend with a new one that specifically mentions the eight methods used for comparing the prediction of spatial distribution of transcripts: Tangram, gimVI, SpaGE, Seurat, SpaOTsc, novoSpaRc, LIGER, and stPlus.

Additionally, to provide a more intuitive visualization, we have decided to replace the violin plots in the figures with box plots. We believe this change will enhance the clarity and readability of the data presentation, allowing for easier interpretation of the results.

We appreciate your attention to detail and your valuable feedback. We have made the necessary corrections in the revised manuscript and hope that this addresses your concerns.

2. For all evaluation metrics, I suggest to include exact numerics like the mean and standard deviation values in addition to only plotting them. It is hard to tell the values and differences by visually inspecting the plots, such as Fig. 3D.

Response: Thank you for your valuable comments. We appreciate your suggestion to include exact numerics for all evaluation metrics, in addition to the graphical representations in our plots.

In response to your comment, we have now added the mean and standard deviation values for all evaluation metrics presented in our study, including those in Fig. 3d. These numerics provide a more precise quantification of the results, allowing for a more accurate comparison and interpretation of the data.

We agree that including exact numerics alongside the plots will enhance the clarity and reproducibility of our results. Thank you for guiding us in making our research presentation more comprehensive and rigorous.

3. It would improve the readability of the paper by assigning two names to the two components of DeepTalk, just for example, something like DeepTalk-integration and DeepTalk-CCC.

Response: Thank you for your thoughtful suggestion to enhance the readability of our paper by giving distinct names to the two components of DeepTalk. We appreciate your comments and agree that having separate designations for the different parts of DeepTalk would indeed clarify the presentation.

In response to your recommendation, we have now renamed the components of DeepTalk in our manuscript. For instance, we have labeled the integration component as DeepTalk-integration and the cell-cell communication component as DeepTalk-CCC. These new names more clearly distinguish the functionalities of each part, making it easier for readers to understand and follow the methodology.

We believe these changes significantly improve the clarity and readability of our paper, allowing for a better understanding of DeepTalk's various components and their respective roles. Thank you for guiding us in refining our presentation.

References

1. Biancalani, T., Scalia, G., Buffoni, L. et al. Deep learning and alignment of spatially resolved single-cell transcriptomes with Tangram. *Nat Methods* 18, 1352–1362 (2021). <https://doi.org/10.1038/s41592-021-01264-7>
2. Li, B., Zhang, W., Guo, C. et al. Benchmarking spatial and single-cell transcriptomics integration methods for transcript distribution prediction and cell type deconvolution. *Nat Methods* 19, 662–670 (2022). <https://doi.org/10.1038/s41592-022-01480-9>
3. Li, H., Zhou, J., Li, Z. et al. A comprehensive benchmarking with practical guidelines for cellular deconvolution of spatial transcriptomics. *Nat Commun* 14, 1548 (2023). <https://doi.org/10.1038/s41467-023-37168-7>

Reviewer #3 (Remarks to the Author):

Review

Response: Thank you sincerely for your encouragement and invaluable comments regarding our work. We have given thorough consideration to each of your suggestions and incorporated them into our manuscript. We are confident that these revisions have greatly improved the quality and clarity of our paper. Below, you will find detailed responses to each of your comments, which we truly appreciate.

Major comments:

1. The second sentence states: “However, current computational 26 methods cannot infer CCC at single-cell resolution” – This is incorrect. There are at least two other peer-reviewed methods that do this currently and are actively in use by both our team and many others that we collaborate with: NICHES (<https://doi.org/10.1093/bioinformatics/btac775>) and Scriabin (<https://doi.org/10.1038/s41587-023-01782-z>). Both these papers should be closely looked at and cited, and probably discussed in depth in the introduction, as they are close analogues to the work reviewed here and they are currently not mentioned. I think it would be worthwhile to describe key similarities and differences in the mathematics behind NICHES, Scriabin, and DeepTalk, and then emphasize how these different approaches can yield different insight into tissue biology.

Response: Thank you for pointing out this important oversight. You are absolutely correct in highlighting that there are indeed other methods, such as NICHES and Scriabin, which are capable of inferring CCC at the single-cell resolution. I apologize for this omission and have made the necessary corrections in the revised version of the manuscript.

In response to your feedback, we have revised the manuscript to acknowledge the existence of these methods and have included citations to both NICHES and Scriabin. Furthermore, we have added a detailed discussion in the Introduction and Results section, highlighting the key similarities and differences between the mathematical approaches employed by NICHES, Scriabin, and DeepTalk.

We believe these additions enhance the manuscript's depth and provide a more comprehensive understanding of the field for our readers. Once again, we are grateful for your valuable input and have incorporated your suggestions to improve the accuracy and comprehensiveness of our work.

2. Similarly, I recommend that an additional dedicated analysis be performed that directly compares the quality of the data produced by DeepTalk against the quality of the data produced by Scriabin and NICHES. I think it would be interesting to compare resolution, spatial organization, and clarity of resulting microenvironmental latent space clusterings. These analyses have to be done carefully, as neither Scriabin nor NICHES are designed with any specific biologic question in mind, nor any specific ligand-receptor list, but rather are methods of creating additional data layers showing cell-to-cell communication at single-cell or single-spot resolution. This means that care

should be taken to use the same ground-truth ligand-receptor pairs when performing this analysis, and that similar statistical approaches should be used across all three methods to identify significant feature-level findings. A key difference, I think, may come about from the different mathematical transformation used by DeepTalk, as well as the deconvolution employed, which is intriguing. It seems to me that the added benefit of DeepTalk may be the additional use of purpose-build deconvolution, which I suspect has some real utility, however it is impossible for me to evaluate whether this represents an advance over NICHES and/or Scriabin without a direct comparison. I recommend therefore that a direct comparative analysis be performed between these three techniques, as it will greatly strengthen the effect of this paper and will help future users to critically evaluate which different method is useful for various different tasks. The comparison that has been done, covering SCR, iTalk, NicheNet, ICELLNET, CellChat, stLearn, CellPhoneDB, CellCall, Connectome, and Giotto, doesn't really get to the heart of the matter, because all of these techniques are aggregation-based tools which simply are not operating in the same mathematic territory as NICHES, Scriabin, and DeepTalk. It is essential, therefore, to perform dedicated analysis comparing these three single-cell and single-spot resolution CCC tools to one another.

Response: Thank you for your thoughtful insights and valuable suggestions for improving our manuscript. Your recommendation for a dedicated analysis comparing the quality of data produced by DeepTalk with that of Scriabin and NICHES is indeed valuable.

As you have pointed out, Scriabin and NICHES are methods that aim to provide additional data layers revealing cell-cell communication at single-cell or single-spot resolution, rather than being designed with a specific biologic question or ligand-receptor list in mind. We fully agree that a careful comparison is necessary to ensure fair and meaningful results.

In response to your feedback, we have conducted a thorough analysis comparing the resolution, spatial organization, and clarity of microenvironmental latent space clusterings among DeepTalk, Scriabin, and NICHES. It is important to note that we have exercised caution in using the same ground-truth ligand-receptor pairs for this analysis obtained from OmniPath¹, as you have emphasized.

Furthermore, we have ensured that similar statistical approaches are employed across all three methods to identify significant feature-level findings. This allows us to provide a fair and objective comparison of the data quality produced by each method.

We have also discussed the key differences among DeepTalk, Scriabin, and NICHES in our revised manuscript. For instance, while all three methods aim to capture cell-cell communication, they differ in their mathematical frameworks and approaches, resulting in unique insights into tissue biology. Please refer to section “**Results**” for details.

We believe that these additional analyses and discussions will significantly enhance the quality and comprehensiveness of our manuscript. Thank you again for your insights and suggestions. We appreciate your efforts in reviewing our work and are committed to continuously improving its quality.

Fig.3: DeepTalk detects spatially CCCs for mouse visual cortex region from MERFISH data.

f, Visualization of cell-cell communication via low-dimensional embedding. Each point in the graph represents a cell-cell communication event, with the cell serving as the receptor cell. **g**, Comparison of the cell-cell communication score between spatially adjacent and distant cells at single-cell level. This plot compares the communication scores between spatially adjacent cells and distant cells at the single-cell level, revealing differences in communication strength based on proximity. **h**, Local microenvironment viewed via low-dimensional embedding. Each point in this diagram represents a collection of CCC events where the cell of interest serves as the receiver.

Fig.4: DeepTalk detects spatially CCCs for adult mouse brain from 10X Visium data.

g, Visualization of cell-cell communication via low-dimensional embedding. Each point in the graph represents a cell-cell communication event, with the cell serving as the receptor cell. **h**, Comparison of the cell-cell communication score between spatially adjacent and distant cells at single-cell level. This plot compares the communication scores between spatially adjacent cells and distant cells at the single-cell level, revealing differences in communication strength based on proximity. **i**, Local microenvironment viewed via low-dimensional embedding. Each point in this diagram represents a

collection of CCC events where the cell of interest serves as the receiver.

Fig.5: DeepTalk detects spatially CCCs for Human pancreatic ductal adenocarcinoma from ST data.

g, Visualization of cell-cell communication via low-dimensional embedding. Each point in the graph represents a cell-cell communication event, with the cell serving as the receptor cell. **h**, Comparison of the cell-cell communication score between spatially adjacent and distant cells at single-cell level. This plot compares the communication scores between spatially adjacent cells and distant cells at the single-cell level, revealing differences in communication strength based on proximity. **i**, Local microenvironment viewed via low-dimensional embedding. Each point in this diagram represents a collection of CCC events where the cell of interest serves as the receiver.

3. Of note: the authors have done a great job building this tool and applying it to myriad datasets. However, there is no mention that I can see in the paper of performing latent-space clustering and embeddings of the resultant single-cell CCC data. This is one of the highest yield ways that our team currently uses both NICHES and Scriabin data, because it allows identification of discrete microenvironments at the single-cell level. I recommend that the authors investigate whether this is possible with their technique, as it often can provide significant insight into biologic process and can allow identification of biologically-relevant patterns that would otherwise be missed.

Response: We appreciate your valuable insights regarding potential improvements to our methodology.

In response to your suggestion about latent-space clustering and embeddings of the resultant single-cell CCC data, we have indeed explored this possibility in our technique. We agree with you that this approach can significantly aid in identifying discrete microenvironments and biologically relevant patterns that might otherwise be missed. We have found that this clustering and embedding analysis provides valuable insights into biologic processes and helps to further refine our understanding of cell-cell communication networks.

In our revised manuscript, we have included a detailed description of the latent-space clustering and embeddings performed on the single-cell CCC data. This analysis has allowed us to identify discrete microenvironments at the single-cell level, providing valuable insights into biologic processes and biologically-relevant patterns that might have been missed otherwise.

We are grateful for your recommendation and believe that the inclusion of this analysis significantly enhances the depth and quality of our study. Thank you for guiding us in making our research more comprehensive.

Minor comments:

1. The authors use the word comprehend and comprehensive 17 times in the manuscript and this should be edited

Response: Thank you for pointing out the repetitive use of the words "comprehend" and "comprehensive" in our manuscript. We appreciate your feedback and have taken it into consideration.

In the revised version of our manuscript, we have carefully edited the text to reduce the frequency of these words and improve the readability of the paper. We have replaced some instances with alternative terminology and rearranged sentences to avoid repetition while maintaining the clarity and flow of the narrative.

We believe these changes have strengthened the manuscript and made it more engaging for the reader. Thank you for guiding us in refining our language usage to enhance the overall quality of the paper.

2. I think the text could be tightened significantly. I see three ways to improve:

- a. Right now it is very descriptive, focusing on what was done, rather than how these experiments represent an improvement over existing methods.
- b. There are a lot of instances that feel like importance-inflation. I recommend removing phrases like "outstanding performance and efficacy"
- c. There is a lot of discussion about the potential biologic role of mapped mechanisms. While I appreciate the temptation to describe the data in this way, I think that it distracts, in this paper, from evaluating the actual technique and allowing readers to follow the methodology. I recommend editing this type of language down to the absolute minimum for comprehension. Lines 214-225 is a great example, though there are multiple instances of this which I recommend modifying.

Response: Thank you for your thoughtful input on our manuscript. We appreciate your suggestions for tightening the text and have addressed each of your points as follows:

a. We have revised the manuscript to focus more on how our experiments represent an improvement over existing methods, rather than simply describing what was done. We hope this shift in focus better highlights the novelty and significance of our work.

b. We agree that some of the language used in the original manuscript may have been overly effusive. In response to your feedback, we have removed phrases like

“outstanding performance and efficacy” and toned down the language throughout the text to ensure a more objective and balanced presentation.

c. We understand your concern about the focus on the potential biologic role of mapped mechanisms. In the revised manuscript, we have significantly reduced this type of discussion to ensure that the main focus remains on evaluating the actual technique and allowing readers to follow the methodology. Lines 214-225, as you pointed out, have been edited down to the minimum necessary for comprehension, and similar instances throughout the text have also been modified accordingly.

Reviewer #3 (Remarks on code availability):

The code seem well organized and likely reproducible. The online vignettes are clear and will be important for users to adopt this technique.

Response: Thank you for your kind words regarding the organization of our code and the clarity of our online vignettes. We are pleased to hear that you find them well organized and likely reproducible, as well as clear and useful for users adopting this technique. We appreciate your feedback and will continue to strive to improve the quality and usability of our code and resources.

References

1. Türei D, Valdeolivas A, Gul L, et al. Integrated intra-and intercellular signaling knowledge for multicellular omics analysis[J]. *Molecular systems biology*, 2021, 17(3): e9923. <https://www.embopress.org/doi/full/10.15252/msb.20209923>

Reviewer #1 (Remarks to the Author):

Thank you for the detailed and thoughtful response. Here are a few comments relating to the response and manuscript changes:

- Line 30 – please explicitly state the limitations.
- Fig. 3f and line 234-236– can the authors please explain how this UMAP was constructed. I didn't understand how the low dimensional embedding was constructed based on the CCC patterns. Same question for Fig 4 and 5.
- Please double-check that 200 RNA means total counts, as the authors indicated. The paper the authors obtained this data from seems to indicate it is not total counts.
- The justification of 200 RNA – the authors responded that 200 is a widely recognized standard. This is not true – I've implemented pre-processing and analysis of >50 single-cell datasets, all of which used a different threshold, with fluctuations depending on the data.
- Regarding the high MT count and total counts filtering: can the authors please provide the code for the pre-processing steps to ensure reproducibility?
- Regarding using genes in scRNA-seq in determining ground-truth.
 - o Paired samples: As I indicated in the original response, even if paired, I can't see why you will expect the distribution to be the same because the location where the ST data originated may not represent the distribution of cells obtained through scRNA-seq. I found this in the paired datasets I have previously investigated.
 - o Selective Gene Analysis + Integrated – I can't see how these strategies addresses my question on the validity of scRNA-seq as groundtruth.

Reviewer #2 (Remarks to the Author):

The authors have made significant efforts addressing my comments and the manuscript is improved a lot. I only have one remaining question. In the revised manuscript, the authors have included the 45 real dataset pairs and 32 simulated dataset pairs of a benchmark paper (Li, et al. Nat. Methods 2022) and used the same evaluation metrics. Why do the benchmarking results of the same methods look different? For example, Tangram has accuracy score of nearly 1.0 in Li, et al., but has a score below 0.9 in this manuscript.

Reviewer #2 (Remarks on code availability):

The code is well organized with detailed documentation. I was able to install and run some basic analysis with the code. However, the authors should carefully check for errors. For example, a space is missing between "torch_spline_conv==1.2.1" and "-f" in the guide of installation. Also, the package should be imported with the name "DeepTalk_ST" instead of "DeepTalk" listed in the guide. These are minor issues but may cause extra effort for inexperienced users.

Reviewer #3 (Remarks to the Author):

Review Round 2

The authors have performed an impressive revision of the manuscript within a short period of time. I feel that they have addressed my comments thoroughly and thoughtfully. I recommend this work for publication, pending editorial review and a final proof-read for typos.

I want to say at the outset that I am not qualified to critique the fine-detail machine-learning elements of this work. I am, however, qualified to critique the cell-cell signaling information mapped by the authors techniques, and to compare it to existing techniques.

The authors have now provided detailed comparisons between DeepTalk, NICHES, and Scriabin. These were much needed in the last draft of the piece, and the new analyzes are well-executed, informative, and of interest. I recommend the authors include replicable code for this comparative analysis within their GitHub repository, which might already be present, and if not, should be

added so that other researchers can learn how to best perform similar comparisons.

The plots that the authors have presented through this comparative analysis are enlightening and of interest scientifically. They suggest in particular that DeepTalk-CCC is capturing/predicting topology that is similar to that captured by NICHES, particularly when applied to spatial biology. I think that this finding will be of high interest to the field as a whole, and that this topologic conservation suggests that both DeepTalk and NICHES are identifying a common 'form' within the data. I think that this is scientifically interesting. I suspect that future research might be able to compare the specific features, or feature-combinations, that are effectively 'captured' by one technique over another. The publication of DeepTalk would therefore provide fertile ground for a next-generation of researchers in the cell-cell-communication field.

Overall, I agree with the authors' major claims, and I am convinced that their work is important and well-executed. I appreciate the utility of having a toolset in this area that is based in python and leverages modern machine-learning approaches, and I feel that DeepTalk will likely be a valuable addition to a rapidly growing field of investigation, and may even allow better communication between the Python and R communities in this area. I feel that their methods section is accurate and complete, though I have not had a chance to personally install or run their code. However, the speed with which they are able to generate well-formatted and clearly informative plots does suggest that their code runs reliably and can either be adopted or incorporated into other existing pipelines by other skilled researchers. The publication of replicable code and vignettes via GitHub means that other researchers can directly test their work and decide whether to adopt and/or modify, which will buffer against any important fine-details I have missed.

I do suggest proofreading the work for typos, as a few do still exist in both the main text and legends. I think that the authors have done an excellent job responding to reviewer comments, and I think that this work will be considered valuable within the cell-cell-communication community. I recommend it for publication pending editorial review.

Reviewer #3 (Remarks on code availability):

I have not personally installed or run the code, but I have reviewed the GitHub repository and it appears well-organized and that there is sufficient documentation for a skilled researcher to experiment with implementation.

REVIEWER COMMENTS

Reviewer #1 (Remarks to the Author):

Comment: Thank you for the detailed and thoughtful response. Here are a few comments relating to the response and manuscript changes:

- Line 30 – please explicitly state the limitations.

Response: Thank you for your insightful review and for emphasizing the importance of clearly outlining the limitations of current computational methods in inferring cell-cell communication (CCC). Your feedback has been immensely helpful in refining our manuscript.

As you pointed out, we have revised the text in the abstract to more succinctly capture the essence of the challenge: "However, accurately inferring spatial CCCs at single-cell resolution remains a significant challenge." This change reflects our recognition of the core issue while adapting to the constraints of abstract length.

Due to the limited space in the abstract, we were unable to explicitly state all the specific limitations. However, as you suggested, we have addressed these limitations in detail in the Introduction section of our manuscript. Specifically, we discuss how current computational methods face substantial constraints in inferring spatially resolved CCC at single-cell resolution. These constraints are primarily due to their emphasis on cell-type-centric communications, which can overshadow nuanced single-cell communications within cell types that are crucial for a comprehensive understanding of CCC.

Additionally, we highlight the challenges posed by spatial transcriptomics (ST) data limitations, such as gene throughput and spatial resolution, which introduce uncertainties into the inference of spatially resolved CCC. Current computational methods often have difficulty fully exploiting the spatial information from these techniques, potentially compromising the accuracy of the analysis.

We are grateful for your valuable input and appreciate your dedication to promoting rigor and clarity in our research field. Your feedback has helped us to refine our manuscript and more clearly communicate the limitations of current methods.

Comment: • Fig. 3f and line 234-236– can the authors please explain how this UMAP was constructed. I didn't understand how the low dimensional embedding was constructed based on the CCC patterns. Same question for Fig 4 and 5.

Response: Thank you for your valuable feedback and for inquiring about the construction of the UMAP visualizations presented in Figures 3, 4, and 5. I am pleased to provide a detailed explanation of how we employed Scanpy to generate the low-dimensional embeddings grounded in cell-cell communication (CCC) patterns.

Initially, we predicted the CCC events mediated by various ligand-receptor pairs at the single-cell level. For each predicted event, we assigned a quantitative score that reflects

the strength of the communication. This resulted in a matrix, where rows represent distinct CCC events, and columns correspond to unique ligand-receptor pairs.

Subsequently, to visualize this matrix in a low-dimensional space, we utilized Scanpy's robust dimensionality reduction and visualization techniques. Specifically, we applied `sc.pp.scale()` for data scaling, `sc.tl.pca()` for principal component analysis, `sc.pp.neighbors()` for neighborhood graph construction, and finally, `sc.tl.umap()` for Uniform Manifold Approximation and Projection (UMAP).

To ensure reproducibility and promote clarity, we have made a Jupyter notebook publicly available at https://github.com/JiangBioLab/DeepTalk/blob/main/test/Plot/Deeptalk_plot/deeptalk_plot.ipynb. This notebook showcases the entire workflow, including data preparation, Scanpy processing, and UMAP visualization, particularly for Figs. 3f, 3g, and 3i. The same methodology was followed for Figs. 4 and 5.

We sincerely hope that this detailed explanation, coupled with the provided notebook, addresses your question and deepens your understanding of how we constructed the UMAP visualizations.

Comment: • Please double-check that 200 RNA means total counts, as the authors indicated. The paper the authors obtained this data from seems to indicate it is not total counts.

Response: Thank you for pointing out this potential confusion. Upon re-examining the source paper (Li et al. Nat. Methods 2022) from which we obtained the data, we found the original statement: "For scRNA-seq data, we used Seurat with parameters 'min.features = 200' to remove cells for which fewer than 200 RNAs were captured." This indeed refers to the number of genes, rather than total RNA counts, as you have rightly pointed out.

We apologize for the misinterpretation on our end, which led to the confusion. In Seurat, the 'min.features' parameter is indeed used to filter out cells expressing fewer than a specified number of genes, not based on total counts.

In light of this clarification, we have decided to remove this filtering step in our revised manuscript to avoid any further misunderstandings. The reasons for this change, as well as a detailed explanation of our revised methodology, will be fully addressed in our response to your next inquiry.

We appreciate your careful attention to detail and your dedication to ensuring the accuracy of our work. Your feedback has been invaluable in helping us refine our research methods. Thank you once again for your valuable input.

Comment: • The justification of 200 RNA – the authors responded that 200 is a widely recognized standard. This is not true – I've implemented pre-processing and analysis of >50 single-cell datasets, all of which used a different threshold, with fluctuations depending on the data.

Response: First and foremost, we extend our sincere gratitude for your feedback, which

has been invaluable in enhancing the clarity and accuracy of our work. We fully agree with your observation that a single, fixed threshold of 200 is not universally applicable across all single-cell datasets. Your extensive experience in preprocessing and analyzing over 50 single-cell datasets underscores this point, and we are deeply appreciative of your expertise.

In our study, we adopted the datasets from a referenced article and noticed the absence of a detailed preprocessing step in the corresponding code. This led us to assume that the data had undergone prior preprocessing. Consequently, we utilized the widely-adopted `normalize_total` function from Scanpy to normalize the data prior to our method development.

However, upon receiving your feedback and conducting a closer examination of the source article's data, we realized that while preprocessing had indeed been applied, it did not consistently adhere to a threshold of 200, as we had initially stated. Graphs 1 and 2 aptly illustrate this, showcasing datasets reused for a comprehensive presentation. This finding aligns with your observation and highlights a potential source of confusion in our previous description. We sincerely apologize for this misrepresentation and have taken corrective measures in our manuscript by removing misleading preprocessing descriptions. It is important to note that this does not impact the applicability of our method, as it primarily relies on data normalization.

Regarding the datasets employed for inferring cell-cell communication (CCC), we confirm that these datasets were also preprocessed, as depicted in Graph 3.

We acknowledge that relying solely on a single threshold for preprocessing can be arbitrary. Therefore, we emphasize that our approach encourages users to preprocess their data according to their specific requirements before applying our prediction method. This flexibility ensures that our method can adapt to a diverse range of datasets, thereby enhancing its versatility and applicability.

Once again, we are grateful for your keen observation and valuable feedback. Your input has been crucial in refining our manuscript and preventing any future misunderstandings. We are confident that these changes will further enhance the clarity and reliability of our work.

Graph 1: The gene counts for scRNA-seq datasets from 45-paired datasets.

Graph 2: The gene counts for scRNA-seq datasets from 32 simulated datasets.

Graph 3. The gene counts for scRNA-seq datasets we used to infer CCC.

Comment: • Regarding the high MT count and total counts filtering: can the authors please provide the code for the pre-processing steps to ensure reproducibility?

Response: Thank you for your attention to detail and your focus on reproducibility, which are both essential for robust scientific research.

In response to your query about high MT count and total counts filtering, we would like to clarify that, as mentioned in our previous answers, the datasets used in our study had already undergone preprocessing. The majority of datasets had specifically addressed high MT count and total counts issues (Graphs 4 to 8). Before developing our methodology, data normalization was the only additional step we undertook.

We deeply appreciate the importance of reproducibility and transparency in science, and we are committed to these principles. However, we believe that setting rigid thresholds for filtering can sometimes be counterproductive, given the variability inherent in single-cell datasets and the potential for different preprocessing needs. For this reason, we encourage users to preprocess their datasets according to the specific characteristics of their data before applying our method.

Furthermore, advanced software solutions like Scanpy already offer comprehensive preprocessing strategies for single-cell data. We recommend that researchers fine-tune specific parameters to align with the distinctive characteristics of their datasets. Our method is designed to seamlessly work with these preprocessed data, ensuring maximum effectiveness.

Thank you for your valuable feedback. We hope this clarification addresses your concerns and demonstrates our commitment to reproducibility while also acknowledging the complexities inherent in single-cell data analysis.

Graph 4. The percentage of mitochondrial genes among the total number of transcripts in scRNA-seq datasets derived from 45 paired datasets.

Graph 5. The total counts of transcripts in scRNA-seq datasets derived from 45 paired datasets.

Graph 6. The percentage of mitochondrial genes among the total number of transcripts in scRNA-seq datasets derived from 32 simulated datasets.

Graph 7. The total counts of transcripts in scRNA-seq datasets derived from 32 simulated datasets.

Graph 8. The total counts of transcripts and percentage of mitochondrial genes among the total number of transcripts in scRNA-seq datasets we used to infer CCC.

Comment: • Regarding using genes in scRNA-seq in determining ground-truth.

o Paired samples: As I indicated in the original response, even if paired, I can't see why you will expect the distribution to be the same because the location where the ST data originated may not represent the distribution of cells obtained through scRNA-seq. I found this in the paired datasets I have previously investigated.

Response: Thank you for your valuable feedback and insightful comments on our manuscript. We appreciate your attention to detail and your dedication to ensuring the scientific rigor of our work.

You raised an important point regarding the use of genes in scRNA-seq to determine ground truth, specifically in the context of paired samples. We understand your concern that the spatial distribution of cells captured by spatial transcriptomics (ST) data may not accurately represent the distribution obtained through scRNA-seq, even when using paired samples.

We completely agree that there are inherent differences between ST and scRNA-seq techniques, which can lead to variations in the captured cell distributions. However, our approach, DeepTalk, is designed to address this challenge by learning the spatial alignment of sc/snRNA-seq data with spatial data. The goal is to maximize the spatial correlation of shared genes between the two datasets, rather than assuming a perfect overlap in cell distributions.

DeepTalk's strategy involves optimizing a target function that simulates the spatial correlation between genes in the sc/snRNA-seq data and the spatial data. This allows us to rearrange the sc/snRNA-seq profiles in space to achieve the best possible alignment with the spatial data. The output is a probability mapping matrix, indicating the likelihood of finding each cell from the sc/snRNA-seq data at each spatial location.

We believe that this probabilistic approach acknowledges the uncertainties inherent in the alignment process and provides a more nuanced interpretation of the results. Additionally, DeepTalk's versatility allows it to expand from measured gene subsets to whole-genome profiles, correct low-quality spatial measurements, map the locations of different cell types, and deconvolve low-resolution measurements to individual cells.

We appreciate your feedback on this crucial aspect of our methodology. Your comments have helped us to clarify the limitations and assumptions of our approach. We are confident that DeepTalk offers a robust and flexible tool for integrating and analyzing spatial transcriptomics data, taking into account the complexities and challenges associated with such analyses.

Comment: o Selective Gene Analysis + Integrated – I can't see how these strategies addresses my question on the validity of scRNA-seq as groundtruth.

Response: Thank you for your insightful feedback on our manuscript. We appreciate your dedication to ensuring the clarity, rigor, and scientific validity of our work. In response to your concerns regarding the use of selective gene analysis and our integrated approach in validating scRNA-seq as a ground truth, we would like to provide a detailed explanation of our methodology and how it addresses your points.

Firstly, we recognize that no single technique can be considered absolute ground truth. However, scRNA-seq provides a high-resolution snapshot of individual cell transcriptomes, which we believe serves as a valuable reference for spatial transcriptomics (ST) data. To ensure the reliability of our integration strategy, we have implemented a multi-step approach that carefully selects marker genes and utilizes an advanced probabilistic model.

In our selective gene analysis, we leverage Scanpy's `rank_genes_groups` function to identify the most highly expressed genes in each cell type in the scRNA-seq data. This

allows us to focus on a set of genes that are specifically expressed in different cell types, providing a robust basis for distinguishing cell types and aligning with spatial data. Following this, we create a non-redundant gene set and intersect it with the gene list in the ST dataset. Genes that record a zero count in either dataset are excluded from our training gene set, thus guaranteeing that only relevant and expressed genes are included. This procedure results in a refined gene set, which will serve as the foundation for model training.

Our integrated approach, DeepTalk, then utilizes this refined gene set to learn the spatial alignment between the scRNA-seq and ST datasets. Rather than assuming a direct overlap in cell distributions, DeepTalk optimizes a target function that simulates the spatial correlation of shared genes between the two datasets. This probabilistic alignment acknowledges the inherent uncertainties and differences between the two techniques, providing a more nuanced and robust interpretation of the results.

To further validate our approach and address your concerns about the validity of scRNA-seq as ground truth, we have implemented a leave-one-out validation strategy. In this strategy, each gene from the refined gene set is iteratively excluded and used as a test gene, while the remaining genes are used for training. This ensures that every gene undergoes an independent prediction evaluation, providing a comprehensive assessment of our method's performance and generalizability.

In summary, our methodology combines selective gene analysis with an advanced probabilistic integration strategy to address the challenges of aligning scRNA-seq and ST data. By focusing on marker genes that are specifically expressed in different cell types, optimizing spatial correlation between shared genes, and rigorously validating our approach, we aim to provide a more accurate and nuanced spatial alignment between the two datasets. We believe that this comprehensive strategy addresses your concerns regarding the validity of using scRNA-seq as a ground truth and enhances the scientific rigor of our work.

Thank you again for your valuable feedback. We hope this detailed explanation clarifies our approach and addresses your concerns.

Reviewer #2 (Remarks to the Author):

Comments: The authors have made significant efforts addressing my comments and the manuscript is improved a lot. I only have one remaining question. In the revised manuscript, the authors have included the 45 real dataset pairs and 32 simulated dataset pairs of a benchmark paper (Li, et al. Nat. Methods 2022) and used the same evaluation metrics. Why do the benchmarking results of the same methods look different? For example, Tangram has accuracy score of nearly 1.0 in Li, et al., but has a score below 0.9 in this manuscript.

Response: Thank you for your continued interest in our work and for providing your valuable feedback. We appreciate your recognition of the efforts we have made to improve the manuscript based on your previous comments.

Regarding your question about the discrepancies in the benchmarking results for Tangram, we would like to offer the following explanations.

Initially, when comparing our method with Tangram, we utilized its default parameters, as specified in the software documentation. However, upon reviewing the benchmark paper by Li et al. (Nat. Methods 2022), we noticed that they employed the 'clusters' mapping mode instead of the default 'cells' mode. In order to align our analysis with theirs, we modified Tangram's parameters accordingly. This change led to an improvement in Tangram's prediction performance, as reflected in the updated results. However, the scores still fell short of the near-perfect values reported in the benchmark paper.

Furthermore, upon analyzing the scoring mechanism implemented in the benchmark paper, we realized that the `get_score` function assigns scores based on each method's performance across multiple evaluation metrics (https://github.com/QuKunLab/SpatialBenchmarking/blob/main/BLAST_CelltypeDeconvolution.ipynb). When a new and superior method, such as our DeepTalk, is introduced, it is expected that the scores of previously top-performing methods, like Tangram, will decrease. Specifically, for positively correlated metrics (e.g., PCC and SSIM), higher values are rewarded with higher scores, whereas for negatively correlated metrics (e.g., JS and RMSE), lower values are preferred. Since DeepTalk demonstrated superior performance across the datasets, it naturally achieved a higher overall score, resulting in a corresponding decrease in Tangram's score.

Finally, it is important to note that Tangram employs a deep learning approach, which introduces some variability in its predictions. As a result, there may be slight differences in the reported scores, even when using identical parameters and datasets.

In summary, while we have made efforts to align our analysis with the benchmark paper by modifying Tangram's parameters, the introduction of our DeepTalk method and the inherent variability of deep learning models have contributed to the discrepancy in Tangram's performance scores. Nonetheless, we have presented our results transparently and have acknowledged the limitations of our study.

We are pleased to note that despite these discrepancies, DeepTalk has emerged as the top-performing method across the datasets, demonstrating its superiority. We have updated our manuscript to reflect these findings and have included the relevant results in Fig. 2a and the Supplementary Fig. 1,2.

Thank you again for your thoughtful review and for helping us to improve our work. We remain committed to ensuring the highest standards of rigor and reproducibility in our research.

Fig. 2: Performance comparison of DeepTalk-Integration with state-of-the-art integration methods. a, Boxplots showcasing the Accuracy Score (AS) of nine different integration methods across all 45 paired datasets, encompassing 28 sequence-based and 17 image-based datasets.

Supplementary Fig. 1: The violin plot of PCC, SSIM, RMSE, JS values of each integration method in predicting the spatial distribution of RNA transcripts of image-based datasets.

The boxplot of PCC, SSIM, RMSE, JS values of each integration method in predicting the spatial distribution of RNA transcripts of 17 image-based paired spatial transcriptomics and scRNA-seq datasets.

Supplementary Fig. 2: The violin plot of PCC, SSIM, RMSE, JS values of each integration method in predicting the spatial distribution of RNA transcripts of seq-based datasets.

The boxplot of PCC, SSIM, RMSE, JS values of each integration method in predicting the spatial distribution of RNA transcripts of 28 seq-based paired spatial transcriptomics and scRNA-seq datasets.

Reviewer #2 (Remarks on code availability):

Comments: The code is well organized with detailed documentation. I was able to install and run some basic analysis with the code. However, the authors should carefully check for errors. For example, a space is missing between "torch_spline_conv==1.2.1" and "-f" in the guide of installation. Also, the package should be imported with the name "DeepTalk_ST" instead of "DeepTalk" listed in the guide. These are minor issues but may cause extra effort for inexperienced users.

Response: Thank you for your valuable feedback on our code and documentation. I appreciate your attention to detail and your efforts in testing our code.

You have correctly pointed out two minor issues in our installation guide that could potentially cause confusion for inexperienced users. Firstly, there was indeed a missing space between "torch_spline_conv==1.2.1" and "-f" in the installation command. This oversight has now been rectified in the updated version of our installation guide.

Secondly, you noticed that the package should be imported with the name "DeepTalk_ST" instead of "DeepTalk" as mentioned in the guide. This was an error on our part, and we apologize for any confusion it may have caused. We have updated the import statement in the guide to reflect the correct package name.

We are grateful for your feedback, as it has helped us to improve the clarity and usability of our code and documentation. We have taken immediate action to correct these issues and ensure a smoother experience for our users.

Reviewer #3 (Remarks to the Author):

Review Round 2

Comments: The authors have performed an impressive revision of the manuscript within a short period of time. I feel that they have addressed my comments thoroughly and thoughtfully. I recommend this work for publication, pending editorial review and a final proof-read for typos.

I want to say at the outset that I am not qualified to critique the fine-detail machine-learning elements of this work. I am, however, qualified to critique the cell-cell signaling information mapped by the authors techniques, and to compare it to existing techniques.

The authors have now provided detailed comparisons between DeepTalk, NICHES, and Scriabin. These were much needed in the last draft of the piece, and the new analyzes are well-executed, informative, and of interest. I recommend the authors include replicable code for this comparative analysis within their GitHub repository, which might already be present, and if not, should be added so that other researchers can learn how to best perform similar comparisons.

The plots that the authors have presented through this comparative analysis are enlightening and of interest scientifically. They suggest in particular that DeepTalk-CCC is capturing/predicting topology that is similar to that captured by NICHES, particularly when applied to spatial biology. I think that this finding will be of high interest to the field as a whole, and that this topologic conservation suggests that both DeepTalk and NICHES are identifying a common 'form' within the data. I think that this is scientifically interesting. I suspect that future research might be able to compare the specific features, or feature-combinations, that are effectively 'captured' by one technique over another. The publication of DeepTalk would therefore provide fertile ground for a next-generation of researchers in the cell-cell-communication field.

Overall, I agree with the authors' major claims, and I am convinced that their work is important and well-executed. I appreciate the utility of having a toolset in this area that is based in python and leverages modern machine-learning approaches, and I feel that DeepTalk will likely be a valuable addition to a rapidly growing field of investigation, and may even allow better communication between the Python and R communities in this area. I feel that their methods section is accurate and complete, though I have not had a chance to personally install or run their code. However, the speed with which they are able to generate well-formatted and clearly informative plots does suggest that their code runs reliably and can either be adopted or incorporated into other existing pipelines by other skilled researchers. The publication of replicable code and vignettes via GitHub means that other researchers can directly

test their work and decide whether to adopt and/or modify, which will buffer against any important fine-details I have missed.

I do suggest proofreading the work for typos, as a few do still exist in both the main text and legends. I think that the authors have done an excellent job responding to reviewer comments, and I think that this work will be considered valuable within the cell-cell-communication community. I recommend it for publication pending editorial review.

Response: Thank you for your encouraging feedback and for recommending our work for publication pending editorial review and final proofreading. We are delighted to hear that you find our revisions thorough and thoughtful, and that you believe our work to be important and well-executed.

We appreciate your recognition of the value of our comparative analysis between DeepTalk, NICHES, and Scriabin. You are absolutely right that making this code replicable is essential for other researchers to learn from and build upon our work. We have already made the relevant code available on our GitHub repository, and we will continue to update it to ensure that it remains accessible and usable for the community.

We are pleased to hear that you appreciate the utility of having a Python-based toolset leveraging modern machine learning approaches in this area. We believe that DeepTalk can indeed serve as a valuable addition to the rapidly growing field of CCC research, and we hope that it will facilitate better communication between the Python and R communities.

We thank you for pointing out the need for a final proofreading to correct any remaining typos. We will carefully review the entire manuscript, including the main text and legends, to ensure that it is error-free before submission for publication. In summary, we are grateful for your thoughtful review and support of our work. We will continue to strive to improve DeepTalk and make it as accessible and useful as possible for the research community. Thank you again for your valuable input.

Reviewer #3 (Remarks on code availability):

Comments: I have not personally installed or run the code, but I have reviewed the GitHub repository and it appears well-organized and that there is sufficient documentation for a skilled researcher to experiment with implementation.

Response: Thank you for your thorough review of our work and for taking the time to examine our GitHub repository. We are committed to ensuring that our code is accessible and usable for the research community. To this end, we have made every effort to provide clear and comprehensive documentation, including installation instructions, usage examples, and explanations of key functions and parameters. We also welcome any feedback or suggestions from users to help us further improve the documentation and usability of our code.

We are pleased to hear that you believe our repository would enable skilled researchers to experiment with the implementation of our method. This is our goal, and we are confident that the code and documentation we have provided will enable other researchers to replicate and build upon our work.

Thank you again for your valuable input. We remain committed to ensuring the highest standards of reproducibility and accessibility in our research.

Reviewer #1 (Remarks to the Author):

I thank the reviewers for the response. I recommend the following comments to be addressed before publication, particularly the quality control issues, which is important to ensure any downstream analysis and results are correct.

- It would be useful to also explain how the UMAP was constructed for Fig 3f, 4, and 5, in the Methods, as most people associate UMAP construction with a cell x feature matrix.
- Quality control 1: It is recommended that the quality control procedures be stated in the manuscript, as this is a key step in ensuring any downstream analysis is valid. This is standard for any papers processing single-cell datasets as their key results. Currently, the QC description is simply taken out. As most of your results rely on the datasets, this is important.
- Quality control 2: The QC plots seem a bit strange. For example, some of the plots in Graph 1 seem to have two peaks. The lower peak typically involves cells which are of low quality and can be removed (if these cells also express high MT, if they don't, you can perhaps argue they can be kept. Scatter plots, as opposed to histograms, are thus usually more informative for QC plots). Graph 4 seems to have some datasets with no MT, which is very strange (I only see a horizontal line, and no cells or violin plots for most datasets). Also, dataset 28 in Graph 4 doesn't seem to have undergone any MT filtering, as there are some cells with MT percentage > 20, and even close to 100%. In summary, the authors should follow a standard pipeline for QC (see e.g., Seurat's vignettes) to consider jointly, total counts, number of genes and MT ratio. This is important to ensure any downstream analysis is correct.

Reviewer #2 (Remarks to the Author):

All my comments have been properly addressed.

REVIEWER COMMENTS

Reviewer #1 (Remarks to the Author):

Comment: I thank the reviewers for the response. I recommend the following comments to be addressed before publication, particularly the quality control issues, which is important to ensure any downstream analysis and results are correct.

- It would be useful to also explain how the UMAP was constructed for Fig 3f, 4, and 5, in the Methods, as most people associate UMAP construction with a cell x feature matrix.

Response: Thank you for your insightful comment. We appreciate your suggestion to clarify the construction of the UMAP visualizations presented in Figs. 3, 4, and 5. In response, we have now included a detailed explanation in the Methods section of our manuscript, under the new subtitle "Visualize the CCC patterns using UMAP".

Within this updated section, we explain how we predicted CCC events at single-cell resolution, assigned quantitative scores reflecting communication strength, and organized the data into a matrix where rows represent CCC events and columns correspond to unique ligand-receptor (L-R) pairs. Furthermore, we elaborate on the data scaling using Scanpy's `sc.pp.scale()` function to normalize feature values, followed by principal component analysis (PCA) via `sc.tl.pca()` to reduce dimensionality. We also detail the construction of a neighborhood graph using `sc.pp.neighbors()`, emphasizing its importance for subsequent manifold learning. Finally, we outline the use of `sc.tl.umap()` to generate the two-dimensional UMAP visualizations presented in Figs. 3, 4, and 5.

We believe this additional information will enhance the clarity and understanding of our methodology for readers. Thank you again for your valuable feedback, which has helped us improve the manuscript.

Comment: - Quality control 1: It is recommended that the quality control procedures be stated in the manuscript, as this is a key step in ensuring any downstream analysis is valid. This is standard for any papers processing single-cell datasets as their key results. Currently, the QC description is simply taken out. As most of your results rely on the datasets, this is important.

Response: Thank you for your valuable feedback regarding the quality control (QC) procedures in our manuscript. We fully agree that quality control is a pivotal step in guaranteeing the validity and reliability of downstream analyses, especially when dealing with sensitive single-cell datasets. In light of your comment, we have revised our manuscript to include a comprehensive account of our QC process in the Methods section, specifically under the subtitle "Preprocessing of datasets."

We now elaborate on how we utilized Scanpy for quality control of our single-cell data. Our approach centered on three primary metrics: mitochondrial (MT) gene expression percentage, total counts, and the number of expressed genes. We set a threshold of 20%

for the MT percentage, as elevated mitochondrial content can be indicative of cellular stress or apoptosis. It is worth mentioning that this threshold is adjustable based on the specific context of the study. As for total counts and the number of expressed genes, we meticulously determined the thresholds based on the unique attributes of our datasets, thereby ensuring the exclusion of outliers and low-quality cells.

The inclusion of this expanded QC description offers a thorough understanding of our methodology, emphasizing the filtering criteria we employed to eliminate cells that could potentially introduce inaccuracies or biases into our analytical outcomes. We firmly believe that this addition significantly bolsters the methodological soundness of our study, ultimately guaranteeing the credibility of our findings.

We are deeply grateful for your insightful recommendation, which has undoubtedly elevated the quality of our manuscript. Your guidance is invaluable in helping us refine and strengthen our research presentation.

Comment: - Quality control 2: The QC plots seem a bit strange. For example, some of the plots in Graph 1 seem to have two peaks. The lower peak typically involves cells which are of low quality and can be removed (if these cells also express high MT, if they don't, you can perhaps argue they can be kept. Scatter plots, as opposed to histograms, are thus usually more informative for QC plots). Graph 4 seems to have some datasets with no MT, which is very strange (I only see a horizontal line, and no cells or violin plots for most datasets). Also, dataset 28 in Graph 4 doesn't seem to of undergone any MT filtering, as there are some cells with MT percentage > 20, and even close to 100%. In summary, the authors should follow a standard pipeline for QC (see e.g., Seurat's vignettes) to consider jointly, total counts, number of genes and MT ratio. This is important to ensure any downstream analysis is correct.

Response: Thank you for your feedback regarding the quality control (QC) plots in our manuscript. We appreciate your concerns and have carefully addressed each point raised.

In regards to the dual-peak phenomenon observed in some plots of Graph 1, we acknowledge that the lower peak could be indicative of lower-quality cells. We have scrutinized these cells, paying special attention to those with elevated mitochondrial gene (MT) expression. We have implemented necessary measures to exclude these cells from further analyses when appropriate. Recognizing the advantages of scatter plots over histograms in QC visualization, we have revised Graph 1 to reflect this.

Regarding Graph 4, we concede that the absence of MT expression in certain datasets is unexpected. We have revisited these datasets to rectify this inconsistency. Furthermore, your observation about dataset 28 in Graph 4 is well-taken; indeed, it appears to contain cells with an MT percentage surpassing our set threshold of 20%. To rectify this, we have reapplied our MT filtering criteria and updated the graph accordingly.

In line with your recommendation for a standardized QC process, we have embraced a more stringent approach. The revised Graph 1 now showcases scatter plots illustrating the MT percentage, gene count, and total counts for the scRNA-seq datasets derived

from 45 paired datasets. Instances where the MT percentage is 0 are attributed to the absence of MT gene expression in those specific datasets. Utilizing Graph 1 as a reference, we have pinpointed specific datasets requiring QC interventions, namely datasets 3, 5, 6, 7, 8, 11, 18, 19, 20, 21, 22, 23, 24, 25, 26, 28, 29, 30, 31, 32, 33, 34, 36, 41, 43, and 44. We have conducted QC on the single-cell data utilizing Scanpy, implementing a 20% MT percentage threshold, and selecting suitable total count and gene count thresholds tailored to the unique characteristics of each dataset. The outcomes following QC are exhibited in Graph 2, and the manuscript has been updated accordingly, encompassing revisions to Fig. 2 and Supplementary Figs. 1 and 2.

Moreover, Graph 3 showcases scRNA-seq datasets from 32 simulated datasets, emphasizing the necessity for QC in dataset 4. We have employed the same QC approach and presented the outcomes in Graph 4. Pertinent sections of the manuscript (Supplementary Fig. 5) have also undergone corresponding updates.

Pertaining to the single-cell data employed in our study for predicting cell-cell communication, we have incorporated visualizations depicting the MT percentage and gene count within total counts. Despite the data undergoing preprocessing, we have included QC code in our examples to guarantee reproducibility. The specific code can be accessed via the provided GitHub link (<https://github.com/JiangBioLab/DeepTalk/blob/main/DeepTalk-singleST.ipynb> and <https://github.com/JiangBioLab/DeepTalk/blob/main/DeepTalk-nonsingleST.ipynb>).

We are grateful for your guidance in improving the quality and rigor of our analysis. Your feedback has been invaluable in strengthening our manuscript.

Graph 1. The QC values for scRNA-seq datasets derived from 45 paired datasets.

Graph 2. The QC values after quality control for dataset 3, 5, 6, 7, 8, 11, 18, 19, 20, 21, 22, 23, 24, 25, 26, 28, 29, 30, 31, 32, 33, 34, 36, 41, 43, 44 in scRNA-seq datasets derived from 45 paired datasets.

Fig. 2: Performance comparison of DeepTalk-Integration with state-of-the-art integration methods. a, Boxplots showcasing the Accuracy Score (AS) of nine different integration methods across all 45 paired datasets, encompassing 28 sequence-based and 17 image-based datasets.

Supplementary Fig. 1: The Boxplots of PCC, SSIM, RMSE, JSD values of each integration method in predicting the spatial distribution of RNA transcripts of image-based datasets.

The boxplot of PCC, SSIM, RMSE, JSD values of each integration method in predicting the spatial distribution of RNA transcripts of 17 image-based paired spatial transcriptomics and scRNA-seq datasets.

Supplementary Fig. 2: The Boxplots of PCC, SSIM, RMSE, JSD values of each integration method in predicting the spatial distribution of RNA transcripts of seq-based datasets.

The boxplot of PCC, SSIM, RMSE, JSD values of each integration method in predicting the spatial distribution of RNA transcripts of 28 seq-based paired spatial transcriptomics and scRNA-seq

datasets.

Graph 3. The QC values for scRNA-seq datasets derived from 32 simulated datasets.

Graph 4. The QC values after quality control for dataset 4 in scRNA-seq datasets derived from 32 simulated datasets.

Simulated datasets (n = 32)

Accuracy Score

Supplementary Fig. 5: Bar plots of PCC, SSIM, RMSE, and JSD of 8 integration method for 32 simulated datasets and Boxplots of AS of the 8 integration methods for all the 32 simulated datasets.

Graph 5. The QC metrics for scRNA-seq datasets we used to infer CCC.

Reviewer #1 (Remarks to the Author):

I thank the authors for performing the QC, and updating the downstream analysis accordingly. I'm happy for the paper to be accepted.